# PREDICTD PaRallel Epigenomics Data Imputation with Cloud-based Tensor Decomposition

Timothy J. Durham [1], Maxwell W. Libbrecht[1], J. Jeffry Howbert [1], Jeff Bilmes[2] & William Stafford Noble [1,3]

The Encyclopedia of DNA Elements (ENCODE) and the Roadmap Epigenomics Project seek to characterize the epigenome in diverse cell types using assays that identify, for example, genomic regions with modified histones or accessible chromatin. These efforts have produced thousands of datasets but cannot possibly measure each epigenomic factor in all cell types. To address this, we present a method, PaRallel Epigenomics Data Imputation with Cloud-based Tensor Decomposition (PREDICTD), to computationally impute missing experiments. PREDICTD leverages an elegant model called "tensor decomposition" to impute many experiments simultaneously. Compared with the current state-of-the-art method, ChromImpute, PREDICTD produces lower overall mean squared error, and combining the two methods yields further improvement. We show that PREDICTD data captures enhancer activity at noncoding human accelerated regions. PREDICTD provides reference imputed data and open-source software for investigating new cell types, and demonstrates the utility of tensor decomposition and cloud computing, both promising technologies for bioinformatics.

[1] Department of Genome Sciences, University of Washington, Foege Building S-250, Box 355065, 3720 15th Ave NE, Seattle, WA 98195, USA. [2] Department of Electrical Engineering, University of Washington, Paul Allen Center AE100R, Box 352500, 185 Stevens Way, Seattle, WA 98195, USA. [3] Department of Computer Science and Engineering, University of Washington, Foege Building S-250, Box 355065, 3720 15th Ave NE, Seattle, WA 98195, USA. Correspondence and requests for materials should be addressed to W.S.N. (email: william-noble@uw.edu)

nderstanding how the genome is interpreted by varied cell types, in different developmental and environmental contexts, is the key question in biology. With the advent of high-throughput next-generation sequencing technologies, over the past decade, we have witnessed an explosion in the number of assays to characterize the epigenome and interrogate the chromatin state, genome wide. Assays to measure chromatin accessibility (DNase-seq, ATAC-seq, FAIRE-seq), DNA methylation (RRBS, WGBS), histone modification, and transcription factor binding (ChIP-seq) have been leveraged in large projects, such as the Encyclopedia of DNA Elements (ENCODE)[1] and the Roadmap Epigenomics Project[2] to characterize patterns of biochemical activity across the genome in many different cell types and developmental stages. These projects have produced thousands of genome-wide datasets, and studies leveraging these datasets have provided insight into multiple aspects of genome regulation, including mapping different classes of genomic elements[3,4], inferring gene regulatory networks[5], and providing insights into possible disease-causing mutations identified in genome-wide association studies (GWAS)[2].

Despite the progress made by these efforts to map the epigenome, much work remains to be done. Due to time and funding constraints, data have been collected for only a fraction of the possible pairs of cell types and assays defined in these projects (Fig. 1a). Furthermore, taking into account all possible developmental stages and environmental conditions, the number of possible human cell types is nearly infinite, and it is clear that we will never be able to collect data for all cell type/assay pairs.

However, understanding the epigenome is not an intractable problem, because in reality many of the assays detect overlapping signals, such that most of the unique information can be recovered from just a subset of experiments. One solution is thus to prioritize experiments for new cell types, based on analysis of existing data[6]. Alternatively, one may exploit existing data to accurately impute the results of missing experiments.

Ernst and Kellis pioneered this imputation approach, and achieved remarkable accuracy with their method, ChromImpute[7]. Briefly, this method imputes data for a particular target assay in a particular target cell type by: (1) finding the top ten cell types most correlated with the target cell type, based on data from non-target assays, (2) extracting features from the data for the target assay from the top ten non-target cell types, and also extracting features from the data for non-target assays in the target cell type, and (3) training a regression tree for each of the top ten most correlated cell types. Data points along the genome are imputed as the mean predicted value from the collection of trained regression trees. Although ChromImpute produces highly accurate imputed data, this training scheme is complicated and not very intuitive, and results in a fragmented model of the epigenome that is very difficult to interpret. We hypothesized that an alternative approach, in which a single joint model learns to impute all experiments at once, would simplify the model training and improve the interpretability, while maintaining accurate imputation of missing data.

Accordingly, we present PaRallel Epigenomics Data Imputation with Cloud-based Tensor Decomposition (PREDICTD),

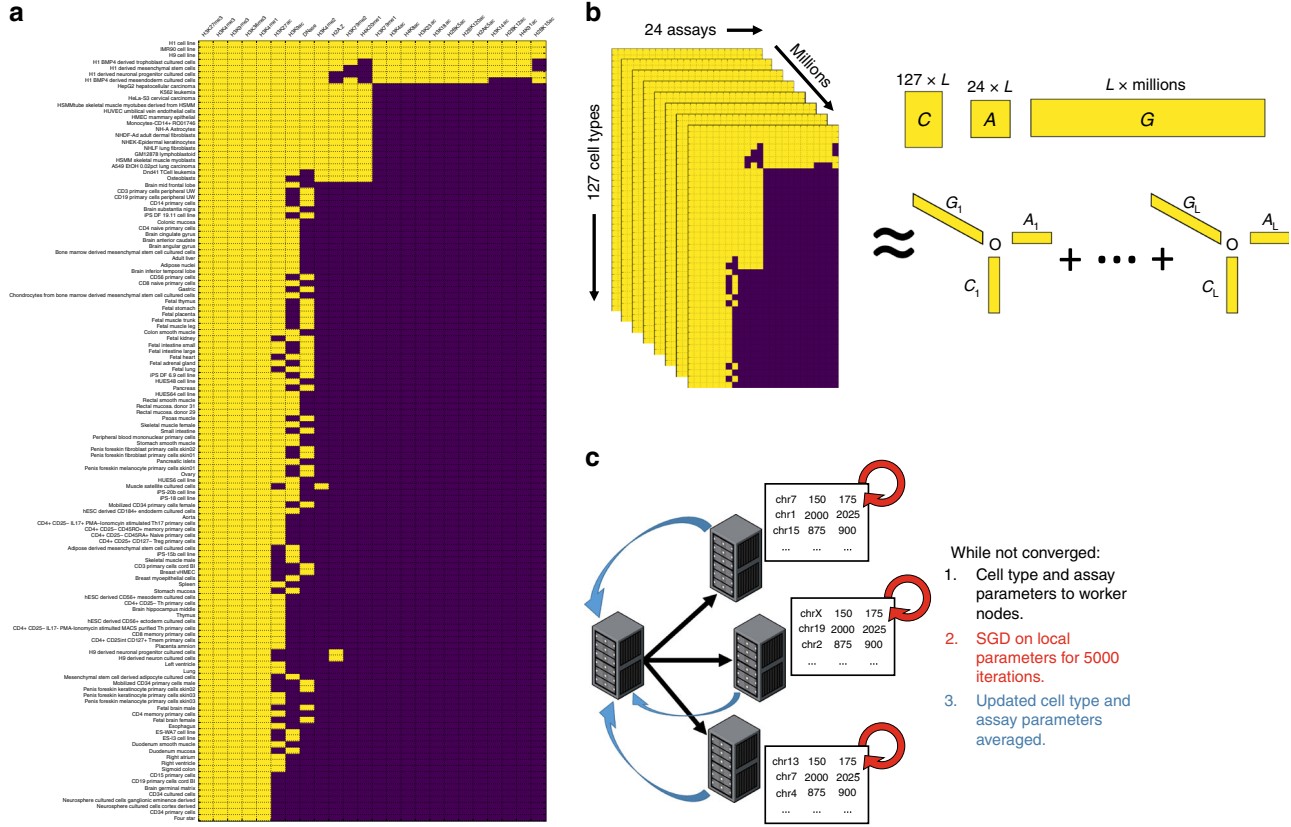

**Fig. 1** Overview. **a** Matrix representing the subset of the Roadmap Epigenomics consolidated data set used in this study. Experiments in yellow have observed data, while missing experiments are purple. **b** We model the experiments in **a** as a three-dimensional tensor, and find three low-rank factor matrices (denoted $C$, $A$, and $G$) that can be combined by summing the outer products of each of the $L$ latent factor vector triplets to reconstruct a complete tensor with no missing values that both approximates the existing data and imputes the missing data. **c** The genome dimension is very large, so in order to fit all of the data in memory and to speed up training, we distribute the tensor across multiple cluster nodes running Apache Spark. Then we use parallel stochastic gradient descent[37] to share the $A$ and $C$ matrices across all nodes

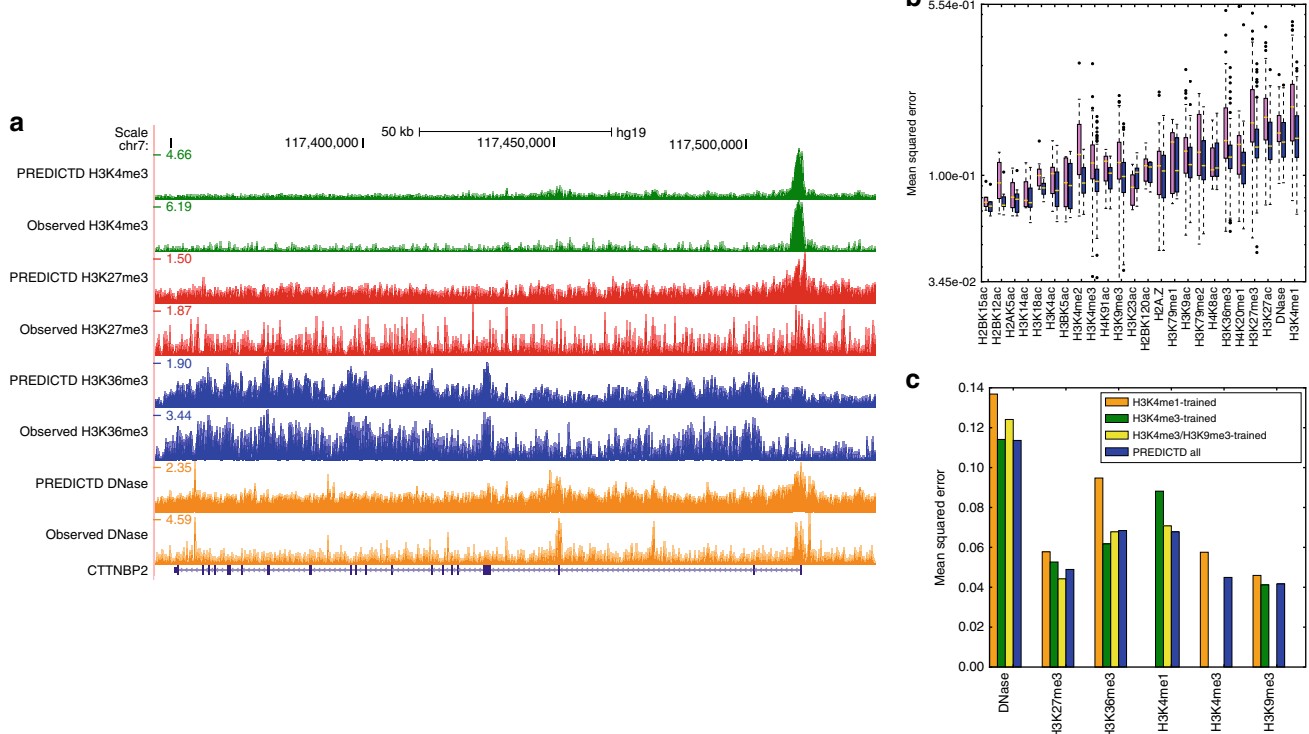

**Fig. 2** PREDICTD imputes missing epigenomics data with high accuracy. **a** Comparison of PREDICTD and observed data in H1 embryonic stem cells, which is one of the three cell types with observed data for all assays. For each assay, the top track is PREDICTD and the bottom is observed. See Supplementary Fig. 1 for more assays. **b** Global mean squared error (MSEglobal) for each dataset, shown as box plots for the baseline Main Effects model (pink), and PREDICTD (blue). For each distribution, the box shows the inter-quartile range (IQR), whiskers show 1.5 times the IQR, and flier points show individual datasets that are outliers. The median is indicated by a horizontal gold line on each box plot. See Supplementary Fig. 2 for more quality measures. **c** Simulated analysis of a cell type with few available datasets. The bar graph shows the MSEglobal measure for the six available assays for the "CD3 primary cells from cord blood" cell type on four models. In blue is the full PREDICTD model shown in **a** and **b**, while the other bars represent models trained with all observed data for other cell types, but only H3K4me1 (orange), only H3K4me3 (green), or only H3K4me3 and H3K9me3 (yellow) for "CD3 primary cells from cord blood." See Supplementary Fig. 3 for more quality measures

which treats the imputation problem as a tensor completion task, and employs a parallelized algorithm based on the PARAFAC/ CANDECOMP method[8,9,]. Our implementation, developed on consumer cloud infrastructure, achieves high accuracy imputation of ENCODE and Roadmap Epigenomics data, and predicts all datasets jointly in a single model. We used PREDICTD to impute the results for 3048 experiments across 127 cell types and 24 assays from the Roadmap Epigenomics project, and these imputed data are available for download through ENCODE (https://www.encodeproject.org/). In the following sections, we explain the model, discuss its performance on held out experiments from the Roadmap Epigenomics Consolidated data[2], show that the model parameters summarize biologically relevant features in the data, and demonstrate that imputed data can recapitulate important cell type-specific gene regulatory signals in noncoding human accelerated regions (ncHARs) of the genome[10].

## Results

**Epigenomic maps can be imputed using tensor factorization**. Data from Roadmap and ENCODE projects can be organized into a 3D tensor, with axes corresponding to cell types, assays, and genomic positions (Fig. 1b). This tensor is long and skinny, with many fewer cell types and assays than genomic positions, and the data for experiments that have not been done yet are missing in the tensor fibers along the genome dimension. Our strategy for imputing these fibers is to jointly learn three factor matrices that can be combined mathematically to produce a complete tensor that both approximates the observed data and predicts the missing data. These three factor matrices are of shape $C \times L$, $A \times L$, and $G \times L$, where $C$, $A$, and $G$ indicate the numbers of cell types, assays, and genomic positions, respectively, and $L$ indicates the number of "latent factors" that the model trains (Fig. 1b), and thus the number of model parameters.

We developed and trained our implementation of this tensor-factorization model, PREDICTD, using 1014 datasets from the Roadmap Epigenomics Consolidated Epigenomes[2] (Fig. 1a). To assess model performance, we split the datasets into five training/ test splits, and we report the results of imputing each test set at 25 base pair resolution. The model training proceeds by distributing the data and genome parameters across the nodes of the cluster, and then sharing the cell type and assay parameters across all nodes, using a parallelized training procedure (See Methods, Fig. 1c). We find that training on a randomly selected 0.01% of the genome provides enough signal for learning the cell type and assay parameters; these parameters are then applied across all genomic positions of interest by training the genome parameters for each position while holding the cell type and assay parameters constant. We report the results from imputing just over 1% of the genome, including the ENCODE Pilot Regions[1] and 2640 ncHARs[10]. All subsequent references to the genome dimension in this manuscript refer to this subset of loci.

Our model formulation and implementation offer several important advantages. First, training a single model to impute all datasets at once is a straightforward and intuitive way of solving

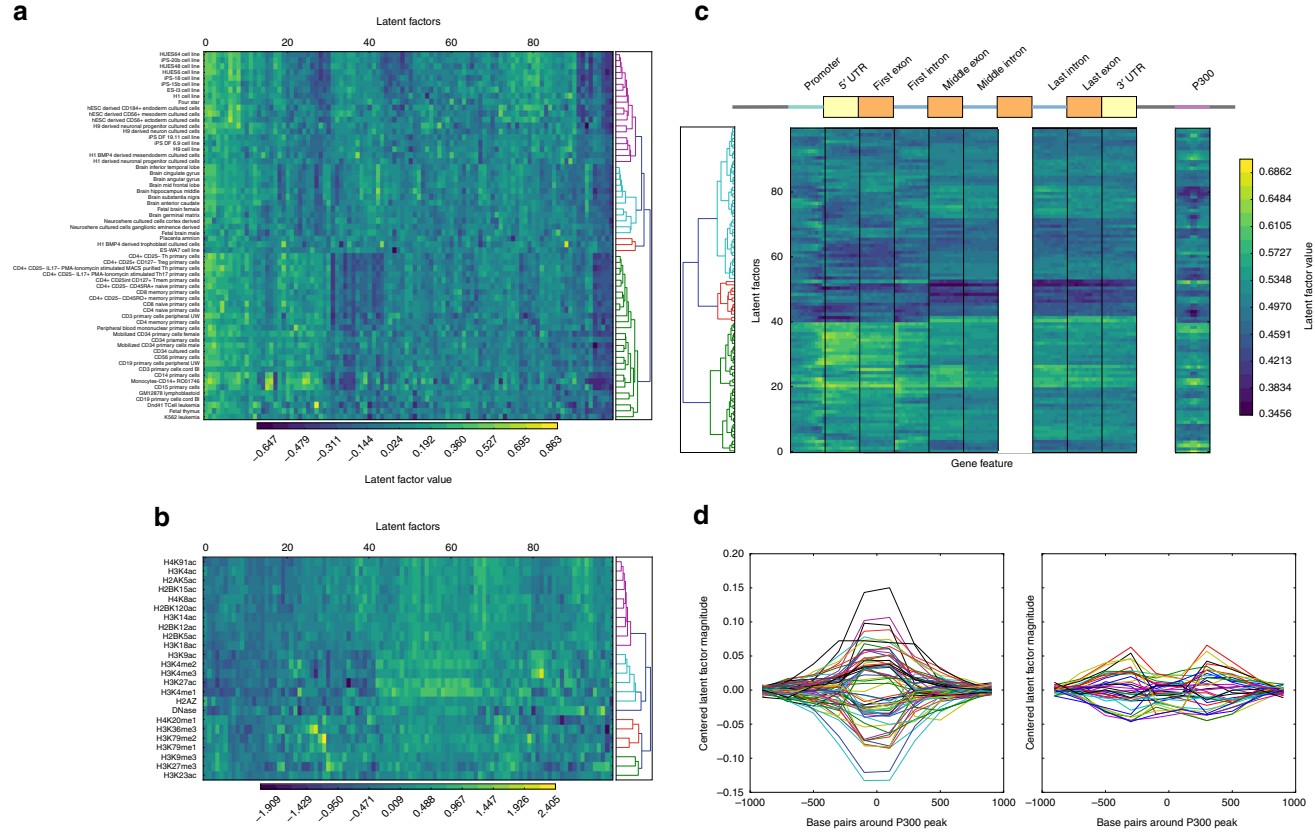

**Fig. 3** The model parameters can distinguish among elements in each tensor dimension. Plots show the values (or average values) for 100 latent factors from one of the models trained for this manuscript, and that these values show patterns that distinguish among cell types, assays, and genomic elements. **a** Hierarchical clustering of the cell types, based on cell type factor matrix values, shows that similar cell types tend to cluster together. This is a subset of cell types; for a clustering of all cell types see Supplementary Fig. 9. **b** Hierarchical clustering of the assays, based on assay factor matrix values, shows that similar assays tend to cluster together. **c** Average values from the genome-factor matrix show different patterns at different parts of the gene, and P300 peaks called from ENCODE data. **d** Average values from the genome factor matrix, for each latent factor, plotted as a line spanning the region ±1 kb around the center of P300 peaks. Parameter values are centered at zero and plotted based on whether they show the highest magnitude at the peak (left, 64 latent factors) or flanking the peak (right, 36 latent factors)

this problem. Second, as we demonstrate below, the model can leverage the joint training to perform well even in cell types with a single informative experiment. Third, the parameters of the trained model have the same semantics across all input datasets and, although a full investigation of model interpretability is outside the scope of this work, we show that the trained parameters show different patterns for different cell types, assays, and genomic elements. We take these results as evidence that the PREDICTD model itself holds the potential to be interrogated to learn about relationships among assays, cell types, and genomic loci. Last, PREDICTD software is open-source (https://bitbucket. org/noblelab/predictd), and is also implemented and distributed on the consumer cloud, which makes our model immediately accessible to, and easily runnable by, nearly anyone.

**PREDICTD imputes epigenomics experiments with high accuracy.** PREDICTD imputes missing data with high accuracy, based on both visual inspection and quality measures (Fig. 2). Visually, the imputed signal pattern closely matches that of observed data, and recapitulates the known associations of epigenomic marks with genomic features (Fig. 2a, Supplementary Fig. 1). For example, as expected, H3K4me3-imputed signal is strongly enriched in narrow peaks at promoter regions near the transcription start site of active genes, and H3K36me3, known to mark transcribing regions, is enriched over gene bodies.

We also show strong performance of PREDICTD on ten different quality measures (see Methods, Supplementary Figs 2, 14, 15, Supplementary Data 13–16), especially the global mean squared error quality measure (MSEglobal). As a key part of the PREDICTD model's objective function, MSEglobal is explicitly optimized during model training (see Methods). The MSEglobal measure has a mean of 0.1229, and it ranges from 0.0359 for H3K4me3 in the "NHLF Lung Fibroblasts" cell type to 0.4511 for H4K20me1 in "Monocytes CD14+ RO01746." Other key quality measures include the genome-wide Pearson correlation (GWcorr, mean: 0.6886, min: 0.0790 for H3K36me3 in "Right Atrium," max: 0.9391 for H3K4me3 in "HUES64 Cell Line"), and the area under the receiver operating characteristic curve for recovering observed peak regions from imputed data (CatchPeakObs, mean: 0.9565, min: 0.5503 for H3K36me3 in "Right Atrium," max: 0.9984 for H3K4me3 in "NHLF Lung Fibroblasts"). Note that seven of our ten quality measures, including GWcorr and CatchPeakObs, were also used in the ChromImpute publication[7].

As a baseline, we compared the performance of PREDICTD to a simple "Main Effects" model, which computes the global mean of the observed data, and then the column and row residuals of each two-dimensional (2D) slice of the tensor along the genome dimension, and imputes a given cell in the tensor by summing the global mean and the corresponding row and column residual

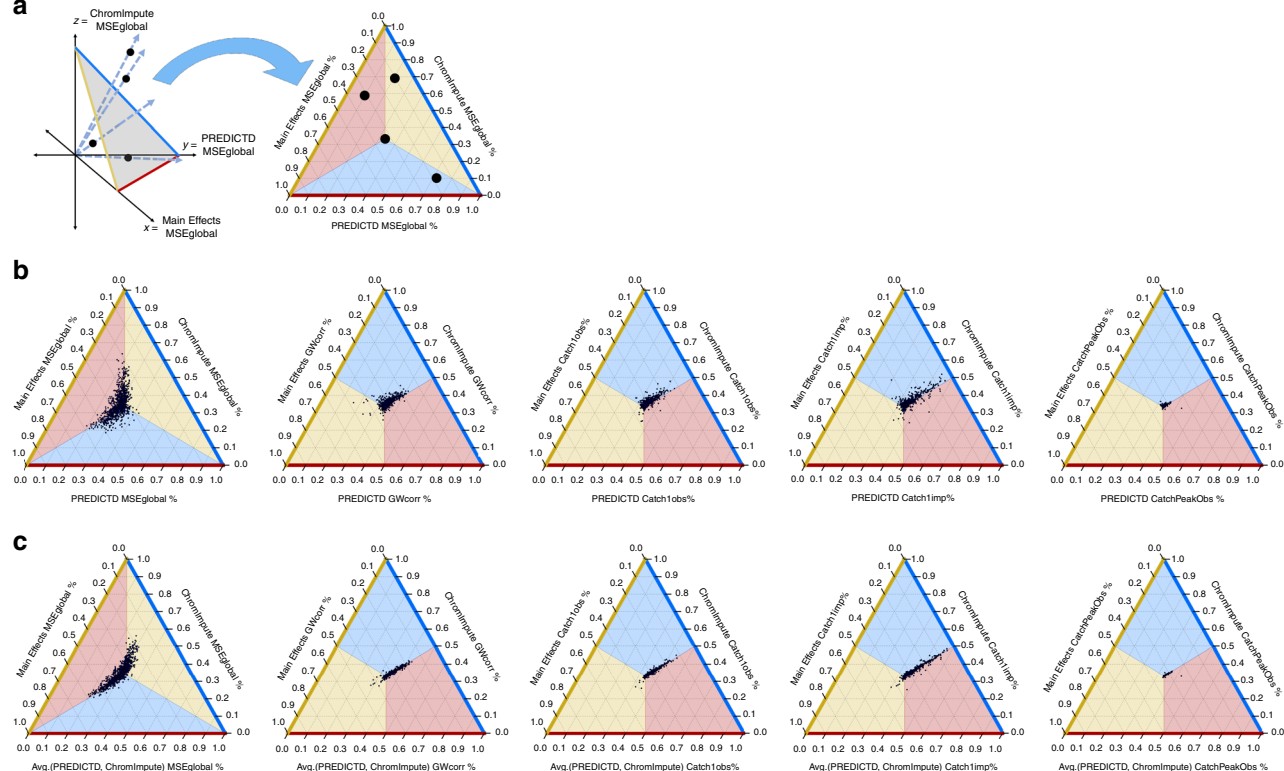

**Fig. 4** PREDICTD performs comparably to ChromImpute, and combining the models improves the result. **a** Schematic describing how a ternary plot relates to Cartesian coordinates. Each experiment (represented by a black dot) is plotted in Cartesian space based on the values of a particular quality score for imputed data (in this example, MSEglobal) from PREDICTD, ChromImpute, and Main Effects. Each point in this space is then projected onto a plane by a vector drawn through the point and the origin. The resulting ternary plot summarizes the relative magnitude of the quality score for the three models. If all models achieve the same quality measure score for a particular experiment, then that point will be projected onto the center of the ternary plot. Deviation toward a corner of the triangle indicates that one model has a higher value for that quality measure than the other two, and deviation from the center toward one of the edges of the triangle indicates that one model has a lower value. Color shading of the plot area marks the regions of the ternary plot that indicate superior performance of each model on a particular quality measure. The pattern of the colors changes based on whether it is better to have a low value on that quality measure (as with mean squared error) or a high value (for example, the genome-wide correlation). **b** Comparing PREDICTD, ChromImpute, and Main Effects models across five quality measures: the global mean squared error (MSEglobal), the genome-wide Pearson correlation (GWcorr), the percent of the top 1% of observed data windows by signal value found in the top 5% of imputed windows (Catch1obs), the percent of the top 1% of imputed windows by signal value found in the top 5% of observed windows (Catch1imp), and the area under the receiver operating characteristic curve for recovery of observed peak calls from all imputed windows ranked by signal value (CatchPeakObs). **c** The same as in **b**, except that the quality measures for the averaged results of ChromImpute and PREDICTD are plotted along the bottom (red) axis, instead of the measures of PREDICTD alone

means. PREDICTD outperforms this baseline model for MSE-global on all but two assays (Fig. 2b). Furthermore, PREDICTD similarly outperforms the Main Effects on all additional performance measures (Supplementary Fig. 2).

**PREDICTD performs well on cell types with few assays**. The key application of PREDICTD will be to impute results for cell types that may have only one or two datasets available. To investigate the performance of PREDICTD in this context, we trained a model on all available data for all cell types, except that we only included one or two experiments for the "CD3 cord blood primary cells" cell type. In particular, one model had just H3K4me1 in the training set for this cell type, one had just H3K4me3, and one had both H3K4me3 and H3K9me3. Comparing the performance measures between these experiments and the imputed results from our original models trained on the five test sets, we find that the results of training with just H3K4me3 or both H3K4me3 and H3K9me3 are nearly as good as (and sometimes better than) the results from the original models with training data that included five or six experiments for this cell type (Fig. 2c, Supplementary Fig. 3). Imputing based on H3K4me1 signal did not perform as well as imputing based on

only H3K4me3. This observation is consistent with previous results of assay prioritization[6] indicating that H3K4me3 is the most information-rich assay. Furthermore, this result is not specific to the "CD3 cord blood primary cells" cell type. We find that the results for imputing four other cell types ("GM12878 lymphoblastoid," "fetal muscle trunk," "brain anterior caudate," and "lung"), just based on H3K4me3 signal, showed similar results (Supplementary Figs. 5, 6, 7, 8). We conclude that PRE-DICTD performs well on under-characterized cell types, and will be useful for studying new cell types for which few datasets are currently available.

**Model parameters capture patterns in each tensor dimension**. The fact that PREDICTD performs well on the imputation task implies that the parameters learned by the model capture patterns that can distinguish among different cell types, assays, and genomic positions, and we next present results showing that this is the case. We think it important to note that it would be incorrect, at this point, to interpret any particular latent factor as having a specific biological meaning. We place no a priori constraints on what patterns in the data PREDICTD uses to arrive at a solution, and any signal with relevance to a particular biological

**Table 1 Statistics comparing models across five quality measures show PREDICTD outperforms Main Effects and has similar performance to ChromImpute**

| Measure | PREDICTD vs ChromImpute | | | PREDICTD vs Main Effects | | | Main Effects vs ChromImpute | | |
|---|---|---|---|---|---|---|---|---|---|
| | Corr | Log ratio | | Corr | Log ratio | | Corr | Log ratio | |
| | | Mean | Std | | Mean | Std | | Mean | Std |
| MSEglobal | 0.689 | −0.151 | 0.266 | 0.835 | −0.188 | 0.212 | 0.510 | 0.037 | 0.373 |
| GWcorr | 0.977 | −0.039 | 0.072 | 0.883 | 0.100 | 0.163 | 0.866 | −0.139 | 0.164 |
| Catch1obs | 0.979 | −0.028 | 0.055 | 0.916 | 0.097 | 0.161 | 0.886 | −0.125 | 0.177 |
| Catch1imp | 0.973 | −0.023 | 0.073 | 0.876 | 0.155 | 0.293 | 0.848 | −0.178 | 0.306 |
| CatchPeakObs | 0.923 | −0.008 | 0.017 | 0.776 | 0.025 | 0.037 | 0.812 | −0.032 | 0.035 |

See Supplementary Data 1 for the statistics on all quality measures.

feature is likely distributed across multiple latent factors. As such, here we simply show that the parameters, in aggregate, exhibit different patterns between different cell types, assays, and genomic loci; a full investigation on the ways to gain biological insight, from these parameters, is outside the scope of our present study.

Although we cannot definitively assign semantics to individual latent factors, we find that their values in aggregate show patterns that recover known relationships among the cell types, assays and genomic loci (Fig. 3). Hierarchical clustering on the rows of the cell type factor matrix shows that similar cell types are grouped together (Fig. 3a), producing large clades for embryonic stem cells (magenta), immune cell types (green), and brain tissues (cyan), among others (Fig. 3a, Supplementary Fig. 9). In the same way, assays with similar targets cluster together (Fig. 3b), with the colored clades from top to bottom representing acetylation marks generally associated with active promoters (magenta), marks that are strongly associated with active regulatory regions (cyan/blue), and broad marks for transcription (red) and repression (green). The assays cluster perfectly except that, biologically, H3K23ac should be grouped with either the active regulatory marks (cyan/blue) or the active acetylation marks (magenta). This is one of the two assays for which PREDICTD failed to outperform the Main Effects, and it was one of the worst performing assays for ChromImpute as well, so it appears to be a difficult mark to impute. Nevertheless, most of the cell types and assays cluster correctly, and these results are highly nonrandom. We quantified this by comparing our clustering results to randomly shuffled cluster identities, using the Calinski–Harabaz Index, which assesses how well the clustering separates the data by comparing the average distance among points between clusters to the average distance among points within clusters (Supplementary Fig. 10).

For the genome factor matrix, we projected the coordinates of each gene, from the GENCODE v19 human genome annotation[11] (https://www.gencodegenes.org/releases/19.html), onto an idealized gene model that includes nine parts from 5′ to 3′ in the gene: the promoter, 5′ untranslated region (UTR), first exon, first intron, middle exons, middle introns, last intron, last exon, and 3′ UTR. This procedure produced a summary of the genome latent factors (Fig. 3c) that, when reading each column of the heat map as a feature vector for a particular location in a gene, shows distinct patterns at different gene components. For example, latent factors that on average have high or low values at regions (i.e., heat map columns) near the transcription start site are different from those with high or low values at other gene components, like exons and introns.

In addition to investigating patterns in the genome parameters at genes, we checked to see whether distal regulatory regions showed a pattern distinct from gene components. P300 is a chromatin regulator that associates with active enhancers[12]. We therefore decided to search for patterns in the genome latent factors at windows ±1 kb around annotated P300 sites from published ENCODE tracks (see Methods). Note that no P300 data was used to train PREDICTD. Nevertheless, we find a striking pattern, with many latent factors showing average values of larger magnitude within the 400 bp region surrounding the center of the peak, and some others showing larger average magnitude in a flanking pattern in the bins 200–400 bp away from the peak center (Fig. 3c, d). Again, note that these results do not imply a biological meaning for any particular latent factor; instead, we hypothesize that the genome latent factors as a whole might be useful as features for classification or deeper characterization of genomic elements. Last, if we randomize the latent factors at each genomic location and do the same analyses, we find no discernible pattern (Supplementary Fig. 11). We thus conclude that the trained model parameters encode patterns that correspond to biology.

**PREDICTD and ChromImpute data are similar and complementary.** As described in the Introduction, the ChromImpute method[7] provides high-quality imputed data, but employs a complicated model and training procedure tuned to each individual experiment. In contrast, our tensor decomposition approach imputes all missing experiments by using a single model, which we argue is conceptually simpler and addresses the problem in a more natural way. Furthermore, we find that our model outperforms ChromImpute on our primary performance measure (MSEglobal), and yields similar performance on nine additional measures (Fig. 4a, Table 1, Supplementary Figs. 14, 15, and Supplementary Data 1, 13–16). Also see Supplementary Figs. 4, 2, 3, 5, 6, 7, 8 for figures similar to those in Fig. 2, but with ChromImpute values included. The correlation of quality measures between PREDICTD and ChromImpute is higher than the correlation between the Main Effects method and ChromImpute, indicating that PREDICTD agrees with ChromImpute more often than Main Effects does. Furthermore, the mean log ratio of quality measures on corresponding experiments imputed by PREDICTD and ChromImpute show smaller differences than the log ratios for Main Effects and ChromImpute (Table 1, Supplementary Data 1, Fig. 4, Supplementary Figs. 14, 15). Thus, PREDICTD produces high-quality imputed data that is almost as good, or better than, ChromImpute predictions, depending upon which quality measure is employed.

We also calculated the distribution of the differences between imputed values and observed values for experiments imputed by both PREDICTD and ChromImpute, and we found that ChromImpute tends to impute higher values than PREDICTD (Supplementary Fig. 13). We hypothesized that the two models each perform better on different parts of the genome, and so we tried averaging the PREDICTD and ChromImpute results. By the

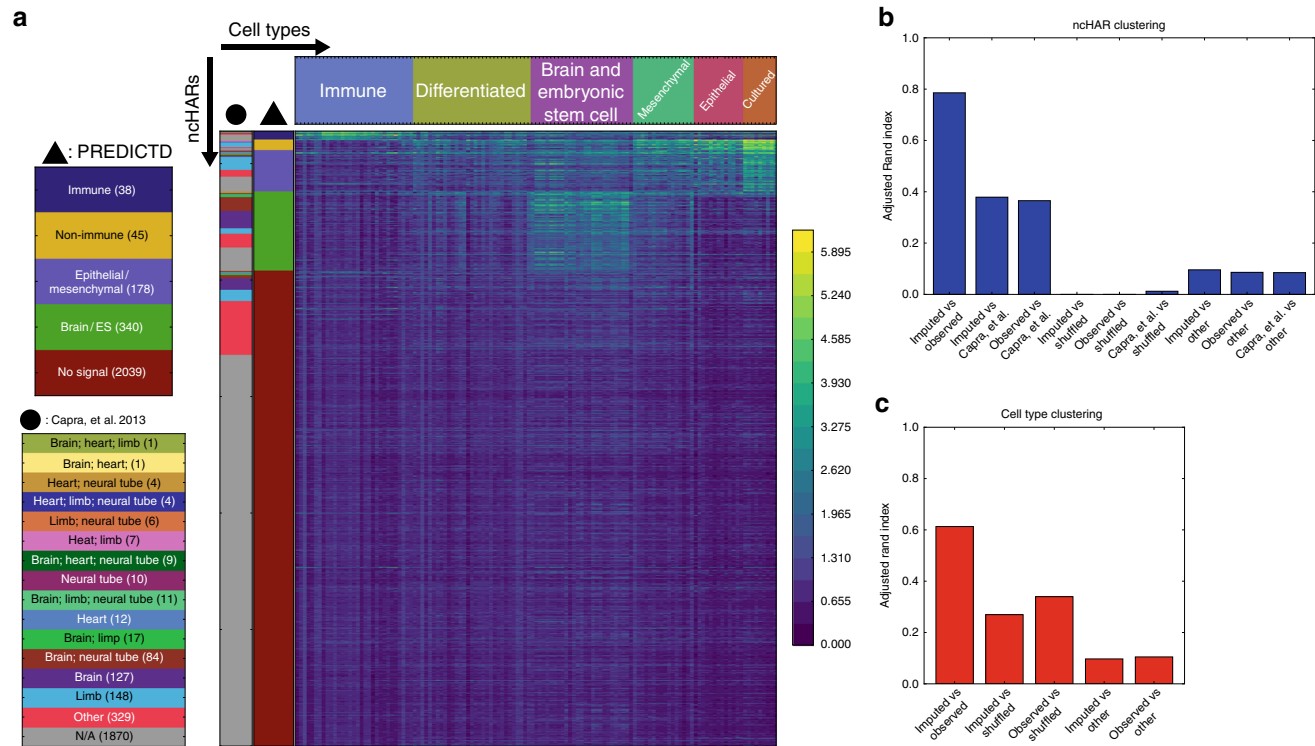

**Fig. 5** Imputation of enhancer marks reveals tissue-specific patterns of enhancer-associated marks at noncoding human accelerated regions (ncHARs). **a** Average PREDICTD signal at each ncHAR was compiled for H3K4me1, H3K27ac, and DNase assays from all cell types. The first principal component with respect to the three assays was used in a biclustering to find six and five clusters along the cell type, and ncHAR dimensions, respectively. The inverse hyperbolic sine-transformed signal from each of these assays was summed per cell type and ncHAR, and the resulting values were plotted as a heat map. The column marked with a black triangle at the top designates the color key for the ncHAR clusters. The leftmost column, designated with a black circle, identifies ncHARs with predicted tissue-specific developmental enhancer activity, based on EnhancerFinder analysis from Capra et al.[10]. **b** and **c** Evaluation of the clustering results with the adjusted Rand index. The clustering results for observed data and PREDICTD for the ncHAR (**b**) and cell type clusterings (**c**), and also those from Capra et al.[10] for the ncHARs, all show higher adjusted Rand index scores than the clustering results for observed data with shuffled ncHAR coordinates (shuffled) or for observed data from non-enhancer-associated marks (other)

MSEglobal measure, we do see a marked improvement relative to both models, and other quality measures on which ChromImpute outperformed PREDICTD alone show parity between ChromImpute and the averaged model (Fig. 4b).

**Imputed data recovers cell type-specific enhancer signatures**. Human accelerated regions (HARs) are genomic loci that are highly conserved across mammals, but harbor more mutations in human than would be expected for their level of conservation (reviewed in ref. [13]). Although some HARs overlap coding regions, the overwhelming majority (>90%) are found in non-coding portions of the genome (ncHARs)[10,13], and ncHARs are thought to be enriched for mutations that affect the regulation of genes underlying human-specific traits. Noncoding variation is thought to account for much of our phenotypic divergence from other primates[14], and additional evidence in support of this hypothesis comes from observations that ncHARs cluster around developmental and transcription factor genes[10,13], transgenic assays for functional validation of enhancer activity[10,15–17], computational epigenomics, and population genetics studies[10,18,19].

In particular, ncHARs are enriched in developmental enhancer activity[10,18]. In[10] EnhancerFinder[18], a program to predict genomic regions with tissue-specific developmental enhancer activity, was trained on ENCODE epigenomics maps[1] and results from the VISTA enhancer database[20], and applied to ncHARs. EnhancerFinder predicted enhancer activity for 773 of 2649 ncHARs, but the authors note that the characterization of these

regions remains incomplete due to limitations in the available data. To our knowledge, no one has yet analyzed enhancer signatures of ncHARs in the context of the Epigenomics Roadmap data. Thus, we decided to address this question as a way to validate PREDICTD in a biological application, and to extend the EnhancerFinder results by assessing cell type-specific enhancer activity in the ncHARs based on the Roadmap dataset.

Briefly, we imputed data for three enhancer-associated assays (DNase, H3K27ac, and H3K4me1) in all cell types, and averaged the imputed signal over each ncHAR to produce a small tensor with axes corresponding to three assays, 2640 ncHARs, and 127 cell types. We flattened the assay dimension of this tensor by taking the first principal component, and then used a biclustering algorithm to group the ncHARs and cell types (see Methods). The resulting cell type groups are consistent with tissue of origin (Fig. 5a, Supplementary Data 2), and the ncHARs cluster based on enhancer-associated signal in different cell type clusters as follows: no signal (77% of the ncHARs), brain/ES (13%), epithelial/mesenchymal (7%), non-immune (2%), and immune (1%) (Fig. 5a, Supplementary Data 3). Using the same strategy to cluster the available observed data gives very similar results, as quantified by the adjusted Rand index (Fig. 5b), especially when compared to two background models: shuffled, in which the ncHAR coordinates have been randomly shuffled along the genome; and other, in which the enhancer-associated marks were exchanged for three non-enhancer-associated marks (H3K4me3, H3K27me3, and H3K36me3). A heat map showing the clustering of observed data is provided in Supplementary Fig. 17.

**Table 2 Ontology search results are consistent with ncHAR cluster cell type identities**

| ncHAR cluster | Ontology | Enriched term | FDR |
|---|---|---|---|
| Non-immune | Mouse phenotype | Abnormal craniofacial development | 2.506e-03 |
| | | Abnormal embryogenesis/development | 8.269e-03 |
| | | Hemorrhage | 2.022e-02 |
| | | Abnormal embryonic tissue morphology | 2.204e-02 |
| | | Abnormal basioccipital bone morphology | 2.907e-02 |
| | | Partial neonatal lethality | 2.944e-02 |
| | | Abnormal skeleton development | 3.439e-02 |
| | | Abnormal placental labyrinth vasculature morphology | 3.465e-02 |
| | | Perinatal lethality | 3.601e-02 |
| | | Abnormal embryo size | 3.605e-02 |
| | | Abnormal craniofacial morphology | 3.703e-02 |
| | | Decreased embryo size | 3.805e-02 |
| | | Abnormal blood circulation | 3.873e-02 |
| | | Decreased skeletal muscle fiber number | 3.931e-02 |
| | | Abnormal embryonic growth/weight/body size | 4.263e-02 |
| | | Neonatal lethality | 4.344e-02 |
| Epithelial/mesenchymal | GO biological process | Embryonic organ development | 2.487e-02 |
| | | Embryo development ending in birth or egg hatching | 2.502e-02 |
| | | Somite development | 2.514e-02 |
| | | Tissue morphogenesis | 2.746e-02 |
| | | Mesenchyme development | 2.948e-02 |
| | | Stem cell differentiation | 3.109e-02 |
| | | Anterior/posterior pattern specification | 3.237e-02 |
| | | Mesenchymal cell development | 3.310e-02 |
| | | Chordate embryonic development | 3.431e-02 |
| | | Somitogenesis | 3.569e-02 |
| Brain/ES | GO biological process | Telencephalon cell migration | 2.150e-02 |
| | | Cerebral cortex cell migration | 2.897e-02 |
| | | Forebrain cell migration | 3.563e-02 |
| | | Cerebral cortex radially oriented cell migration | 4.523e-02 |

We used GREAT to find enriched ontology terms associated with genes that are possibly regulated by ncHARs from each cluster. The list of all ncHARs was used as the background, and the terms are significant at FDR < 0.05 for the hypergeometric test, and have at least a two-fold enrichment over expected

These biclustering results also agree with and expand upon previously published tissue specificity predictions from EnhancerFinder[10,18]. The brain enhancer predictions from that study are visibly enriched in our brain/ES cluster, and the limb and heart predictions are enriched in our clusters showing activity in differentiated, epithelial, and mesenchymal cell types (Fig. 5a). If we treat the EnhancerFinder tissue assignments[10] as another clustering of the ncHARs, we find that they are more similar to our clustering (both for observed and imputed data) than to either background clustering (Fig. 5b). In addition, our results expand on EnhancerFinder by assigning to cell type-associated clusters 289 ncHARs (11% of ncHARs) that were characterized by EnhancerFinder as either having activity in "other" tissues (98 ncHARs) or no developmental enhancer activity ("N/A," 191 ncHARs). We also find that our clustering successfully predicts enhancer activity for many functionally validated ncHARs, and furthermore assigns most of them to the correct cell types (Supplementary Data 4). Briefly, we correctly identify enhancer activity in ten of 23 ncHARs with evidence in the VISTA database[10,20], and 6 of 7 ncHARs with validation results suggesting enhancer activity specific to the human allele and not the chimp allele[10]; we find evidence of enhancer identity for one of three ncHARs associated with *AUTS2*, a gene associated with autism spectrum disorder, and this was one of two from that study that showed transgenic enhancer activity[17]; *NPAS3* is a gene associated with schizophrenia that lies in a large cluster of 14 ncHARs, and we find enhancer signal for seven of them, six of which have validated enhancer activity[16]; last, HAR2 is a ncHAR with validated human-specific limb enhancer activity that clusters with our brain/ES category[15]. Thus, assessing the potential enhancer activity based on the Roadmap Epigenomics data,

which encompasses different cell types and developmental stages than ENCODE, agrees with previous results and expands on them to characterize more ncHARs as having potential tissue-specific enhancer activity.

Finally, we asked what types of biological processes these putative enhancers might regulate. We extracted the genomic coordinates of the ncHARs in each cluster, and used the Genomic Regions Enrichment of Annotations Tool (GREAT)[21] to test for enriched ontology terms. Using the total list of ncHARs as the background, we found that the brain/ES cluster of ncHARs is enriched for GO biological process terms associated with cell migration in different brain regions; the epithelial/mesenchymal cluster shows enrichment for terms associated with tissue development, particularly mesenchymal cell differentiation; and, although there are no significantly enriched GO biological process terms for the non-immune cluster, there are enriched terms from a mouse phenotype ontology indicating these ncHARs could be associated with embryonic development and morphology (Table 2, Supplementary Data 5–12). We found no significantly enriched terms for the immune cluster.

The question of whether ncHARs are active enhancers in modern humans, or whether they are regions that formerly had enhancer activity that has been lost over the course of our evolution is a central question to the study of ncHAR biology. With this analysis, we shed more light on which ncHARs have enhancer activity, and even provide some insight into the relevant developmental stage for such activity, as our cell types are derived from embryonic, fetal, and adult tissues. Taken together, these results show that PREDICTD imputed data can capture cell type-specific regulatory signals, and that PREDICTD can be used as a tool to study the biology of new and under-characterized cell types in the future.

## Discussion

PREDICTD imputes thousands of epigenomics maps in parallel using a 3D tensor factorization model. Our work makes several important contributions. First, the model leverages a machine learning method, tensor decomposition, that holds particular promise in genomics for analyzing increasingly high-dimensional datasets. Tensor factorization with the PARAFAC/CANDE-COMP procedure was first proposed by two groups independently, in 1970, in the context of analyzing psychometric electroencephalogram data[8,9]. Tensor decomposition by this and related methods has since been applied in many other fields[22,23], and increasingly in biomedical fields as well[24–26]. Tensor decomposition has advantages over 2D methods, because taking into account more than 2D reduces the rotational flexibility of the model and helps drive the factors to a solution that can explain patterns in all dimensions at once. Our particular application, completing a tensor with missing data, is an area of active research[27], and is analogous to methods for matrix factorization that have proven effective in other machine learning applications like recommender systems[28]. To our knowledge, PREDICTD is just the third application of the tensor decomposition approach to epigenomics data[25,26], and the first to use a tensor completion approach to impute missing data in this setting. As such, our method demonstrates another way forward for integrating and jointly analyzing increasingly large and complex datasets in the field.

Second, PREDICTD provides some key advantages over the current state-of-art for epigenomics data imputation. The best alternative method for predicting raw epigenomics signal is ChromImpute[7]. Our tensor factorization approach is simpler and arguably more elegant than ChromImpute, because it naturally models the three key dimensions of the imputation problem, while training on and imputing all data at once. In addition, PREDICTD is less computationally intensive than ChromImpute, and scales better to imputing large numbers of experiments (see Methods—Computing resource requirements). Furthermore, as a single model that describes all experiments, the parameters PREDICTD learns during training have the same semantics across different cell types, assays, and genomic positions. We show that these parameters contain information that can be used to distinguish different types of cells, assays, or genomic elements, and future work will investigate how the PREDICTD model itself might be used to gain biological insight. Last, we show that PREDICTD outperforms ChromImpute on the MSEglobal quality measure, despite generally slightly under-performing ChromImpute on other measures (Fig. 4, Supplementary Fig. 14). There could be multiple reasons for this observation. First, as a tree-based model, ChromImpute can learn nonlinear relationships in the data that PREDICTD cannot, and it is possible that this accounts for some of the difference in performance between the two approaches. Second, the mean squared error (MSE) is central to the PREDICTD objective function, and so it is the quality measure on which the model should perform best; if another quality measure were used in the objective function, then PREDICTD might outperform ChromImpute on that one instead. Nevertheless, the fact that averaging the PREDICTD and ChromImpute results outperforms both methods alone suggests that the two approaches are complementary, and we are interested in exploring additional methods, particularly nonlinear models like deep neural networks, that might be able to combine the best of both approaches to further improve the imputed data quality.

Last, the imputed data represents an important tool for guiding epigenomics studies. Such data is far cheaper to produce than observed data, closely matches the data observed from experimental assays, and is useful in a number of contexts to generate hypotheses that can be explored in the wet lab. We showed that imputed data can provide insights into ncHARs; and Ernst and Kellis[7] previously showed that imputed data tend to have a higher signal-to-noise ratio than the observed data, that imputed data can be used to generate high-quality automated genome annotations, and that regions of the genome with high imputed signal tend to be enriched in single nucleotide polymorphisms identified in GWAS. In addition, raw imputed data includes information about signal amplitude and shape, which can provide insight into the types of regulators and binding events that are producing that signal[29–31]. In contrast, other methods that use epigenomics data for various prediction tasks[32–34] all impute binarized epigenomics signal (i.e., peak calls), and do not preserve peak shape or amplitude. Raw imputed datasets, such as those produced by PREDICTD, make no assumptions about what research questions they will be used to address, and are widely applicable to any study that analyzes ChIP-seq or DNase-seq data. Thus, in conclusion, imputed data can provide insight into cell type-specific patterns of chromatin state, and act as a powerful hypothesis generator. With just one or two epigenomics maps from a new cell type, PREDICTD can leverage the entire corpus of Roadmap Epigenomics data to generate high-quality predictions of all assays.

## Methods

**Data**. We downloaded the consolidated genome-wide signal ($-\log_{10} p$) coverage tracks in bigWig format from the Roadmap Epigenomics data portal (http://egg2.wustl.edu/roadmap/web_portal/processed_data.html#ChipSeq_DNaseSeq)[2]. These tracks are uniformly processed and currently represent the best curated collection of epigenomic maps available. In addition, these are the same tracks that Ernst and Kellis[7] used to train ChromImpute, making it easier to compare our modeling approaches.

All observed signal tracks show a higher variance at regions of high signal than at regions of low signal. In order to stabilize this variance across the genome and to make the data more tractable for PREDICTD's Gaussian error model, we applied an inverse hyperbolic sine transform. This transformation, which has been used in previous studies of epigenomic maps[35], is similar to a log transform, but is defined for zero values.

After variance stabilization, we defined five training and test splits such that each observed experiment was in one test set. First, we removed any cell types or assays with fewer than five completed experiments to ensure that there would be enough support for training in each dimension in our model. This left 127 cell types, 24 assays, and a total of 1014 completed experiments (66.6% missing). Next, we split these experiments into five test sets, by randomly generating five disjoint subsets of experiments that each contained a stratified sample from across the available cell types and assays. Thus, in each split, 20% of experiments comprise the test set and 80% comprise the training set. In addition to the held out test set, PREDICTD requires a held out validation set to detect model convergence. To ensure that all data in the training dataset contributed equally to the final imputation, the training data for each test set were further split into eight validation sets by cell type/assay pair, so that for any pair of test and validation sets the data split is 20% test (203), 10% validation (100), and 70% training (711). The imputed values reported in this paper are the average test set predictions from eight models trained on the eight validation sets corresponding to that test set. 153 experiments from the first test set were held out of our model tuning procedure as a final test set to show that the model generalizes (Supplementary Fig. 15).

Last, the data for each experiment was averaged into 25 bp bins across the genome using the bedtools map command[36], and the bins overlapping the ENCODE Pilot Regions and 1 kb windows centered at ncHARs were extracted for training the PREDICTD model. The resulting dataset contains just over 1.3 million bins, or about 1% of the genome. All experiments reported here were conducted using models trained on this subset of the genome. We find that this is more than enough data to train the model, and imputing the entire genome is a relatively simple matter of applying the learned cell type and assay factors across all positions in the genome.

**Model**. In the following sections, we present the PREDICTD model. As mentioned above, the dataset can be represented as a 3D tensor with the axes being the cell types, the assays, and the locations across the genome. We refer to these axes as the cell type, assay, and genome dimensions, respectively. We use capital letters, $J$, $K$, and $I$ to refer to the cardinality of each of these dimensions, and lowercase $j$, $k$, $i$, to refer to specific indices in each corresponding dimension. We use the same convention to refer to the number of latent factors in the model, $L$, and individual latent factor indices, $l$. Each dimension has two learned data structures associated with it: a factor matrix, and a bias vector. We use bold capital letters to refer to the

factor matrices, and bold lowercase letters to refer to the bias vectors. The cell type factor matrix and bias vector, and their dimensions are $C_{J \times L}$, and $c_{J \times 1}$, respectively. Similarly, for the assay factor matrix and bias vector, the dimensions are: $A_{K \times L}$, and $a_{K \times 1}$, and for the genome dimension: $G_{I \times L}$, and $g_{I \times 1}$.

Three main "axes" contribute to the observed biological signal in epigenomic maps: the cell type, the assay, and the genomic location that was measured. Having three axes, on which to distribute the available datasets, naturally lends the full dataset the structure of the 3D tensor (i.e., a stack of 2D matrices), in which the size of one dimension corresponds to the number of cell types in the dataset, another to the number of assays, and the third to the number of genomic locations. One might want to use other qualities or attributes to analyze the data (subject to one's research question and having enough training data), such as the lab that generated the data, the treatment that was applied, etc., but we are interested in parsing the data along the main cell type, assay, and genomic locus axes so that our model can most generally describe the biological phenomena in normal tissues.

Motivated by this 3D structure, we use tensor decomposition because this type of method factors the full data tensor into smaller components that summarize the contributions of each axis to the total data. The key to using tensor decomposition for imputing the missing data is that the smaller components that are learned by the model do not have missing values by definition. This means that when we recombine the components to reconstruct the original tensor, the resulting reconstructed tensor will not only have values that approximate the existing data in the original tensor, but also predicted values for any missing entries in the original tensor. Our particular strategy, known in the literature as PARAFAC, finds a 2D matrix (i.e. one "smaller component") for each dimension. All such matrices are of the same size along one dimension; this dimension is what we refer to as the number of "latent factors." The number of latent factors determines the complexity of the model, how well it can capture the information in each axis of the tensor, and thus how well the reconstructed tensor matches the original tensor. Each element of the reconstructed tensor is calculated by multiplying the three corresponding values (one from each matrix) for each latent factor, and then summing those products to arrive at a single number. In order to perform imputation, we train the PREDICTD tensor decomposition model, using the PARAFAC/CANDECOMP procedure[8,9], which can very naturally model the 3D problem explained above. It also has several additional advantages: it is relatively simple to implement, it has the ability to scale to a large tensor size, and it holds the possibility of producing latent factors that can provide biological insight. Briefly, in this procedure the 3D tensor is factored into three low-rank matrices, each with the same (user specified) number of column vectors. These column vectors are called "latent factors," and the tensor is reconstructed by summing the outer products of the corresponding latent factor vector triplets. These factor matrices have no missing values, so when they are combined to reconstruct the original data tensor, the reconstructed tensor contains imputed data values that not only approximate the existing tensor data, but also fill in the missing values. More precisely, we start with a 3D data tensor $D$ with dimensions $J \times K \times I$, where $J = 127$ is the number of cell types, $K = 24$ is the number of assays, and $I = 1,309,125$ is the number of genomic locations (in our case the ENCODE Pilot Regions and 2640 ncHARs at 25 bp resolution), represented by the tensor. This tensor has missing data in fibers along the genome dimension, corresponding to experiments on cell type/assay pairs that have yet to be completed. The completed experiments, corresponding to tensor fibers that contain data, are split into training, validation, and test subsets, or $S^{train}$, $S^{valid}$, and $S^{test}$, respectively.

We factor the tensor $D$ into three factor matrices and three bias vectors $a$, $c$, and $g$. These bias vectors are meant to capture global biases for each cell type, assay, or genomic location, respectively. Essentially, these terms subtract out the mean for each cell type, assay, and genomic location, which helps to mathematically center all of the data in the tensor around the same point, so that the patterns that we want the model to learn are not obscured by trivial differences in scale along the axes. It is a common strategy for models like PARAFAC, that perform best on data that is all on the same scale.

We train the model to find the values of these terms that minimize the following objective function:

$$\text{argmin}_{C,A,G,c,a,g} \sum_{j,k,i \in S^{train}} \left( D_{j,k,i}^{train} - \left[ \sum_{l=1}^{L} C_{j,l} * A_{k,l} * G_{i,l} + c_j + a_k + g_i \right] \right)^2 \\ + \lambda_C \parallel C \parallel_2^2 + \lambda_A \parallel A \parallel_2^2 + \lambda_G \parallel G \parallel_2^2 \quad (1)$$

The objective function (Eq. (1)) has two main parts. The first part calculates the squared error between the training data, $D_{j,k,i}^{train}$ and the model's prediction, $\sum_{l=1}^{L} C_{j,l} * A_{k,l} * G_{i,l} + c_j + a_k + g_i$. This term penalizes the distance between the imputed and observed data. The last three terms, $\lambda_C \parallel C \parallel_2^2 + \lambda_A \parallel A \parallel_2^2 + \lambda_G \parallel G \parallel_2^2$ implement L2 regularization on the factor matrices. This type of regularization penalizes large parameter values, and thus causes the model to strongly prefer a solution with small values on the parameters. Such regularization helps to reduce the flexibility of the model and helps to avoid overfitting the training data. Furthermore, we note that our choice of PARAFAC, which is a linear model with a limited number of latent dimensions, is itself a form of regularization in the sense that such a model is less flexible than more complex models, like deep

neural nets. PARAFAC is therefore inherently less prone to overfitting the training data, compared to a nonlinear model given the same model dimensionality.

Equation (1) cannot be solved analytically, so we solve it numerically using stochastic gradient descent (SGD). In SGD, we first initialize the three factor matrices with random values from a uniform distribution on the domain (−0.33 to 0.33), and the three bias vectors with the mean value from each corresponding plane in the tensor. Then, we randomly iterate over the training set data points in the tensor at each iteration, calculating the gradient of the objective function (Eq. (1)) with respect to each factor matrix and bias vector, and then adding a fraction of this gradient to the corresponding parameter values. Over time, as more and more gradients are calculated and used to update the parameter values in the factor matrices and bias vectors, the model as a whole "moves" along the high-dimensional surface defined by the objective function and "down" toward a minimum that (ideally) represents a good solution. We track the model's progress toward this solution by periodically saving the value of the MSE on the heldout validation data points. Eventually, the validation MSE stops decreasing, which indicates that the model parameters have converged on a solution. Importantly, there is no guarantee that this solution is the best possible one, as in the case of PREDICTD (and PARAFAC more generally) the objective function is not convex.

We should also note that PREDICTD incorporates several other modifications to this SGD procedure to improve the speed, reliability, and accuracy of training. First, in order to take full advantage of our compute cluster, we use parallel SGD[37], which is discussed in detail in the Implementation section below. And second, to improve model convergence under SGD training, PREDICTD implements the Adam optimizer[38] with Nesterov Accelerated Gradient[39] (Fig. 1). Finally, we note that because there is no non-negativity constraint on the model training, a small fraction of imputed values are negative (Supplementary Fig. 12). Negative values are invalid for $-\log_{10} p$-value tracks, so we set any such imputed values to zero in the final output.

There are many tensor decomposition methods (reviewed in ref. [23]); however, we chose the PARAFAC model because of its relative simplicity. It is not only straightforward to implement and parallelize, but it also requires fewer parameters than other tensor factorization methods[22,23]. Note that we implemented the model as described in the original publication[8], and we included no additional constraints on the model during training except what was imposed by the L2 regularization terms in the objective and the constraints naturally imposed by using a relatively simple linear model on complex data with potentially nonlinear underlying factors. The PARAFAC model also has the nice property that as long as mild conditions hold it will find a solution that is unique with respect to rotation transformations[22,23]; this is not a property of other tensor factorization approaches, including Tucker decomposition, which was used in ref. [25].

**Implementation**. PREDICTD is implemented in Python 2.7 and built using the Apache Spark 1.6.2 distributed computation framework (http://spark.apache.org). The code is open-source and available on BitBucket (https://bitbucket.org/noblelab/predictd), and the environment we used to train the model is available on Amazon Web Services (AWS) as an Amazon Machine Image (see the BitBucket repository for info). Models were trained using AWS Elastic Compute Cloud (EC2) (http://aws.amazon.com) and Microsoft Azure Spark on HDInsight (http://azure.microsoft.com). We bootstrapped an EC2 cluster running Apache Spark 1.6.2 by running the spark-ec2 script (https://github.com/amplab/spark-ec2) on a small EC2 instance (e.g., m3.medium) that we subsequently terminated after the cluster was up and running. Standard cluster configuration was a single m4.xlarge head node instance and one r3.8xlarge worker instance, giving a total cluster size of two nodes, 36 cores, and 260 GB of memory. Whenever possible, we used SPOT instances to make the computation more affordable. Microsoft Azure HDInsight clusters had similar resources. All data input to the model and all model output was written to cloud storage; either Simple Storage Service (S3) on AWS, or Blob Storage on Azure.

The data tensor is assembled into a Spark Resilient Distributed Dataset Structure (RDD) and partitioned among the cluster nodes, such that each partition is stored on a single node and contains the data for 1000 genomic loci. This results in about 1300 partitions. The data in each of the 1000 elements in each partition is represented as a scipy.sparse.csr_matrix[40] object storing all observed data values for a particular genomic position. Each element of the data RDD also contains the corresponding entries from the G factor matrix, g bias vector, and data structures for the Adam optimizer[38] that are specific to each genomic locus (Fig. 1).

The first step of training selects a random 1% of available genomic positions (~13,000 positions, or ~0.01% of all 25 bp bins in the genome) for training the cell type and assay parameters. Although this seems like a small sample of the data, our results indicate that this is enough data to faithfully represent the distribution of signal across the tensor. We do see a slight improvement in performance if we include more of the genome in training, but at a cost of correspondingly increased memory usage and compute time. The main training phase then proceeds through a series of parallel SGD[37] iterations (Fig. 1c) on this subset of positions. Briefly, at the start of each parallel iteration, copies of the cell type and assay parameters, C, c, A, and a, are sent out to each partition. Each partition undergoes local SGD for 5000 iterations and applies the updates to the local copies of the assay and cell type parameters. The updated cell type and assay parameter values are then passed back to the master node where they are averaged element-wise with the results from all other partitions. The resulting averaged parameters are then copied and distributed

to the partitions for the next round of parallel SGD. Note that over all rounds of SGD, we use a learning rate decay schedule of $\eta_t = \eta \times (\varphi_\eta)^{t-1}$, where the learning rate decay parameter $\varphi_\eta = 1 - 1e^{-6}$, and similarly for the Adam first moment parameter: $\beta 1_t = \beta 1 \times (\varphi_{\beta 1})^{t-1}$, where $\varphi_{\beta 1} = 1 - 1e^{-6}$.

Averaging the parameters after the parallel SGD updates allows the model to share information across the genome dimension; however, the averaging can initially make it harder for the model to converge. The $\mathbf{C}$ and $\mathbf{A}$ matrices are initialized randomly from a uniform distribution on the domain $(-0.33, 0.33)$, and thus during the first round of parallel SGD, the independent nature of the local updates can lead to inconsistent updates to the latent factors in different partitions. When the results of these inconsistent updates are averaged, they produce poor parameter values, and it then takes many parallel iterations before the parameter values begin to converge. To combat this effect, the main training phase begins with a burn-in stage before attempting parallel SGD. In the burn-in stage, local SGD is performed for half an epoch on 8000 genomic loci in a single partition, and after this, the updated $\mathbf{C}$, $\mathbf{c}$, $\mathbf{A}$, and $\mathbf{a}$ parameters are used in a round of local SGD across the entire training subset to bring the genome dimension up to the same number of updates. This burn-in procedure allows the latent factors to have a consistent initial "identity" across the cluster when starting the parallel SGD updates.

Every three parallel SGD iterations, the MSE is computed for each subset of data (training, validation, and test) and recorded. If the validation MSE is the lowest yet encountered by the model, the parameters from that iteration are copied and saved. Once a minimum number of parallel iterations have completed, the model tests for convergence by collecting the MSE on the validation set for iterations $t-35$ to $t-20$ (window 1), and $t-15$ to $t$ (window 2), and using a Wilcoxon rank-sum test to determine if window $2 + 1e^{-5} >$ window 1, with one-tailed $p < 0.05$. If this convergence criterion is met, then one of two things happens. First, the model will check whether or not the user has requested a line search on the learning rate. If so, then it will reset the cell type, assay, and genome parameters to those found at the iteration with the minimum validation MSE and resume parallel SGD after halving the learning rate, and reducing the Adam first moment weight $\beta 1_{new} = \beta 1_{old} - (1.0 - \beta 1_{old})$. When training the model, we used a line search of length three, so the model was restarted from the current minimum and learning rate halved and $\beta 1$ adjusted three times. See Supplementary Fig. 18 for an example of what the error curves from the parallel SGD look like after training a PREDICTD model. Once the line search is complete, or if no line search was requested, then the model stops parallel SGD, fixes the assay and cell type parameters, and finishes training on the genome parameters only.

Once the main phase of training is complete, the last phase of model training applies the cell type and assay parameters across all genomic positions. This is accomplished by fixing the cell type and assay parameters and calculating the second-order solution on the genome parameters only. This requires just a single parameter update per genomic position, which is possible using least squares because fixing the cell type and assay parameters makes our objective function convex over the genome parameters. Once the final genome parameters are calculated, the assay, cell type, and genome parameters are saved to cloud storage, and the imputed tensor is computed and saved to the cloud for further analysis. On average, the entire training takes about 750 parallel iterations, and about 2 h (wall-clock time).

The above procedure is executed for every validation set associated with a given test set, and then the final imputed values for the held out test datasets are calculated as a simple average of the corresponding imputed values from each validation set. Thus, for the results we report here, each imputed value represents the consensus of eight trained models.

**Hyperparameter selection.** One of the challenges of working with this type of model is that there are many hyperparameters to tune. These include the number of latent factors $L$, the learning rate $\eta$, the learning rate decay rate $\varphi_\eta$, the Adam first moment coefficient $\beta_1$ and its decay rate $\varphi_{\beta 1}$, a regularization coefficient for each latent parameter matrix ($\lambda_A$, $\lambda_C$, $\lambda_G$), and one more regularization parameter for the second-order genome updates ($\lambda_{G2}$).

Of these hyperparameters, perhaps the most important one for PREDICTD performance is the number of latent factors. This setting controls the dimensionality of the model, and thus the number of parameters that must be optimized during model training. Ideally, assuming a perfect match between the modeling approach and the data, the number of latent factors will equal the true underlying dimensionality of the dataset. However, in practice this assumption does not really hold. First, real world data is often noisy enough that the "true" dimensionality of the input data is the full rank, and so instead we are forced to use fewer latent factors that approximate the dimensionality of theoretical, noiseless, data. Second, PREDICTD implements the original PARAFAC specification[8], which relies on simple linear combinations of the corresponding latent factors in each dimension. However, in real data there could be factors that have nonlinear relationships, and there is evidence that PARAFAC in some cases will attempt to fit these relationships by adding additional factors to explicitly take them into account as if they are additional linear terms. This phenomenon was explored in an example from the original PARAFAC paper in which the best PARAFAC solution for a rank-2 synthetic dataset with an interaction between the two latent dimensions used three latent factors: one for each dimension, plus another for the product of the two[8]. In the end, the best number of latent factors to use is simply

the number that minimizes the error of the model while preserving its generalization performance, and this must be evaluated empirically.

Empirically searching for the best number of latent factors is nontrivial. The number of latent factors changes the dimensionality of the model, and thus the balance between bias and variance, which means that the regularization coefficients must be tuned in parallel with the latent factors. A simple strategy that has been shown to be surprisingly effective searching high-dimensional hyperparameter space is simple random search, in which different random hyperparameter values are tested until a combination is found that provides good performance of the model[41]. Although the simplicity of the random choice strategy makes it very appealing, it can still require many iterations before one is confident that good hyperparameters have been found, which is a severe drawback when trying to optimize settings for a model like PREDICTD that takes multiple hours to train. Thus, hoping to find good hyperparameter settings in as few iterations as possible, we decided to use an auto-tuning software package called Spearmint[42]. Spearmint treats the PREDICTD model as a black box function and iteratively tries different hyperparameter settings; it uses Bayesian optimization to fit a Gaussian process that can predict the hyperparameter settings that will maximize the improvement in model performance in the next iteration. There is still some debate in the field as to whether or not this kind of auto-tuning strategy reliably finds better hyperparameter values than simple random search[43]; however, evidence shows that such Bayesian approaches tend to converge to a good selection of hyperparameters in fewer iterations than random search[42], and thus minimize the time spent searching hyperparameters.

We ran Spearmint multiple times as we developed the PREDICTD model, each time holding out the first test set so that we would have new data to test the generalizability of PREDICTD. Early Spearmint runs and some manual grid search of the hyperparameters suggested that 100 latent factors was a good setting for the model dimensionality. Once we settled on 100 latent factors, we ran Spearmint again to fine tune the learning rate and regularization coefficients. We let it train 188 PREDICTD models with different hyperparameter settings and selected the settings from the model that gave the lowest observed validation MSE. During this process, we discovered that PREDICTD is relatively insensitive to the particular values of the three regularization coefficients $\lambda_C$, $\lambda_A$, and $\lambda_G$, but that it seemed to prefer extremely low values (essentially, no regularization) on at least one of the matrix factors. In contrast, the hyperparameter search revealed that PREDICTD performance depends more heavily on particular values for the learning rate, $\eta$, and the second-order genome update regularization, $\lambda_{G2}$. We also found that our imputation scheme of averaging eight models trained with different validation sets imposed extra regularization on the ultimate averaged solution, and that to achieve the best generalizability of our averaged solution we had to compensate for the regularization introduced by the averaging by choosing a lower $\lambda_{G2}$ than the one suggested by Spearmint as the best setting for a single model. After trying different $\lambda_{G2}$ values (Supplementary Fig. 21), we decided to reduce $\lambda_{G2}$ by a factor of 10 since this showed that the validation MSE stayed roughly constant or a little bit lower than the minimum validation MSE from the parallel SGD iterations, and thus we were not lowering the regularization so much that the model overfit and increased the validation MSE. Our final chosen hyperparameter values are given in Table 3.

In addition to using Spearmint for model selection, we also used it to systematically explore the effects of changing the model dimensionality by changing the number of latent factors (Supplementary Figs. 19, 20). In this hyperparameter search, we fixed the dimensionality at one of 17 levels between two and 512 latent factors, and then used Spearmint to optimize the other hyperparameters ($\eta$, $\lambda_C$, $\lambda_A$, $\lambda_G$, and $\lambda_{G2}$). We allowed the Spearmint runs with larger numbers of latent factors to train longer to give them more chances to explore the more complex solution space of these higher dimensionality models. We used a systematic stopping criterion as follows: each Spearmint search had to train for at least 50 iterations or 40% of the number of latent factors, whichever was more, and had to stop after it had trained at least 20 iterations or 15% of the number of latent factors, whichever was more, past its best result (Supplementary

### Table 3 Hyperparameter values

| Hyperparameter | Value | Spearmint? |
|---|---|---|
| $\eta$ | 0.0045 | Y |
| $\varphi_\eta$ | $1-1e^{-6}$ | N |
| $\beta_1$ | 0.9 | N |
| $\varphi_{\beta 1}$ | $1-1e^{-6}$ | N |
| $\beta_2$ | 0.999 | N |
| $L$ | 100 | Y* |
| $\lambda_C$ | 4.792 | Y |
| $\lambda_A$ | $8.757e^{-27}$ | Y |
| $\lambda_G$ | $8.757e^{-27}$ | Y |
| $\lambda_{G2}$ | 0.4122 | Y* |

The third column indicates whether the hyperparameter value was selected using Spearmint, and an asterisk (*) indicates the final value was tuned by hand after Spearmint optimization.

Fig. 19 blue/red bars). After this search, we noticed that there was a plateau in the validation MSE from 16 latent factors to 64 latent factors, so to gain more resolution on this range of latent factors, we trained the 32 and 64 Spearmint searches out to 120 iterations. We found that the solutions for both models improved, and 64 latent factors improved more than 32 latent factors, but that neither model found a better solution than 100 latent factors (Supplementary Fig. 19 brown/orange bars). In order to avoid biasing Spearmint's choice of hyperparameter settings for a particular validation set or subset of genomic locations, we had allowed the validation set and the training subset of genomic windows to vary randomly over the course of the hyperparameter search. However, this meant that any given best Spearmint result could still be due to the model getting "lucky" and finding a validation set or set of genomic windows that was particularly favorable for training. To convince ourselves that the trend our Spearmint search revealed is real, we took the best hyperparameter settings for each latent factor level (for 32 and 64 latent factors these were the results of the expanded search) and trained ten models each with fixed validation sets and a fixed set of genomic windows, only varying the random initialization of the factor matrices from model to model (Supplementary Fig. 20). The results show the same trend as a function of model dimensionality as in our original hyperparameter search (Supplementary Fig. 19), and we also verified that the distribution of validation MSE for 64 latent factors is significantly different than that for 100 latent factors (Wilcoxon rank-sum test $p < 0.05$).

To save time on model training during the Spearmint iterations, we relaxed the convergence criteria to use a larger shift between the two samples in the Wilcoxon rank-sum test ($5e^{-05}$ instead of $1e^{-05}$) and we only did a single line search after the model first converged instead of three. It is important to note that, despite our efforts, there may be even better hyperparameter settings that our search did not encounter. As new discoveries concerning hyperparameter tuning unfold in the machine learning literature the settings for PREDICTD can be revisited to perhaps further increase its performance.

**Imputing the whole genome**. Although for the purpose of analyzing the PRE-DICTD model we only imputed about 1% of the genome, we generated whole genome imputed tracks in bigWig format for the UCSC Genome Browser. These tracks are available for download from the ENCODE project website (http://encodeproject.org).

**Imputing data for a novel cell type**. We provide a tutorial on the BitBucket site (https://bitbucket.org/noblelab/predictd/wiki/Home) that details how a user can train a PREDICTD model to generate imputed data for a new cell type. Briefly, a user can upload $-log_{10}$ p-value tracks in bigWig format to an Amazon S3 bucket, and then PREDICTD will add that data to the Roadmap Epigenomics tensor, train the model, and write imputed data for the new cell type back to S3 in bigWig format. The tutorial demonstrates how this is done with seven datasets from the Fetal Spinal Cord cell type that we downloaded from the ENCODE portal (http://www.encodeproject.org).

**Computing resource requirements**. The resource requirements of PREDICTD are not very great considering the size of the model. We find that training a single PREDICTD model on the tensor described in the paper ($127 \times 24 \times 1.3e6$) takes on average just under two hours on a two node cluster consisting of a head node with four cores (Intel Xeon E5-2676 v3 Haswell or Xeon E5-2686 v4 Broadwell processors) and 16 GB of memory (e.g., an m4.xlarge AWS EC2 instance), and a worker node with 32 cores (Intel Xeon E5-2670 v2 Ivy Bridge) and 244 GB of memory (e.g., an r3.8xlarge AWS EC2 instance). For this manuscript, each experiment was imputed as an average of eight models trained with random starts and different validation sets, so one could train these models to use for imputation in about 16 h. After training the models, imputing values for the limited subset of genomic positions used for training is quite fast. However, if one needs to impute the whole genome it takes longer because the learned cell type and assay factors must be applied across all genomic locations. To do this without having to store the entire tensor in memory at once (all genomic positions and no missing values), we read in data for batches of genomic positions, train the corresponding genome parameters based on the existing cell type and assay parameters, and then write out the imputed values for each batch. For imputing whole genome data for one new cell type (that is, 24 whole genome imputed experiments) the cluster configuration described above requires an additional 24 h, for a total of ~40 h for model training and whole genome imputation.

In this manuscript we present a more extreme case in which we impute all 3048 possible experiments in the Roadmap Epigenomics tensor at 25 bp resolution, and to do this we used a larger worker node to increase throughput. If we use a x1.16xlarge instance as the worker node, which has 64 cores (Intel Xeon E7-8880 v3 Haswell) and 976 GB of memory, we can use the trained models to impute the whole genome for all 3048 experiments in approximately 88 h. The resulting imputed tracks represent the consensus of eight models for each experiment, and these experiments were split into five test sets, giving a total of 40 models that took about 76.5 h to train. Thus, training and imputation for the 3048 Roadmap Epigenomics tracks takes a total time of ~164.5 h.

To compare with ChromImpute's run time, we can convert this wall-clock time to an approximate number of CPU hours required to run PREDICTD on the full

tensor. Using the smaller cluster to train the 48 models, we calculate PREDICTD requires about 36 cores × 76.5 h = 2754 CPU hours. Switching to the larger cluster for imputation, we find that PREDICTD consumes about an additional 68 cores × 88 h = 5984 CPU hours. Thus, in total PREDICTD can train the models and impute 3048 experiments in ~8738 CPU hours. This run time is more than an order of magnitude less than that quoted in the ChromImpute supplement[7], which reports that ChromImpute requires a total run time of 103,560 CPU hours for model training and output generation. Even taking into account the fact that we imputed about 25% fewer experiments for this paper than were imputed in the ChromImpute manuscript, ChromImpute still requires on the order of ten times more CPU hours to train the models and impute the Roadmap Epigenomics tensor than PREDICTD does.

**Advantages of the consumer cloud**. Cloud computing is becoming a powerful tool for bioinformatics. Large consortia such as the ENCODE[1] and The Cancer Genome Atlas (http://cancergenome.nih.gov) are making their data available on cloud platforms. As computational analyses grow more complex and require more computing resources to handle larger datasets, the cloud offers two distinct advantages. First, cloud services provide a centralized way to host large datasets used by the community that makes data storage, versioning, and access more simple and efficient. Transferring gigabytes, or even terabytes, of data is slow and expensive in terms of network bandwidth, but moving code and computation to the data is fast and cheap. Second, in addition to hosting datasets, cloud services can host several computing environments. Such virtual machine images can help with reproducibility of results for complex analyses because the code can be written in such a way that other users can not only use the same code and data as the original authors, but they can run the analysis in the same computing environment. One downside of cloud computing for labs that have access to a local cluster is that cloud resources are charged by usage; nevertheless, generating high-quality imputed data using PREDICTD is extremely cost effective compared to collecting the observed data. Training the models and generating the final imputed data for this paper costs on the order of US $0.10 per dataset, which is orders of magnitude lower than the cost of completing these experiments in the wet lab, and this cost can be expected to drop as computational resources become cheaper and more efficient optimization methods are devised.

**Imputation quality measures**. We generated tracks for the imputed data by extracting the data for each 25 bp bin from the imputed results, writing the results to file in bedGraph format, then converting to bigWig using the bed-GraphToBigWig utility from UCSC. Imputed tracks were visually inspected alongside Roadmap Consolidated data tracks and peak calls in the UCSC Genome Browser. We did not reverse the variance stabilizing inverse hyperbolic sine transform when evaluating model performance. This is appropriate because it maintains the Gaussian error model that underlies the PREDICTD optimization.

We also implemented ten different quality assessment measures (listed below), the last seven of which were first reported for ChromImpute[7]. We report these measures for heldout test set experiments and compute them over the ENCODE Pilot Regions (Supplementary Fig. 14).

- MSEglobal: Mean squared error between the imputed and observed values at all available genomic positions.
- MSE1obs: Mean squared error between the imputed and observed values in the top 1% of genomic positions ranked by the observed signal values.
- MSE1imp: Mean squared error between the imputed and observed values in the top 1% of genomic positions ranked by the imputed signal values.

  MSE1imppred: Meansquared error between the imputed and observed values in the top 1% of genomic positions ranked by the signal values imputed by PREDICTD.
  MSE1impchrimp: Mean squared error between the imputed and observed values in the top 1% of genomic positions ranked by the signal values imputed by ChromImpute.
  MSE1impme: Mean squared error between the imputed and observed values in the top 1% of genomic positions ranked by the signal values imputed by Main Effects.

- GWcorr: Pearson correlation between imputed and observed values at all available genomic positions.
- Match1: Percentage of the top 1% of genomic positions ranked by observed signal that are also found in the top 1% of genomic positions ranked by imputed signal.
- Catch1obs: Percentage of the top 1% of genomic positions ranked by observed signal that are also found in the top 5% of genomic positions ranked by imputed signal.
- Catch1imp: Percentage of top 1% of genomic positions ranked by imputed signal that are also found in the top 5% of genomic positions ranked by observed signal.
- AucObs1: Recovery of the top 1% of genomic positions ranked by observed signal from all genomic positions ranked by imputed signal calculated as the area under the curve of the receiver operating characteristic.

- AucImp1: Recovery of the top 1% of genomic positions ranked by imputed signal from all genomic positions ranked by observed signal calculated as the area under the curve of the receiver operating characteristic.
- CatchPeakObs: Recovery of genomic positions at called peaks in observed signal from all genomic positions ranked by imputed signal calculated as the area under the curve of the receiver operating characteristic.

**Analyzing model parameters**. The parameter values corresponding to individual latent factors are not individually interpretable, but intuitively we can understand that each latent factor describes some pattern in the data that the model finds useful for imputation. For example, the first latent factor (i.e., column 0 in each of the three factor matrices) might contain values that capture a pattern of high signal in promoter marks, in blood cell types, at active genes. In such a case the value at this latent factor for a particular assay might suggest how often that mark is found at promoters; for a particular cell type its relatedness to blood; and for a genomic locus how many promoter-associated features occur there in blood cell types. If these three conditions hold, then the model is likely to have more extreme values for these parameters that end-up imputing a high value for that cell type/assay pair at that genomic position.

**Clustering cell types and assays**. The rows of the cell type and assay factor matrices, with each row containing the model parameters for a particular cell type or assay, respectively, were clustered using hierarchical clustering. This analysis was implemented in Python 2.7 using scipy.spatial.distance.pdist with metric = 'cosine' to generate the distance matrix and scipy.cluster.hierarchy.linkage with method = 'average' to generate clusters. The columns of each factor matrix (i.e., the latent factor vectors) were also clustered in the same way to help with visualizing the clusters. The parameter values were plotted as a heat map with rows and columns ordered according to the results of the hierarchical clustering.

**Summarizing latent factor patterns at genomic elements**. The genome factor matrix is too large to usefully visualize as a heat map, so we sought to aggregate the parameter values across different types of genomic features. We mapped all annotated protein-coding genes from the GENCODE v19 human genome anno-tation[11] (https://www.gencodegenes.org/releases/19.html) with a designated primary transcript isoform (called by the APPRIS pipeline) to a canonical gene model consisting of nine components: promoter, 5′ UTR, first exon, first intron, middle exon, middle intron, last exon, last intron, and 3′ UTR. The promoter for each gene was defined as the 2 kb region flanking the 5′ end of the gene annotation, while the other components were either taken directly from the GENCODE annotation (5′ UTR, 3′ UTR, exons) or were inferred (introns). For each gene, each component was split into ten evenly spaced bins and the values for each latent factor were averaged so that there was a single value for each latent factor for each bin. Coding regions for genes with a single exon or two exons were mapped only to first exon, or first exon and last exon components, respectively. Genes with only one or two introns were handled analogously. For genes with multiple middle exons and introns, each exon/intron was binned independently and the data for each middle exon/intron bin was averaged across all middle exons/introns. In order to plot the results, outlier values in the bins (defined as any values outside $1.5 \times$ IQR) were removed and the remaining values averaged across corresponding bins for all binned gene models. This resulted in a matrix containing latent factors on the rows and gene model bins on the columns. The latent factors (rows) were clustered using hierarchical clustering, with scipy.spatial.distance.pdist(metric = 'euclidean') to generate the distance matrix and scipy.cluster.hierarchy.linkage(method = 'ward') to generate clusters, and this matrix was plotted as a heat map.

To compile a reference list of genome coordinates containing distal regulatory elements that is orthogonal to our imputed data, we downloaded P300 peak data from six ENCODE cell lines (A549, GM12878, H1, HeLa, HepG2, and K562), filtered for peaks with FDR < 0.01, merged the peak files with bedtools merge to create a single reference list, and averaged genome latent factor values as in the gene model explained above for ten 200 bp bins covering 2 kb windows centered on these peaks.

To validate that the patterns in the genome parameters were not due to chance, we generated the same heat map, but before averaging the bins for each gene model and P300 site we randomly permuted the order of the genome latent factors (Supplementary Fig. 11).

**Comparing to ChromImpute**. To compare the performance of PREDICTD with ChromImpute, we downloaded the ChromImpute results from the Roadmap Epigenomics website and put them through the same pipeline as for the observed data: Convert to bedgraph, use bedtools map to calculate the mean signal over 25 bp bins, extract the bins overlapping the ENCODE Pilot Regions, apply the inverse hyperbolic sine transform, and store the tracks in a Spark RDD containing a list of scipy.sparse.csr_matrix objects.

We calculated all of the quality measures on these ChromImpute datasets and plotted these results against those for PREDICTD for each experiment as a ternary scatter plot (Fig. 4b, Supplementary Fig. 14). We also averaged each element of this ChromImpute RDD with its corresponding element in the PREDICTD results, calculated the quality measures, and compared them in the same way (Fig. 4c). In order to compare both ChromImpute and PREDICTD to the baseline Main Effects

model, we used ternary plots[44] to project the three-dimensional (3D) comparison of each experiment to 2D. Each point on these ternary plots represent the relative magnitude of each dimension for that point. So, each coordinate $(x, y, z)$ in Cartesian space is projected to a point $(x', y', z')$ such that $x' = \frac{x}{x+y+z}$, $y' = \frac{y}{x+y+z}$, and $z' = \frac{z}{x+y+z}$. Thus, for the case where $x = y = z$, the corresponding point $(x', y', z') = (0.33, 0.33, 0.33)$ and will fall at the center of the ternary plot, while points that lie along the Cartesian axes will fall at the extreme points of the ternary plot (e.g., $(x, y, z) = (1, 0, 0) = (x', y', z')$).

It is important to emphasize that the quality measure with the best PREDICTD performance, MSEglobal, is also explicitly optimized by the PREDICTD objective function during training. This shows that PREDICTD is doing well on its assigned learning task, and highlights the importance of designing an objective function that reflects the task that the model will address. As such, it should be possible to tune the objective function to perform better on other quality measures if need be. For example, in an attempt to boost PREDICTD's performance on regions with higher signal we experimented with weighting genomic positions by ranking them by the sum of their signal level ranks in each training dataset. This provided some improvement on the MSE at the top 1% of observed signal windows measure (MSE1obs), but we ultimately decided to pursue the simpler and more balanced objective function presented here.

**Assessing cell type-specific enhancer signatures at ncHARs**. We downloaded the ncHAR coordinates used in Capra et al.[10], removed any that overlapped a protein-coding exon according to the GENCODE v19 annotations[11], and extracted all available observed and imputed data for the enhancer-associated assays H3K4me1, H3K27ac, and DNase at these regions. Some cell types were lacking observed data for H3K27ac (29) and/or DNase (74), but observed data for H3K4me1 was available in all cell types. We took the mean signal for observed experiment at each ncHAR coordinate and used that as input to the subsequent analysis.

First, we extracted the first principal component of the three assays for all ncHARs and cell types using sklearn.decomposition.TruncatedSVD[45] to reduce the assay dimension length from three to one and construct a matrix of ncHARs by cell types. This also had the effect of filling in missing values for the observed data. Next we wanted to cluster the ncHARs and cell types, and so we first used the matrix based on imputed data to assess how many clusters would be appropriate for the data. Briefly, for both the ncHAR and cell type dimensions, we conducted an elbow analysis by calculating the Bayesian information criterion for $k$-means clustering results for all values $2 <= k <= 40$, as well as a silhouette analysis on the same range of values for $k$ (Supplementary Fig. 16). Based on the results, we decided that $k = 5$ for the ncHARs and $k = 6$ for cell types would give us a good balance of distance between clusters and number of clusters.

Next, we clustered the imputed and observed matrices with the scikit-learn sklearn.cluster.bicluster.SpectralBiclustering class[45] to generate a biclustering using six column clusters and five row clusters. And finally, we plotted the clustering results for the imputed data as a heatmap in which each cell is the inverse hyperbolic sine-transformed sum of the mean H3K4me1, H3K27ac, and DNase signals at a particular ncHAR in a particular cell type. We also plotted the tissue assignments for ncHARs with predicted developmental enhancer activity based on EnhancerFinder[18] calls in the Capra et al.[10] paper alongside our ncHAR clusters (Fig. 5a). The same plot for the observed data is shown in Supplementary Fig. 17.

In order to gain further insight into the genes associated with our ncHAR clusters, we extracted the genomic coordinates of the ncHARS in each cluster and input these regions to the GREAT[21] to find enriched ontology terms associated with nearby genes. We used GREAT version 3.0.0 on the human hg19 assembly with the default association rule parameters (Basal + extension: 5000 bp upstream, 1000 bp downstream, 1,000,000 bp max extension, curated regulatory domains included). We first analyzed each cluster for term enrichment against a whole genome background (Supplementary Data 5, 6, 8, 10, 12), and then ran the test with the same parameters using the list of all ncHARs as the background (Table 2, Supplementary Data 7, 9, 11). No terms were significantly enriched for cluster 0 (No Signal) or cluster 4 (Immune) when using the all ncHAR background, and so we omit these results from the supplement. When reporting the results in the main text we used the default GREAT filters for significant terms: FDR < 0.05 for the hypergeometric test with at least a twofold enrichment over expected.

Last, in order to compare the clustering results on the imputed data to the observed data, we used the adjusted Rand index, which assesses how often pairs of data points are put in the same or different clusters, on the ncHAR and cell type clusters independently. As negative controls, we also conducted the same clustering analysis on the observed data after shuffling the ncHARs to other noncoding coordinates (Shuffled), and after switching out the enhancer-associated marks for H3K4me3, H3K36me3, and H3K27me3, which are not associated with enhancers (Other). We compared the resulting clusters with the enhancer-associated imputed data and observed data clusters, again using the adjusted Rand index. Last, we used the adjusted Rand index once more to assess the similarity of our biclustering results to the grouping of ncHARs based on predicted tissue-specific developmental enhancer activity from Capra et al.[10] (Fig. 5b).

**Code availability**. The PREDICTD code base is open source and made available through the MIT License. All code and documentation required to run

PREDICTD, including tutorials and command line usage, are available through the PREDICTD repository hosted on BitBucket: https://bitbucket.org/noblelab/predictd.

**Data availability**.

- The Roadmap Epigenomics Consolidated Data are available through the project data portal, http://egg2.wustl.edu/roadmap/web_portal/processed_data.html#ChipSeq_DNaseSeq.
- ChromImpute datasets are also available through the Roadmap Epigenomics project data portal, http://egg2.wustl.edu/roadmap/web_portal/imputed.html#imp_sig.
- All imputed data generated for this paper are available through the ENCODE project portal, https://www.encodeproject.org/, and the list of accession IDs is provided in the Supplementary Data 17.xlsx file associated with this manuscript.
- The Amazon Machine Image for running the PREDICTD software, along with the associated reference data files, are hosted on AWS. The download locations are provided in the documentation with the PREDICTD code (see the code availability statement above).
- Data for the quality measures reported for PREDICTD in Figs. 2b and 4, and Supplementary Figs. 2, 14, and 15 are provided in Supplementary Data, as are the results from the GREAT analysis of ncHAR clusters (see Supplementary Information).

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

## Acknowledgements

We gratefully acknowledge support from the Amazon Web Services Cloud Credits for Research program and Microsoft Azure for Research program for providing computing cycles to help with the development of PREDICTD. We would also like to thank Dr. Rob Fatland and the UW High Performance Computing Club for assistance with the cloud computing aspects of our project and for granting us additional Amazon Web Services credits. This work was funded by National Institutes of Health awards R01 ES024917 and U41 HG007000.

## Author contributions

W.S.N., J.J.H., and J.B. conceived the tensor decomposition approach for data imputation. T.J.D. implemented PREDICTD, executed all analyses, including designing and executing the model evaluation and noncoding human accelerated region analyses, and wrote the manuscript. W.S.N., J.B., M.W.L., and J.J.H. provided essential input on the mathematical and machine learning components of the project, along with additional critical feedback throughout the publication and review process. All authors read and approved the final manuscript.

## Additional information

**Competing interests:** The authors declare no competing interests.

