## [Peer Review File(PDF 9010 kb) · Nature Communications]

Reviewers' comments:

Reviewer #1 (Remarks to the Author):

The manuscript "PREDICTD: PaRallel Epigenomics Data Imputation with Cloud-based Tensor Decomposition" by Durham et al presents a novel imputation method for epigenomic data, called PREDICTD, which is based on a tensorial decomposition of the data (arranged as a 3-dimensional tensor) using the PARAFAC algorithm. The method is shown to perform similarly to ChromImpute, the current "state-of-the-art", if not better, depending on the specific evaluation measure used. However, there are clear important advantages to using PREDICTD, such as the biological interpretability of the inferred latent factors, which is very noteworthy given that the main drawback with ChromImpute is that it can't offer such ease of biological interpretation.

Thus, this is a really important manuscript. It represents a "tour-de-force", not necessarily in terms of the statistical approach (PARAFAC was proposed back in the 1970s), but in terms of the specific application to large epigenomic data and its implementation, which uses state-of-the-art computer science techniques. Generally speaking, the paper is well written, appears statistically sound and figures are professionally presented. My only big concern is that it is unclear to me how easy other users can apply this method to a wide range of other tasks. I also have a number of other major concerns, some of a technical nature (see below).

Major concerns:

1) Is the method really user-friendly? I am not convinced that this method is that user-friendly. I can see the value of an imputation method like PREDICTD or ChromImpute, but it would appear to me that ChromImpute is easier to implement? I visited the github website where the software is available, but it is very unclear how to implement this on a local cluster (or indeed if this is at all possible), assuming for instance that someone does not want and can't access cloud services. It is also unclear how long it would take to run PREDICTD on a tensor of size $Z \times Y \times W$, as a function of Z , Y and W on say a standard cluster. I think that the authors should really make a better effort to try to make their software more user-friendly and more easily applicable. For instance, a user might want to use PREDICTD to predict an assay for a cell-type for which it has generated some other data-types. But how this user could use PREDICTD to impute their data is unclear. This appears to be a major limitation of this otherwise great work.

2) Resource or software? Related to the previous point, the manuscript seems more "useful" as a "resource" rather than a software that users can use and implement. That is, the value of this MS is in the provision of the imputed data which would make a valuable resource, but this seems to be ongoing work, as it is planned to publish the imputed genome-wide data in a future ENCODE release.

3) Number of latent factors: A technical concern is that it is very unclear how the number of latent factors was estimated. I could not find this key information. From one of the figures (e.g. Fig.3C) it seems that the authors inferred 100 latent factors, but how this number was determined is never explained. This is a notoriously hard problem, specially with an algorithm such as PARAFAC, and so I would like to see a more rigorous description of this step.

4) PARAFAC or nonnegative tensorial factorization? Another technical issue which is not mentioned is whether PARAFAC was implemented as in the original paper dating back to 1970 (i.e. without non-negativity constraints), or whether the authors implemented a more novel non-negative tensorial factorization. Given the non-negative nature of the epigenomic data, the latter would seem more appropriate, yet the heatmaps in the figures seem to indicate negative as well as positive values for the latent factors, and therefore that PARAFAC was implemented on centred data and thus without non-negativity constraints. This leads me to another concern, which is that of redundancy of latent factors (a notorious problem with PARAFAC-see original papers). Many of the inferred latent factors could be capturing exactly the same features yet potentially with a different sign (indeed latent factors from PARAFAC can be multiplied by -1 assuming another mode is also multiplied by -1, so signs are not meaningful), so this means that for instance the data shown in Fig.3D could actually be showing a

lot of redundant information. This concern is also related to point-3 above, since redundancy will increase in line with the number of inferred latent factors. This redundancy issue does not seem to have been addressed.

5) Imputation for undercharacterized cell types (subsection 2.3): The data shown in Fig.2C is very nice, but I would like to see more examples. Only one-cell type (CD3 Cord Blood Primary Cells) was considered. The authors should do this for more cell-types to demonstrate that the accuracy achieved is indeed of a general nature.

6) P300 analysis: I liked this analysis a lot, but it would be even nicer if the authors could also show graphically that the genomic latent factors that are marking P300 binding sites also correspond to latent factors in assay space which have particular large absolute weights for the enhancer marks (K3K27ac, H3K4me1). This would then "link" the latent factors in assay and genomic space together and provide more confidence that the method is working as claimed.

Minor concerns:

Clarifications: (i) At the end of second last paragraph in 4.1, the authors state "The first four test set folds were used for training and tuning model parameters". I know exactly what the authors mean to say but to the uninitiated this may sound very confusing. Maybe the authors want to use some different terminology e.g. "bags". (ii) In the last paragraph under 4.3 the authors state that "PARAFACwill find a solution that is unique in the sense that it is rotationally invariant". I am not sure if this is correct. In fact, isn't the solution unique (up to a sign and scaling btw....) because it is **not** rotationally invariant?

Reviewer #2 (Remarks to the Author):

The paper by Durham et al proposes an imputation strategy for epigenomic data based on tensor decomposition. The method is elegant and builds upon recent work in machine learning in problems such as matrix (and tensor) completion for recommender systems. The idea is to learn a decomposition of the (incomplete) data 3D array in terms of a product of (lower dimensional) matrices, which can then be used to obtain an estimate of the missing data values. The authors evaluate the approach on Encode data versus a simple baseline and a stronger baseline of the ChromImpute method by Ernst and Kellis. They also complete the study with a topical application to human accelerated regions (HARs) and their enhancer characterisation.

Overall, the paper is well written and explores a nice idea, but in my opinion fails to make the mark for Nature Communications both on novelty and significance. Here are some comments to substantiate my assessment:

- Unquestionably, the method is elegant and the application of ideas from machine learning to epigenomics should be welcome. However, the value of such applications is critically dependent on their appropriateness to the problem, and this can only be assessed empirically. In this respect, the results are somewhat disappointing. The method is on some measures comparable with the simpler baseline, and on all measures but one it is inferior to the ChromImpute approach. It could still be justified on the grounds of smaller computational overheads, but no evidence is provided in this direction (in fact, I suspect this method is quite intensive as it needs to be run on cloud infrastructure).

- The simplicity of the method is an advantage if this enables more biological insights, but unfortunately PREDICT is not so transparent, due to the latent dimensions which do not have an immediate interpretation. In this respect, methods that directly predict epigenomic features from sequence or other features (such as Benveniste et al PNAS 2014, Whitaker et al Nat. Methods 2015,

Zhou and Troyanskaya Nat Methods 2015) have an advantage, as their results are directly interpretable and have led to new biological insights. Such issues are only briefly commented upon (and no mention is made of methods for prediction, which could be naturally used for imputation) but appear to be quite important.

- The HAR study is nicely executed but does not seem to lead to novel insights and it is mainly confirmatory of previous analyses on a different data set. This is welcome but unlikely to have major impact on the study of HAR biology.
- Claims to the simplicity of the training of the model vis a vis ChromImpute are not substantiated. In particular, the choice of the dimension of the latent factors seems to be a potentially big complicating factor which is not discussed in detail.

Reviewer #3 (Remarks to the Author):

Durham and colleagues developed a novel computational method, termed PREDICTD, to impute epigenomic data. The authors represent a multi-track, multi-cell-type epigenomic datasets as a three dimensional tensor corresponding to cell-type, assay-type, and genomic position, respectively. The basic idea is to decompose this tensor as multiplication of two matrices, thereby reducing complexity. The entries in the resulting matrices are used to impute any (unmeasured) data. They compared PREDICTD with ChromImpute, an existing model in the literature. Their results seem to suggest that overall the two models perform similarly, although in some areas the new method is slightly better.

Large consortia such as ENCODE and Epigenome Roadmap have generated a large amount of epigenomic data, but the cell-types profiled in these consortia are typically different from the ones studied in individual labs. On the other hand, it is financially challenging for individual labs to generate their own epigenomic data comprehensively. Therefore, it is cost beneficial to develop new computational methods that can effectively borrowing information from existing data. The deconvolution approach used in this paper seems valid, but the advantage of PREDICTD over existing methods seems to moderate, limiting its potential impact.

Specific comments:

1. The method description is somewhat vague and not fully motivated. It would be very helpful to clearly state the underlying assumption of the model, and explain intuitively why this is a useful thing to do. Equation (1), what is the purpose for introducing the different terms, c_j , a_k , g_i ? What undesirable properties do the last few terms intend to penalize? How is equation (1) solved? The authors do not go into any details but only cite a publication. It is advisable to briefly describe how that method works for the sake of self-contain.
2. A central issue that needs clarification is how missing data come into play in the model. It is good to know that the training data D does not have to be complete, but it is unclear how missing data are accounted for in model inference, or how to apply the model to impute data in a test dataset.
3. In Fig.2, the authors should also show results from ChromImpute for comparison.
4. For the three cell lines that PREDICTD does not outperform the "main effects" model; it will be helpful for the readers to examine why.
5. The authors pointed out that in some cases, using only two histone marks is sufficient to reproduce the entire epigenome. I wonder if this is because the cell line in question is highly similar to one included in the training dataset.
6. For model comparison, it would be helpful to describe what each performance parameter means.

Reviewer #1

The manuscript “PREDICTD: PaRallel Epigenomics Data Imputation with Cloud-based Tensor Decomposition” by Durham et al presents a novel imputation method for epigenomic data, called PREDICTD, which is based on a tensorial decomposition of the data (arranged as a 3-dimensional tensor) using the PARAFAC algorithm. The method is shown to perform similarly to ChromImpute, the current “state-of-the-art”, if not better, depending on the specific evaluation measure used. However, there are clear important advantages to using PREDICTD, such as the biological interpretability of the inferred latent factors, which is very noteworthy given that the main drawback with ChromImpute is that it can’t offer such ease of biological interpretation.

Thus, this is a really important manuscript. It represents a “tour-de-force”, not necessarily in terms of the statistical approach (PARAFAC was proposed back in the 1970s), but in terms of the specific application to large epigenomic data and its implementation, which uses state-of-the-art computer science techniques. Generally speaking, the paper is well written, appears statistically sound and figures are professionally presented. My only big concern is that it is unclear to me how easy other users can apply this method to a wide range of other tasks. I also have a number of other major concerns, some of a technical nature (see below).

Major concerns:

1a) Is the method really user-friendly? I am not convinced that this method is that user-friendly. I can see the value of an imputation method like PREDICTD or ChromImpute, but it would appear to me that ChromImpute is easier to implement? I visited the github website where the software is available, but it is very unclear how to implement this on a local cluster (or indeed if this is at all possible), assuming for instance that someone does not want and can’t access cloud services.

We understand that there will be some users who would prefer to run the model on a local cluster, and this is certainly possible to do. PREDICTD does not inherently rely on any proprietary software or features of cloud computing; the most important dependency is the Apache Spark distributed computation engine, which is open source and freely available to install locally. The code base is currently configured to read and write to AWS S3 cloud storage, but changing the code to conduct I/O over HDFS or NFS would be relatively straightforward. Given that PREDICTD is open source, we would certainly welcome any community contributions to add to its flexibility in this regard. All of this being said, we would like to note that running the model on a local cluster would remove the considerable advantages to running on the cloud. The most important of these is that the cloud allows one to move the computation to the data. This removes the burden of each user storing and maintaining a separate copy of the large training data set, which is 980 GB. Furthermore, as long as the user sets up their cluster in the US West (Oregon) AWS region, there is no additional charge to read these data from Amazon cloud storage. Another key advantage is that the cloud offers on-demand computational resources that are pre-configured to run the model with minimal customization required at start up (see the GitHub page updates we describe in response 1c below), and these resources are available from anywhere in the world to anyone with an internet connection and an account with Amazon Web Services. We believe that this maximizes the availability and usability of our software, even for labs that have local cluster resources. It would even be possible, with further software engineering, to run PREDICTD as a web service to make it even simpler to run and thereby accessible to those with little computational training. In the end, given these reasons and the trend that cloud computing technology is becoming increasingly useful and cost-effective, we decided to focus our efforts on making PREDICTD useful in the cloud.

The reviewer’s concern about user-friendliness is addressed in our response to point 1c below.

1b) It is also unclear how long it would take to run PREDICTD on a tensor of size $Z \times Y \times W$, as a function of Z , Y and W on a standard cluster.

We added a new section under Methods to address the computational resource requirements of PREDICTD. It reads as follows:

The resource requirements of PREDICTD are not very great considering the size of the model. We find that training a single PREDICTD model on the tensor described in the paper ($127 \times 24 \times 1.3e6$) takes about two to three hours on a two node cluster consisting of a head node with 4 cores (Intel Xeon E5-2676 v3 Haswell or Xeon E5-2686 v4 Broadwell processors) and 16 GB of memory (e.g. an m4.xlarge AWS EC2 instance) and a worker node with 32 cores (Intel Xeon E5-2670 v2 Ivy Bridge) and 244 GB of memory (e.g. an r3.8xlarge AWS EC2 instance). For this manuscript, each experiment was imputed as an average of eight models trained with random starts and different validation sets, so one could train these models in about 24 hours. After training the models, imputing values for the limited subset of genomic positions used for training is quite fast. However, if one needs to impute the whole genome it takes longer because the learned cell type and assay factors must be applied across all genomic locations. To do this without having to store the entire tensor in memory at once (all genomic positions and no missing values), we read in data for batches of genomic positions, train the corresponding genome parameters based on the existing cell type and assay parameters, and then write out the imputed values for each batch. For imputing whole genome data for one new cell type (that is, 24 whole genome imputed experiments) the cluster configuration described above requires an additional 24 hours, for a total of ~ 48 hours for model training and whole genome imputation.

In this manuscript we present a more extreme case in which we impute all 3048 possible experiments in the Roadmap Epigenomics tensor at 25 base pair resolution, and to do this we used a larger worker node to increase throughput. If we use an x1.16xlarge instance as the worker node, which has 64 cores (Intel Xeon E7-8880 v3 Haswell) and 976 GB of memory along with a 8 TB Elastic Block Storage (EBS) Throughput-Optimized Hard Disk Drive (maximum throughput of 500 MB/s), we can use the trained models to impute the whole genome for all 3048 experiments in approximately 50 hours. The resulting imputed tracks represent the consensus of eight models for each experiment, and these experiments were split into five test sets, giving a total of 40 models that took about 120 hours to train. Thus, training and imputation for the 3048 Roadmap Epigenomics tracks takes a total time of ~ 170 hours.

To compare with ChromImpute’s runtime, we can convert this 170 hour wall-clock time to an approximate number of CPU hours required to run PREDICTD on the full tensor. Using the smaller cluster to train the 48 models, we calculate PREDICTD requires about $36 \text{ cores} \times 120 \text{ hours} = 4,320 \text{ CPU hours}$. Switching to the larger model for imputation, we find that PREDICTD consumes about an additional $68 \text{ cores} \times 50 \text{ hours} = 3,400 \text{ CPU hours}$. Thus, in total PREDICTD can train the models and impute 3048 experiments in $\sim 7,720 \text{ CPU hours}$. This run time is more than an order of magnitude less than that quoted in the ChromImpute supplement [1], which indicates a total run time for training and output generation of 103,560 CPU hours. Even taking into account the fact that we imputed about 25% fewer experiments for this paper than were imputed in the ChromImpute manuscript, ChromImpute still requires tens of thousands more CPU hours to train the models and impute the Roadmap Epigenomics tensor than PREDICTD does.

1c) I think that the authors should really make a better effort to try to make their software more user-friendly and more easily applicable. For instance, a user might want to use PREDICTD to predict an assay for a cell-type for which it has generated some other data-types. But how this user could use PREDICTD to impute their data is unclear. This appears to be a major limitation of this otherwise great work.

We agree with reviewer #1 that it is important to make the software as user-friendly as possible, and that the version available at the time of the first review was somewhat deficient in this respect. Accordingly, we have improved the GitHub website in four ways. First, we automated more of the configuration tasks from the original tutorial so that it is shorter and simpler to follow. The documentation has been expanded on the GitHub repository wiki (see <https://github.com/tdurham86/PREDICTD/wiki>) with more details and links to help users set up their AWS account, and a simplified cluster set-up with only four steps. Second, we added a new tutorial that demonstrates how a user can add their own data from a new cell type to the model (see <https://github.com/tdurham86/PREDICTD/wiki/Imputing-data-for-a-new-cell-type>). Third, we added wiki pages documenting the general work flow and command line options for each

of the scripts so that a user can more easily customize the demo scripts to suit their needs. Fourth, we added more information about the run time and cost of training the model to both the wiki site and the manuscript.

2) Resource or software? Related to the previous point, the manuscript seems more “useful” as a “resource” rather than a software that users can use and implement. That is, the value of this MS is in the provision of the imputed data which would make a valuable resource, but this seems to be ongoing work, as it is planned to publish the imputed genome-wide data in a future ENCODE release.

We have submitted all 3048 imputed data sets from the tensor reported in this manuscript to the ENCODE Data Coordination Center (DCC). Note that due to some technical difficulties that the DCC encountered, the files are not yet available for download, but they will be available soon through the user interface at <https://www.encodeproject.org>. We updated the methods text under the section “Imputing the whole genome” as follows:

Although for the purpose of analyzing the PREDICTD model we only imputed about 1% of the genome, we generated whole genome imputed tracks in bigWig format for the UCSC Genome Browser. These tracks are available for download from the ENCODE project website (<http://encodeproject.org>).

We also highlighted these reference data sets at the end of the Introduction section of the main manuscript in the following sentence:

We used PREDICTD to impute the results for 3048 experiments across 127 cell types and 24 assays from the Roadmap Epigenomics project, and these imputed data are available for download through ENCODE (<https://www.encodeproject.org/>).

3) Number of latent factors: A technical concern is that it is very unclear how the number of latent factors was estimated. I could not find this key information. From one of the figures (e.g. Fig.3C) it seems that the authors inferred 100 latent factors, but how this number was determined is never explained. This is a notoriously hard problem, specially with an algorithm such as PARAFAC, and so I would like to see a more rigorous description of this step.

The reviewer is correct that the number of latent factors is not simple to select in PARAFAC models. We have expanded and clarified our explanation of how we tuned the model hyperparameters, and we also included more discussion of the importance of the latent factor setting of the model. This re-written section reads as follows:

One of the challenges of working with this type of model is that there are many hyperparameters to tune. These include the number of latent factors L , the learning rate η , the learning rate decay ϕ_η , the Adam first moment coefficient β_1 and its decay rate ϕ_{β_1} , and a regularization coefficient for each latent parameter matrix ($\lambda_{\mathbf{A}}$, $\lambda_{\mathbf{C}}$, $\lambda_{\mathbf{G}}$).

Of these hyperparameters, perhaps the most important one for PREDICTD performance is the number of latent factors. This setting controls the dimensionality of the model and thus the number of parameters that must be optimized during model training. Ideally, assuming a perfect match between the modeling approach and the data, the number of latent factors will equal the true underlying dimensionality of the data set. However, in practice this assumption does not really hold. First, real world data is often noisy enough that the “true” dimensionality of the input data is the full rank. In such cases, we are forced to use fewer latent factors that approximate the dimensionality of theoretical, noiseless, data. Second, PREDICTD implements the original PARAFAC specification [2], which relies on simple linear combinations of the corresponding latent factors in each dimension. However, in real data there could be factors that have nonlinear relationships, and there is evidence that PARAFAC in some cases will attempt to fit these relationships by adding additional factors to explicitly take them into account as if they are additional linear terms. This phenomenon was explored in an example from the original

Table 1: **Hyperparameter values.** The third column indicates whether the hyperparameter value was selected using Spearmint, and an asterisk indicates the final value was tuned by hand after Spearmint optimization.

Hyperparameter	Value	Spearmint?
η	0.005	Y
ϕ_η	$1 - 1e^{-6}$	N
β_1	0.9	N
ϕ_{β_1}	$1 - 1e^{-6}$	N
β_2	0.999	N
L	100	Y*
λ_C	3.66	Y
λ_A	$1.23e^{-7}$	Y
λ_G	$1.23e^{-5}$	Y
λ_{G_2}	2.9	N

PARAFAC paper, in which the best PARAFAC solution for a rank-2 synthetic data set with an interaction between the two dimensions used three latent factors: one for each dimension, plus another for the product of the two [2]. In the end, the best number of latent factors to use is simply the number that minimizes the error of the model while preserving its generalization performance, and this must be evaluated empirically.

Empirically searching for the best number of latent factors is non-trivial. The number of latent factors changes the dimensionality of the model and thus the balance between bias and variance, which means that the regularization coefficients must be tuned in parallel with the latent factors. A simple strategy that has been shown to be surprisingly effective at searching high dimensional hyperparameter space is simple random search, in which different random hyperparameter values are tested until a combination is found that provides good performance of the model [3]. Although the simplicity of the random choice strategy makes it very appealing, it can still require many iterations before one is confident that good hyperparameters have been found, which is a severe drawback when trying to optimize settings for a model like PREDICTD that takes two to three hours to train. Thus, hoping to find good hyperparameter settings in as few iterations as possible, we used an auto-tuning software package called Spearmint [4]. Spearmint treats the PREDICTD model as a black box function and iteratively tries different hyperparameter settings; it uses Bayesian optimization to fit a Gaussian process that can select the hyperparameter settings that will maximize the expected improvement in model performance in the next iteration. There is still some debate in the field as to whether or not this kind of auto-tuning strategy reliably finds better hyperparameter values than simple random search [5]; however, evidence shows that such Bayesian approaches tend to converge to a good selection of hyperparameters in fewer iterations than random search [4].

We ran Spearmint multiple times as we developed the PREDICTD model, and each run was allowed between 14 and 45 iterations. To save time on each of these training runs, we relaxed the convergence criteria to use a larger shift between the two samples in the Wilcoxon rank-sum test ($5e^{-05}$ instead of $1e^{-05}$), and we did not attempt to halve the learning rate and continue training as we did for the final model. After running Spearmint, we continued to fine-tune the model by occasionally manually modifying the hyperparameter values until we settled on the values reported here (Table 1). It is important to note that there may be even better hyperparameter settings that our search did not encounter, and as new discoveries concerning hyperparameter tuning unfold in the machine learning literature, the settings for PREDICTD can be revisited to perhaps further increase its performance.

4a) PARAFAC or nonnegative tensorial factorization? Another technical issue which is not mentioned is whether PARAFAC was implemented as in the original paper dating back to 1970 (i.e. without non-negativity constraints), or whether the authors implemented a more

novel non-negative tensorial factorization. Given the non-negative nature of the epigenomic data, the latter would seem more appropriate, yet the heatmaps in the figures seem to indicate negative as well as positive values for the latent factors, and therefore that PARAFAC was implemented on centred data and thus without non-negativity constraints.

PREDICTD is implemented as in the 1970 PARAFAC publication, with no non-negativity constraints on the parameter values. We added the following clarification in the methods text:

Note that we implemented the model as described in the original publication [2], and we included no additional constraints on the model during training except what was imposed by the L2 regularization terms in the objective and the implicit regularization naturally imposed by using a relatively simple linear model on complex data with potentially non-linear underlying factors.

In response to Reviewer 1’s suggestion, we tried training PREDICTD with a simple non-negativity constraint enforced on the parameters during the parallel stochastic gradient descent iterations. This constraint set any parameters assigned a negative value after an SGD update to zero, and continued training. We find that this constraint alone does not improve PREDICTD performance. We also show in Fig. 1 (in this document) that the vast majority of imputed data points have a positive sign, despite the lack of non-negativity constraints on the model parameters, and those imputed values that are negative tend to have small magnitude. Thus, our solution is to simply truncate the lower end of the range of the imputed values to zero. Nevertheless, we appreciate Reviewer 1’s suggestion, and in future work it will be interesting to further explore this direction with more hyperparameter tuning and additional experiments to try other ways of enforcing a non-negativity constraint.

Figure 1: **Most imputed values are positive despite allowing model parameters to have negative values.** The ct, assay, and genome parameters from one of the 48 models trained on the ENCODE Pilot Regions plus non-coding human accelerated regions were used to impute values for 100000 randomly selected genomic positions, and these imputed values are plotted here as a histogram.

4b) This leads me to another concern, which is that of redundancy of latent factors (a notorious problem with PARAFAC-see original papers). Many of the inferred latent factors could be capturing exactly the same features yet potentially with a different sign (indeed latent factors from PARAFAC can be multiplied by -1 assuming another mode is also multiplied by -1, so signs are not meaningful), so this means that for instance the data shown in Fig.3D could actually be showing a lot of redundant information. This concern is also related to point-3

above, since redundancy will increase in line with the number of inferred latent factors. This redundancy issue does not seem to have been addressed.

The reviewer raises a good point here, which we addressed by investigating empirically the extent of redundancy among the latent factors. A correlation analysis (Fig. 2 in this document) suggests that there is indeed some correlation structure in the latent factors over the assay parameters, but this structure almost entirely disappears for the higher dimension cell type and genome parameters.

Figure 2: **There is little overall correlation among latent factors.** Trained parameter values were used to calculate the Pearson correlation between pairs of latent factors in each axis. **a.** Pairwise latent factor Pearson correlation matrix using genome parameters from 100000 randomly-selected genomic positions. **b.** Pairwise latent factor Pearson correlation matrix using cell type parameters. **c.** Pairwise latent factor Pearson correlation matrix using assay parameters.

This empirical analysis is not conclusive, because some redundant information may still be captured in more complicated patterns of redundancy that involve three or more latent factors. In general, we expect redundancy to decrease with fewer latent factors, and that this redundancy would manifest as an increase in modeling error when the number of latent factors drops below some threshold. To check this, we picked a range of latent factor values from 10 to 150 and trained three randomly-initialized PREDICTD models for each (Fig. 3, this document). The results show that our choice of 100 latent factors is among the best based on validation MSE. However, MSE does not dramatically rise if the number of latent factors is reduced to 60-90, suggesting that there may be some redundancy in the latent factors. Nevertheless, models with 130 and 150 latent factors appear to have better generalization performance than the models with 100 latent factors, indicating that if it is present, some redundancy might actually help model performance. Our results here are suggestive, but not definitive. Due to the interaction between the model dimensionality and regularization settings, to fully investigate this phenomenon one would have to conduct a separate hyperparameter search for each individual latent factor value. Such an analysis would require substantial compute resources, and we leave it to future work that will explore whether changing the model dimensionality can improve the interpretability of the trained latent factors.

5) Imputation for undercharacterized cell types (subsection 2.3): The data shown in Fig.2C is very nice, but I would like to see more examples. Only one-cell type (CD3 Cord Blood Primary Cells) was considered. The authors should do this for more cell-types to demonstrate that the accuracy achieved is indeed of a general nature.

As per the reviewer’s suggestion, we have run a similar analysis by holding out data sets for all assays except for H3K4me3 in each of four additional cell types: “GM12878 Lymphoblastoid”, “Fetal Muscle Trunk”, “Brain Anterior Caudate”, and “Lung.” These cell types represent very different tissues with a range of assays available, and in all cases the performance when training solely on H3K4me3 is surprisingly good compared to the better-supported models. We modified the manuscript text as follows to reflect these

additional analyses, and we provide Fig. 4 (this document, Fig. S7 in manuscript), along with Figs. 5, 6, 7, 8 (all this document, Figs. S8-S11 in the manuscript), as an additional supplementary figure in the paper.

Comparing the performance metrics between these experiments and the imputed results from our original models trained on the five test folds, we find that the results of training with just H3K4me3 or both H3K4me3 and H3K9me3 are nearly as good as (and sometimes better than) the results from the original models with training data including five or six experiments for this cell type (Fig. 2C). Imputing only based on H3K4me1 signal did not perform as well as imputing based on only H3K4me3. This observation is consistent with previous results on assay prioritization [6] indicating that H3K4me3 is the most information-rich assay. Furthermore, this result is not specific to the “CD3 Cord Blood Primary Cells” cell type. We find that the results for imputing four other cell types (“GM12878 Lymphoblastoid”, “Fetal Muscle Trunk”, “Brain Anterior Caudate”, and “Lung”) just based on H3K4me3 signal showed similar results (Fig. S7). We conclude that PREDICTD performs well on under-characterized cell types and will be useful for studying new cell types for which few data sets are present.

6) P300 analysis: I liked this analysis a lot, but it would be even nicer if the authors could also show graphically that the genomic latent factors that are marking P300 binding sites also correspond to latent factors in assay space which have particular large absolute weights for the enhancer marks (K3K27ac, H3K4me1). This would then “link” the latent factors in assay and genomic space together and provide more confidence that the method is working as claimed.

This is a very nice suggestion. To address this point, we took the parameters from the assay factor matrix, both alone and combined with the cell type parameters, and split them into two groups based on whether they are from latent factors that correspond to a “peak” or “flank” genome latent factor. We compare the distributions of these parameter values using a Receiver Operating Characteristic (ROC) curve to test whether the “peak” or “flank” latent factors tend to be larger. Our analysis shows that there appears to be some difference between the “peak” and “flank” groups of latent factors in the parameters from the non-genome factor matrices; however, it is not obvious from the assay parameter values alone, for which we find no relationship between the “peak” and “flank” groups of genome latent factors and their corresponding factors at enhancer-associated assays like DNase, H3K4me1, and H3K27ac (Fig. 9a, this document). In contrast, if we take into account the cell type parameters and the interactions among the “peak” and “flank” latent factors, we do see a difference by assay (Fig. 9, this document). We combined the assay and cell type parameters by taking an outer product on each latent factor, which resulted in a tensor of (cell type \times assay \times latent factors), and then taking the dot product of the latent factor fibers of this tensor with the genome parameters for just the “peak” or “flank” latent factors to get two values for each cell type/assay pair: one “peak” value and one “flank” value. We were somewhat surprised to find by this analysis that the “flank” dot product values tend to be higher for enhancer-associated marks (Fig. 9b, this document) than the “peak” values, and that non-enhancer associated marks (i.e. H3K27me3, H3K9me3, and H3K36me3) are the opposite. It could be that in order to model assays that produce sharper, taller peaks (as enhancer-associated assays tend to do) PREDICTD learns to have larger amplitudes in the genome parameters that are then “tuned down” by the cell type and/or assay parameters.

Minor concerns:

Clarifications: **(i) At the end of second last paragraph in 4.1, the authors state “The first four test set folds were used for training and tuning model parameters”. I know exactly what the authors mean to say but to the uninitiated this may sound very confusing. Maybe the authors want to use some different terminology e.g. “bags”.**

Thank you for pointing out this confusing wording. We have corrected the manuscript to use the word “sets” instead of “folds”.

(ii) In the last paragraph under 4.3 the authors state that “PARAFAC .will find a solution that is unique in the sense that it is rotationally invariant”. I am not sure if this is correct. In

Figure 3: **Reducing the number of latent factors does not improve model results.** Bars show the average validation MSE of three randomly initialized PREDICTD models trained with different numbers of latent factors with the rest of the hyperparameters fixed. Error bars show the standard deviation. The red bar highlights the results for models trained using the same number of latent factors as in the manuscript (100 latent factors), and in these models achieve an average validation MSE of 0.1325 (black line).

Figure 4: **PREDICTD imputes data well for a variety of cell types trained only on H3K4me3 assay data.** Global mean squared error of results for four cell types (**a.** Brain Anterior Caudate, **b.** Fetal Muscle Trunk, **c.** GM12878 Lymphoblastoid, and **d.** Lung) imputed by the mean of two PREDICTD models each that were trained only on the H3K4me3 data for that cell type.

Figure 5: Quality measures for using PREDICTD to impute Brain Anterior Caudate assays based only on H3K4me3, as compared with the full results for PREDICTD, ChromImpute, and the average imputation value of PREDICTD and ChromImpute.

Figure 6: Quality measures for using PREDICTD to impute Fetal Muscle Trunk assays based only on H3K4me3, as compared with the full results for PREDICTD, ChromImpute, and the average imputation value of PREDICTD and ChromImpute.

Figure 7: Quality measures for using PREDICTD to impute GM12878 Lymphoblastoid assays based only on H3K4me3, as compared with the full results for PREDICTD, ChromImpute, and the average imputation value of PREDICTD and ChromImpute.

Figure 8: Quality measures for using PREDICTD to impute Lung assays based only on H3K4me3, as compared with the full results for PREDICTD, ChromImpute, and the average imputation value of PREDICTD and ChromImpute.

fact, isn't the solution unique (up to a sign and scaling btw.) because it is ***not*** rotationally invariant?

Again, thank you to the reviewer for pointing out this error. We have corrected this language as follows:

The PARAFAC model also has the nice property that as long as mild conditions hold it will find a solution that is unique with respect to rotation transformations [7, 8]; this is not a property of other tensor factorization approaches, including Tucker decomposition, which was used in [9].

Figure 9: **The relationship among corresponding factors from different tensor axes is non-trivial.** Receiver operating characteristic curves between values for latent factors associated with P300 peak regions, or their flanking regions. Our initial naïve hypothesis was that latent factors with high magnitude over P300 peaks in the genome dimension would have correspondingly high magnitude for enhancer-associated assays in the assay dimension of the model, but the relationship is not so simple. Understanding the relationship between the assay and genome parameters requires contribution from the cell type parameters also. **a.** ROC curves based on assay parameter values that correspond to either the peak- or flank-associated genome latent factors. **b.** ROC curves based on the dot product of cell type and assay parameters that correspond to either peak- or flank-associated genome latent factors.

Reviewer #2

The paper by Durham et al proposes an imputation strategy for epigenomic data based on tensor decomposition. The method is elegant and builds upon recent work in machine learning in problems such as matrix (and tensor) completion for recommender systems. The idea is to learn a decomposition of the (incomplete) data 3D array in terms of a product of (lower dimensional) matrices, which can then be used to obtain an estimate of the missing data values. The authors evaluate the approach on Encode data versus a simple baseline and a stronger baseline of the ChromImpute method by Ernst and Kellis. They also complete the study with a topical application to human accelerated regions (HARs) and their enhancer characterisation.

Overall, the paper is well written and explores a nice idea, but in my opinion fails to make the mark for Nature Communications both on novelty and significance. Here are some comments to substantiate my assessment:

1a) Unquestionably, the method is elegant and the application of ideas from machine learning to epigenomics should be welcome. However, the value of such applications is critically dependent on their appropriateness to the problem, and this can only be assessed empirically. In this respect, the results are somewhat disappointing. The method is on some measures comparable with the simpler baseline, and on all measures but one it is inferior to the ChromImpute approach.

We thank the reviewer for complimenting our interdisciplinary approach, and we agree that the model does not substantially out-perform ChromImpute. However, we argue that the global mean squared error, which is a measure by which PREDICTD outperforms ChromImpute, is the most important measure by which to assess the model's performance for the following reasons. First, these models are designed to predict the signal of epigenomics assays; the mean squared error most directly captures this goal by penalizing any predictions that are far from the observed signal, and thus is the most natural measure to use. Second, this measure is central to the PREDICTD objective function. In principle, one could design alternative objective functions for PREDICTD that would optimize for different quality measures, and in those cases PREDICTD would likely perform better on alternative measures.

1b) It could still be justified on the grounds of smaller computational overheads, but no evidence is provided in this direction (in fact, I suspect this method is quite intensive as it needs to be run on cloud infrastructure).

Considering the size of the model, the resource requirements of PREDICTD are rather modest, and as we mentioned in our response to a comment from Reviewer #1, the cloud is not a strict requirement for running PREDICTD:

We understand that there will be some users who would prefer to run the model on a local cluster, and this is certainly possible to do. PREDICTD does not inherently rely on any proprietary software or features of cloud computing; the most important dependency is the Apache Spark distributed computation engine, which is open source and freely available to install locally. The code base is currently configured to read and write to AWS S3 cloud storage, but changing the code to conduct I/O over HDFS or NFS would be relatively straightforward. Given that PREDICTD is open source, we would certainly welcome any community contributions to add to its flexibility in this regard. All of this being said, we would like to note that running the model on a local cluster would remove the considerable advantages to running on the cloud. The most important of these is that the cloud allows one to move the computation to the data. This removes the burden of each user storing and maintaining a separate copy of the large training data set, which is 980 GB. Furthermore, as long as the user sets up their cluster in the US West (Oregon) AWS region, there is no additional charge to read these data from Amazon cloud storage. Another key advantage is that the cloud offers on-demand computational resources that are pre-configured to run the model with minimal customization required at start up (see the GitHub page updates we describe in response 1c below), and these resources are available from anywhere in the world to anyone with an internet connection and an account with Amazon Web Services.

We believe that this maximizes the availability and usability of our software, even for labs that have local cluster resources. It would even be possible, with further software engineering, to run PREDICTD as a web service to make it even simpler to run and thereby accessible to those with little computational training. In the end, given these reasons and the trend that cloud computing technology is becoming increasingly useful and cost-effective, we decided to focus our efforts on making PREDICTD useful in the cloud.

Furthermore, we added more details about the resource requirements of running PREDICTD to the methods section of the manuscript, and we find that PREDICTD is far less computationally intensive than ChromImpute on a per-experiment basis. Here is the relevant text:

The resource requirements of PREDICTD are not very great considering the size of the model. We find that training a single PREDICTD model on the tensor described in the paper ($127 \times 24 \times 1.3e6$) takes about two to three hours on a two node cluster consisting of a head node with 4 cores (Intel Xeon E5-2676 v3 Haswell or Xeon E5-2686 v4 Broadwell processors) and 16 GB of memory (e.g. an m4.xlarge AWS EC2 instance) and a worker node with 32 cores (Intel Xeon E5-2670 v2 Ivy Bridge) and 244 GB of memory (e.g. an r3.8xlarge AWS EC2 instance). For this manuscript, each experiment was imputed as an average of eight models trained with random starts and different validation sets, so one could train these models to use for imputation in about 24 hours. After training the models, imputing values for the limited subset of genomic positions used for training is quite fast. However, if one needs to impute the whole genome it takes longer because the learned cell type and assay factors must be applied across all genomic locations. To do this without having to store the entire tensor in memory at once (all genomic positions and no missing values), we read in data for batches of genomic positions, train the corresponding genome parameters based on the existing cell type and assay parameters, and then write out the imputed values for each batch. For imputing whole genome data for one new cell type (that is, 24 whole genome imputed experiments) the cluster configuration described above requires an additional 24 hours, for a total of ~ 48 hours for model training and whole genome imputation.

In this manuscript we present a more extreme case in which we impute all 3048 possible experiments in the Roadmap Epigenomics tensor at 25 base pair resolution, and to do this we used a larger worker node to increase throughput. If we use a x1.16xlarge instance as the worker node, which has 64 cores (Intel Xeon E7-8880 v3 Haswell) and 976 GB of memory along with a 8 TB Elastic Block Storage (EBS) Throughput-Optimized Hard Disk Drive (maximum throughput of 500 MB/s), we can use the trained models to impute the whole genome for all 3048 experiments in approximately 50 hours. The resulting imputed tracks represent the consensus of eight models for each experiment, and these experiments were split into five test sets, giving a total of 40 models that took about 120 hours to train. Thus, training and imputation for the 3048 Roadmap Epigenomics tracks takes a total time of ~ 170 hours.

To compare with ChromImpute’s runtime, we can convert this 170 hour wall-clock time to an approximate number of CPU hours required to run PREDICTD on the full tensor. Using the smaller cluster to train the 48 models, we calculate PREDICTD requires about $36 \text{ cores} \times 120 \text{ hours} = 4,320 \text{ CPU hours}$. Switching to the larger cluster for imputation, we find that PREDICTD consumes about an additional $68 \text{ cores} \times 50 \text{ hours} = 3,400 \text{ CPU hours}$. Thus, in total PREDICTD can train the models and impute 3048 experiments in $\sim 7,720 \text{ CPU hours}$. This run time is more than an order of magnitude less than that quoted in the ChromImpute supplement [1], which reports that ChromImpute requires a total run time of 103,560 CPU hours for model training and output generation. Even taking into account the fact that we imputed about 25% fewer experiments for this paper than were imputed in the ChromImpute manuscript, ChromImpute still requires tens of thousands more CPU hours to train the models and impute the Roadmap Epigenomics tensor than PREDICTD does.

In addition, this compute time does not cost very much on Amazon Web Services. The small cluster referenced above costs about \$0.90 per hour, which means that training the 40 models for our manuscript can be done for about \$108.60. The larger cluster that we used for applying the models to the genome is more expensive, at about \$2.20 per hour, but requires less time and adds only about \$110.00. This means that the

cost for the compute resources required to train PREDICTD and impute the whole Roadmap Epigenomics project is about \$218.60, which is only \sim \$0.07 per imputed experiment.

Even in a case that does not benefit as much from economies of scale, in which a researcher wants to impute data for a single new cell type by averaging five PREDICTD models and doing whole genome imputation, that researcher would only pay about \$2 per experiment. The model training and output should take about 48 hours and require significantly less cluster storage (maybe 500 GB instead of the 8TB we used for all of Roadmap), and would cost about $\$1.06/\text{hr} \times 48 \text{ hours} = \50.88 total. Assuming the researcher uses the same 24 assays that we report here, this researcher would have paid \$2.12 per imputed experiment. This is still extremely cost effective compared to doing the experiments at the bench at a cost of hundreds of dollars per sample.

2) The simplicity of the method is an advantage if this enables more biological insights, but unfortunately PREDICT is not so transparent, due to the latent dimensions which do not have an immediate interpretation. In this respect, methods that directly predict epigenomic features from sequence or other features (such as Benveniste et al PNAS 2014, Whitaker et al Nat. Methods 2015, Zhou and Troyanskaya Nat Methods 2015) have an advantage, as their results are directly interpretable and have led to new biological insights. Such issues are only briefly commented upon (and no mention is made of methods for prediction, which could be naturally used for imputation) but appear to be quite important.

The reviewer points out several important works that make predictions from epigenomics data. Benveniste *et al* 2014 [10] present a logistic regression model to predict histone modification patterns from transcription factor ChIP-seq data; Whitaker *et al* 2015 [11] found that DNA motifs can predict histone modification profiles using random forest classifiers; and Zhou and Troyanskaya 2015 [12] used a deep neural network to predict the effect of DNA sequence changes on the patterns of epigenomic marks. These works indeed provide fascinating biological insights; however, they are all of a more limited scope than our study in several important respects.

First, these studies do not make predictions at the same breadth and resolution of PREDICTD. We train PREDICTD on all cell types in the Roadmap Consolidated data set and nearly all assays (24 of 34). Imputing data at this scale is non-trivial, and in contrast, these other studies were limited to predicting on just a few cell types and/or assays at either a lower genomic resolution (DeepSEA trains on 200 bp bins) or at limited locations in the genome (just peak locations or transcription start sites). Thus, although in principle these approaches could be used to impute epigenomic signal genome-wide, it is unlikely that they would realistically scale to include as many data sets as PREDICTD.

Second, each of these studies used a classifier to predict binarized epigenomics signal, which removes biologically relevant information, for example that which is provided by the peak shape [13–15]. Schweikert *et al.* [15] show that peak shape of H3K4me3 is related to which chromatin regulators are depositing the histone modification. Datta *et al.* [13] used the ChIP-seq signal intensity to detect cooperative binding of transcription factors. And Cremona *et al.* [14] show that peak shape for the GATA-1 transcription factor, which is important for erythroid development, distinguishes peaks that exhibit co-binding with other erythroid transcription factors and also that are associated with genes that are more down-regulated in expression after GATA-1 knockout than genes associated with other GATA-1 peaks. Thus, our raw imputed signal could provide more insight into the relationships between ChIP-seq targets than simple peak prediction.

Last, although the models referenced by Reviewer #2 undoubtedly provide interesting biological insight, their output is more specialized and presumes more about what kinds of questions other users will want to ask. PREDICTD is designed to generate minimally processed data sets that researchers can use for custom analyses. For example, we use these imputed data both to recover known enhancer activity patterns and to discover novel enhancer candidates among non-coding human accelerated regions. In particular, the fact that PREDICTD is unsupervised with respect to genomic elements or other known biological phenomena means patterns uncovered by the latent factors hold the possibility of discovering new types or subtypes of genomic elements that would be missed by these other approaches.

Please note that We did not directly incorporate this discussion into the manuscript due to length considerations, but we will be happy to do so if the editor and reviewers would like it included.

3) The HAR study is nicely executed but does not seem to lead to novel insights and it is

mainly confirmatory of previous analyses on a different data set. This is welcome but unlikely to have major impact on the study of HAR biology.

Our analysis uncovers tissue-specific enhancer activity for many ncHARs that previously had no tissue-specific enhancer activity assigned. Furthermore, our analysis leverages data from 21 more assays and 106 more cell types than Capra, et al. 2013 [16], many of which have, to our knowledge, never been examined for ncHAR enhancer activity before. We also include the following statement in the manuscript to clarify our contribution to the study of ncHAR biology:

The question of whether ncHARs are active enhancers in modern humans or whether they are regions that formerly had enhancer activity that has been lost over the course of our evolution is a central question to the study of ncHAR biology. With this analysis we shed more light on which ncHARs have enhancer activity, and even provide some insight into the relevant developmental stage for such activity, as our cell types are derived from embryonic, fetal, and adult tissues.

4) Claims to the simplicity of the training of the model vis a vis ChromImpute are not substantiated. In particular, the choice of the dimension of the latent factors seems to be a potentially big complicating factor which is not discussed in detail.

We briefly describe the training procedure of ChromImpute in the Introduction to our manuscript. We wish to give the reader a sense of how ChromImpute operates, as well as to highlight the number of steps involved in the feature selection strategy. The ChromImpute training strategy is quite complicated and arbitrary compared to the PREDICTD model, which models all data at once and involves no feature extraction steps. This section reads as follows:

Ernst and Kellis pioneered this imputation approach, and achieved remarkable accuracy with their method, ChromImpute [1]. Briefly, this method imputes data for a particular target assay in a particular target cell type by 1) finding the top ten cell types most correlated with the target cell type based on data from non-target assays, 2) extracting features from the data for the target assay from the top ten non-target cell types, and also extracting features from the data for non-target assays in the target cell type, and 3) training a regression tree for each of the top ten most correlated cell types. Data points along the genome are imputed as the mean predicted value from the collection of trained regression trees. Although ChromImpute produces highly accurate imputed data, this training scheme is complicated and not very intuitive, and results in a fragmented model of the epigenome that is very difficult to interpret. We hypothesized that an alternative approach, in which a single joint model learns to impute all experiments at once, would simplify model training and improve interpretability while maintaining accurate imputation of missing data.

Furthermore, our strategy for imputing all Roadmap Epigenomics experiments is simpler than that used for ChromImpute. In their training of ChromImpute, Ernst and Kellis [1] used a hierarchical approach in which Tier 1 assays, which have the most observed data available, are trained only on data from other Tier 1 assays, Tier 2 assays are trained on only Tier 1 and Tier 2 assay data, and Tier 3 assays are trained on all data. For PREDICTD, we simply assigned observed data to test and validation sets and then trained the models.

We agree with Reviewer #2 that the selection of the number of latent factors required more explanation. To address this, we added a detailed discussion of the choice of the number of latent factors, quoted in response to Reviewer #1, point 3.

Reviewer #3

Durham and colleagues developed a novel computational method, termed PREDICTD, to impute epigenomic data. The authors represent a multi-track, multi-cell-type epigenomic datasets as a three dimensional tensor corresponding to cell-type, assay-type, and genomic position, respectively. The basic idea is to decompose this tensor as multiplication of two matrices, thereby reducing complexity. The entries in the resulting matrices are used to impute any (unmeasured) data. They compared PREDICTD with ChromImpute, an existing model in the literature. Their results seem to suggest that overall the two models perform similarly, although in some areas the new method is slightly better.

Large consortia such as ENCODE and Epigenome Roadmap have generated a large amount of epigenomic data, but the cell-types profiled in these consortia are typically different from the ones studied in individual labs. On the other hand, it is financially challenging for individual labs to generate their own epigenomic data comprehensively. Therefore, it is cost beneficial to develop new computational methods that can effectively borrowing information from existing data. The deconvolution approach used in this paper seems valid, but the advantage of PREDICTD over existing methods seems to moderate, limiting its potential impact.

Specific comments:

1a. The method description is somewhat vague and not fully motivated. It would be very helpful to clearly state the underlying assumption of the model, and explain intuitively why this is a useful thing to do.

We apologize for not being more clear in motivating and describing the method, and we agree that more details would make our paper more accessible to a wider audience of readers. We have added more detail and explanations to our description of the PREDICTD model and the stochastic gradient descent optimization procedure. We also added details to the beginning of the Model section of Methods describing the modeling choices that led us to use PARAFAC. This new text reads as follows:

Three main “axes” contribute to the observed biological signal in epigenomic maps: the cell type, the assay, and the genomic location that was measured. Having three axes on which to distribute the available data sets naturally lends the full data set the structure of the three dimensional tensor (i.e., a stack of two dimensional matrices) in which the size of one dimension corresponds to the number of cell types in the data set, another to the number of assays, and the third to the number of genomic locations. One might want to use other qualities or attributes to analyze the data (subject to one’s research question and having enough training data), such as the lab that generated the data, the treatment that was applied, etc., but we are interested in parsing the data along the main cell type, assay, and genomic locus axes so that our model can most generally describe the biological phenomena in normal tissues.

Motivated by this 3D structure, we use tensor decomposition because this type of method factors the full data tensor into smaller components that summarize the key contributions of each axis to the total data. The key to using tensor decomposition for imputing missing data in the tensor is that the smaller components that are learned by the model do not have missing values by definition. This means that when we recombine the components to reconstruct the original tensor, the resulting reconstructed tensor will not only have values that approximate the existing data in the original tensor, but also predicted values for any missing entries in the original tensor. Our particular strategy, known in the literature as PARAFAC, finds a two-dimensional matrix (i.e. one “smaller component”) for each dimension. All such matrices are of the same size along one dimension; this dimension is what we refer to as the number of “latent factors.” The number of latent factors determines the complexity of the model, how well it can capture the information in each axis of the tensor, and thus how well the reconstructed tensor matches the original tensor. Each element of the reconstructed tensor is calculated by multiplying the three corresponding values (one from each matrix) for each latent factor and then summing those products to arrive at a single number.

In order to perform imputation, we train the PREDICTD tensor decomposition model using the PARAFAC/CANDECOMP procedure [2, 17], which can very naturally model the three dimensional problem explained above. It also has the added advantages of being relatively simple to implement, it has the ability to scale to a large tensor size, and it holds the possibility of producing latent factors that can provide biological insight.

1b. Equation (1), what is the purpose for introducing the different terms, c_j , a_k , g_i ?
Here is Equation (1) for reference:

$$\operatorname{argmin}_{\mathbf{C}, \mathbf{A}, \mathbf{G}, \mathbf{c}, \mathbf{a}, \mathbf{g}} \sum_{j,k,i \in \mathcal{S}^{\text{train}}} \left(\mathcal{D}_{j,k,i}^{\text{train}} - \left[\sum_{l=1}^L \mathbf{C}_{j,l} * \mathbf{A}_{k,l} * \mathbf{G}_{i,l} + \mathbf{c}_j + \mathbf{a}_k + \mathbf{g}_i \right] \right)^2 + \lambda_{\mathbf{C}} \|\mathbf{C}\|_2^2 + \lambda_{\mathbf{A}} \|\mathbf{A}\|_2^2 + \lambda_{\mathbf{G}} \|\mathbf{G}\|_2^2 \quad (1)$$

We added the following clarification to the text of the model description in the methods section:

We factor the tensor \mathcal{D} into three factor matrices, and three bias vectors \mathbf{a} , \mathbf{c} , and \mathbf{g} . These bias vectors are meant to capture global biases for each cell type, assay, or genomic location, respectively. Essentially, these terms subtract out the mean for each cell type, assay, and genomic location, which helps to mathematically center all of the data in the tensor around the same point so that the patterns that we want the model to learn are not obscured by trivial differences in scale along the axes. It is a common strategy for models like PARAFAC that perform best on data that is all on the same scale.

1c. What undesirable properties do the last few terms intend to penalize?

The last few terms of the objective implement L2 regularization on the latent factor matrices. Models like PREDICTD have a vast number of free parameters that can theoretically take on any value to fit the training data. The resulting flexibility can be an undesirable property if the model overfits the training data, because it will then fail to generalize well to other data. In the case of PREDICTD, this means that the model would be able to reconstruct the values in the original tensor with high fidelity but that the predicted values for the missing data would not be very good. Regularization helps to restrict the possible values for the free parameters (and thus avoid overfitting the training data) by adding the sum of the squared values of all of the factor matrix elements to the function we are trying to minimize. The L2 regularizer particularly penalizes large parameter values; therefore, with L2 regularization the training procedure will strongly prefer a model with small parameter values. We added the following explanation to the text of the model description in the methods section:

The objective function (Eq. (1)) has two main parts. The first part calculates the squared error between the training data, $\mathcal{D}_{j,k,i}^{\text{train}}$, and the model’s prediction, $\sum_{l=1}^L \mathbf{C}_{j,l} * \mathbf{A}_{k,l} * \mathbf{G}_{i,l} + \mathbf{c}_j + \mathbf{a}_k + \mathbf{g}_i$. This term penalizes the distance between the imputed and observed data. The last three terms, $\lambda_{\mathbf{C}} \|\mathbf{C}\|_2^2 + \lambda_{\mathbf{A}} \|\mathbf{A}\|_2^2 + \lambda_{\mathbf{G}} \|\mathbf{G}\|_2^2$, implement L2 regularization on the factor matrices. This type of regularization penalizes large parameter values, and thus causes the model to strongly prefer a solution with small values on the parameters. Such regularization helps to reduce the flexibility of the model and helps to avoid overfitting the training data. Furthermore, we note that our choice of PARAFAC, which is a linear model with a limited number of latent dimensions, is itself a form of regularization in the sense that such a model is less flexible than more complex models like deep neural nets. PARAFAC is therefore inherently less prone to overfitting the training data compared to a non-linear model given the same model dimensionality.

1d. How is equation (1) solved? The authors do not go into any details but only cite a publication. It is advisable to briefly describe how that method works for the sake of self-contain.

We added more details about stochastic gradient descent to our description of the PREDICTD model:

Equation (1) cannot be solved analytically, so we solve it numerically using stochastic gradient descent (SGD). In SGD, we first initialize the three factor matrices with random values from a uniform distribution on the domain (-0.33 to 0.33) and the three bias vectors with the mean value

from each corresponding plane in the tensor. Then we randomly iterate over the training set data points in the tensor, at each iteration calculating the gradient of the objective function (Eq. (1)) with respect to each factor matrix and bias vector, and then adding a fraction of this gradient to the corresponding parameter values. Over time, as more and more gradients are calculated and used to update the parameter values in the factor matrices and bias vectors, the model as a whole “moves” along the high-dimensional surface defined by the objective function and “down” toward a minimum that (ideally) represents a good solution. We track the model’s progress toward this solution by periodically saving the value of the mean squared error on the held-out validation data points. Eventually, the validation mean squared error stops decreasing, which indicates that the model parameters have converged on a solution. Importantly, there is no guarantee that this solution is the best possible one, as in the case of PREDICTD (and PARAFAC more generally) the objective function is not convex.

2a. A central issue that needs clarification is how missing data come into play in the model. It is good to know that the training data D does not have to be complete, but it is unclear how missing data are accounted for in model inference, ...

We have added the following text to the manuscript to clarify how the missing values are filled in during the calculation of imputed values:

Briefly, in this procedure the three-dimensional tensor is factored into three low-rank matrices, each with the same (user-specified) number of column vectors. These column vectors are called “latent factors,” and the tensor is reconstructed by summing the outer products of the corresponding latent factor vector triplets. These factor matrices have no missing values, so when they are combined to reconstruct the original data tensor, the reconstructed tensor contains imputed data values that not only approximate the existing tensor data, but also fill in the missing values.

And we also more clearly explained in our modeling description quoted above that during stochastic gradient descent, the parameter values are updated based on gradient calculations on the training data points only. Therefore, during model training, the missing values in the tensor are simply ignored. Here is the relevant sentence:

Then we randomly iterate over the training set data points in the tensor, at each iteration calculating the gradient of the objective function (Eq. (1)) with respect to each factor matrix and bias vector, and then adding a fraction of this gradient to the corresponding parameter values.

2b. ...or how to apply the model to impute data in a test dataset.

We agree that our manuscript and PREDICTD documentation could be more clear on how to impute data for a new cell type, so we added a new tutorial and additional documentation to this effect on the GitHub site (<https://github.com/tdurham86/PREDICTD/wiki>). For details, please see our response to comment 1c from Reviewer #1. We also added the following text to the manuscript:

We provide a tutorial on the Github site (<https://github.com/tdurham86/PREDICTD/wiki>) that details how a user can train a PREDICTD model to generate imputed data for a new cell type. Briefly, a user can upload $-\log_{10} p$ -value tracks in bigWig format to an Amazon S3 bucket, and then PREDICTD will add that data to the Roadmap Epigenomics tensor, train the model, and write imputed data for the new cell type back to S3 in bigWig format. The tutorial demonstrates how this is done with seven data sets from the Fetal Spinal Cord cell type that we downloaded from the ENCODE portal (<http://www.encodeproject.org>).

3. In Fig.2, the authors should also show results from ChromImpute for comparison.

We generated additional supplemental figures for manuscript Figure 2 that include the results for ChromImpute. We include them here as Fig. 10, Fig. 11, and Fig. 12. These supplemental figures are referenced from

the ChromImpute section of the paper as manuscript Figures S4, S5, and S6, respectively. We believe that including ChromImpute results directly in manuscript Figure 2 in the main text would disrupt the flow of the paper and be too confusing to the reader, as the purpose of Figure 2 is to describe PREDICTD itself, and ChromImpute has not yet been introduced at this point in the paper.

Figure 10: Selected tracks comparing PREDICTD imputed signal, ChromImpute imputed signal, and observed signal for three assays and six cell types.

4. For the three cell lines that PREDICTD does not outperform the “main effects” model; it will be helpful for the readers to examine why.

There are three assays for which the main effects model outperforms PREDICTD on the global mean squared error (MSE) quality measure: H2BK5ac, H3K23ac, and H4K8ac. Of these, H3K23ac seems to be the most difficult for PREDICTD to impute. We first hypothesized that the observed data for these assays could be more noisy or of otherwise lower quality than that for the other assays. We plotted the mean value for each assay for a variety of measures (Fig. 13, this document), including the distribution of the number of data sets available (NEXPT), the number of sequencing reads available (NREADS), and scores from seven different quality measures published by the Epigenomics Roadmap Consortium (for the quality measure data see goo.gl/6umkG9, and for more detailed information about these quality measures, see <https://genome.ucsc.edu/ENCODE/qualityMetrics.html#definitions> and <http://www.roadmapepigenomics.org/quality-metrics>):

- **NSC (Signal to noise):** To calculate the Normalized Strand Cross-correlation coefficient (NSC), reads mapping to the forward and reverse strands are treated separately and shifted in opposite directions at multiple distances relative to each other. For each distance, the correlation is calculated between the data on the two strands. The maximum correlation will be found at a shift that is equal to the dominant fragment length in the library. The score is calculated as the ratio between the maximum correlation and the minimum.
- **RSC (Phantom peak):** The Relative Strand Cross-correlation coefficient (RSC) is calculated as the difference between the maximum and minimum strand cross-correlation as calculated for NSC, normalized by the difference between the strand cross-correlation at a shift corresponding to the read

Figure 12: Limited data training results for CD3 Cord Blood cell type for all quality measures and compared to ChromImpute.

length for that data set and the minimum strand cross-correlation. This corrects for an artifact in some data sets related to the mapping biases of short reads.

- **Signal Portion of Tags (SPOT):** “Hotspots” of signal are called based on testing read counts against local background using z-scores based on a local background estimate. The quality measure is the percentage of reads falling in these called hotspot regions.
- **FindPeaks:** Peaks are called at an FDR of 1% using the FindPeaks software package [18] and the score is calculated as the percentage of reads mapping to the genome in these called peak regions.
- **Poisson:** The genome is split into 1 kb bins and a Poisson distribution is fit to the number of sequencing reads that fall into each bin. The value reported is the percentage of total reads that are found in “enriched” bins (Poisson p-value < 0.01).
- **Pct 1% Imp in 1% Obs:** Observed data and imputed data from ChromImpute are sorted on genomic position by signal value, and then the percentage of the top 1% of genomic positions by imputed signal that are also found in the top 1% of genomic positions by observed signal is reported.
- **Imp/Obs Corr:** The correlation coefficient between ChromImpute imputed data and the corresponding observed data.

Of the three assays that are better imputed by main effects, none are particularly striking outliers by any of these measures; however, H3K23ac is among the lowest-scoring assays on the FindPeaks and Poisson measures and is the worst-scoring assay on the “Pct 1% Imp in 1% Obs” and “Imp/Obs Corr” measures. Indeed, these assays (and H3K23ac in particular) are among the harder assays to impute for ChromImpute as well as PREDICTD.

Given that the data for the H2BK5ac, H3K23ac, and H4K8ac assays do not seem to be of particularly low quality, we tried to think of another quality of the observed data that could improve the performance of main effects versus PREDICTD or ChromImpute. We reasoned that because main effects operates on simple genomic position-wise averages of the data sets for a given cell type or assay, it may be that these assays are simply more consistent across cell types than others. This could potentially give an advantage to the simpler main effects approach if the more complex PREDICTD/ChromImpute algorithms attempt to apply patterns derived from other assay/cell type combinations that end up injecting more variance into the signal for these three assays than is actually present. To test this, we calculated the pairwise correlation between the observed data for the cell types that have H2BK5ac, H3K23ac, and H4K8ac experiments available, and plotted the mean correlation for each assay (Fig. 14, this document). If our hypothesis holds we would expect to see higher mean levels of correlation for the H2BK5ac, H3K23ac, and H4K8ac assays than others; however, we actually see the opposite – these three assays have rather low mean correlation among observed datasets.

In the end, we could not find a single cause for these assays being imputed better by main effects than PREDICTD. It is important to note that the performance gain for main effects is relatively small and is well-within the standard error of the mean of PREDICTD performance. We suspect that a major contributing factor is the stochastic nature of the training of the imputation approaches, both in terms of the random initialization and training of the factor matrices and the semi-random selection of genomic positions for training. Particularly in the case of H3K23ac, there may also be some issues with the data that make it a particularly difficult assay to impute. It has similar Roadmap Consortium quality score values to broad marks like H3K9me3, which suggests that it has more diffuse signal and potentially more noise to confuse the training of more complex models like PREDICTD and ChromImpute.

5. The authors pointed out that in some cases, using only two histone marks is sufficient to reproduce the entire epigenome. I wonder if this is because the cell line in question is highly similar to one included in the training dataset.

We agree that this is an important concern, and we have addressed this by running the same experiment on four additional cell types from varying tissues: “GM12878 Lymphoblastoid”, “Fetal Muscle Trunk”, “Brain Anterior Caudate”, and “Lung.” In each case, training on only data from the H3K4me3 assay provides surprisingly good performance. See our response to Reviewer #1, comment 5, and Fig. 4 (this document,

Figure 13: Quality measures for observed data as reported by the Epigenomics Roadmap Consortium. Red bars highlight the three assays (H2BK5ac, H3K23ac, and H4K8ac) for which main effects outperforms PREDICTD.

Figure 14: Pairwise correlations for the observed data for seven cell types (“H1 BMP4 Derived Mesendoderm Cultured Cells,” “H1 BMP4 Derived Trophoblast Cultured Cells,” “H1 Cell Line,” “H1 Derived Mesenchymal Stem Cells,” “H1 Derived Neuronal Progenitor Cultured Cells,” “H9 Cell Line,” and “IMR90 Cell Line”) were calculated per assay. Bars show the mean correlation per assay, and the error bars indicate the standard error of the mean. Red bars highlight the three assays (H2BK5ac, H3K23ac, and H4K8ac) for which main effects outperforms PREDICTD.

Fig. S7 in manuscript), along with Figs. 5, 6, 7, 8 (all this document, Figs. S8-S11 in the manuscript). Note that we also included the corresponding ChromImpute performance values for these experiments.

6. For model comparison, it would be helpful to describe what each performance parameter means.

We provided a short description of each quality measure in the Methods section, but we apologize for not making the descriptions more clear. We revised this section of the methods to include better descriptions of each quality measure as follows:

We implemented ten different quality assessment metrics (listed below), the last seven of which were first reported for ChromImpute [1]. We report these metrics for held out test set experiments and compute them over the ENCODE Pilot Regions (Fig. S17).

- **MSEglobal:** Mean squared error between the imputed and observed values at all available genomic positions.
- **MSE1obs:** Mean squared error between the imputed and observed values in the top 1% of genomic positions ranked by the observed signal values.
- **MSE1imp** Mean squared error between the imputed and observed values in the top 1% of genomic positions ranked by the imputed signal values.
 - **MSE1imppred:** Mean squared error between the imputed and observed values in the top 1% of genomic positions ranked by the signal values imputed by PREDICTD.
 - **MSE1impchrmp:** Mean squared error between the imputed and observed values in the top 1% of genomic positions ranked by the signal values imputed by ChromImpute.
 - **MSE1impme:** Mean squared error between the imputed and observed values in the top 1% of genomic positions ranked by the signal values imputed by Main Effects.
- **GWcorr:** Pearson correlation between imputed and observed values at all available genomic positions.
- **Match1:** Percentage of the top 1% of genomic positions ranked by observed signal that are also found in the top 1% of genomic positions ranked by imputed signal.
- **Catch1obs:** Percentage of the top 1% of genomic positions ranked by observed signal that are also found in the top 5% of genomic positions ranked by imputed signal.
- **Catch1imp:** Percentage of top 1% of genomic positions ranked by imputed signal that are also found in the top 5% of genomic positions ranked by observed signal.
- **AucObs1:** Recovery of the top 1% of genomic positions ranked by observed signal from all genomic positions ranked by imputed signal calculated as the area under the curve of the receiver operating characteristic.
- **AucImp1:** Recovery of the top 1% of genomic positions ranked by imputed signal from all genomic positions ranked by observed signal calculated as the area under the curve of the receiver operating characteristic.
- **CatchPeakObs:** Recovery of genomic positions at called peaks in observed signal from all genomic positions ranked by imputed signal calculated as the area under the curve of the receiver operating characteristic.

References

- [1] Jason Ernst and Manolis Kellis. Large-scale imputation of epigenomic datasets for systematic annotation of diverse human tissues. *Nature Biotechnology*, advance online publication, February 2015.
- [2] Richard A. Harshman. Foundations of the PARAFAC procedure: Models and conditions for an “explanatory” multi-modal factor analysis. *UCLA Working Papers in Phonetics*, 16(1), 1970.

- [3] James Bergstra and Yoshua Bengio. Random search for hyper-parameter optimization. *Journal of Machine Learning Research*, 13(Feb):281–305, 2012.
- [4] Jasper Snoek, Hugo Larochelle, and Ryan P. Adams. Practical bayesian optimization of machine learning algorithms. In *Advances in neural information processing systems*, pages 2951–2959, 2012.
- [5] Benjamin Recht. The News on Auto-tuning, June 2016.
- [6] Kai Wei, Maxwell W. Libbrecht, Jeffrey A. Bilmes, and William Stafford Noble. Choosing panels of genomics assays using submodular optimization. *Genome Biology*, 17:229, 2016.
- [7] Rasmus Bro. PARAFAC. Tutorial and applications. *Chemometrics and intelligent laboratory systems*, 38(2):149–171, 1997.
- [8] Tamara G. Kolda and Brett W. Bader. Tensor decompositions and applications. *SIAM review*, 51(3):455–500, 2009.
- [9] Yun Zhu, Zhao Chen, Kai Zhang, Mengchi Wang, David Medovoy, John W. Whitaker, Bo Ding, Nan Li, Lina Zheng, and Wei Wang. Constructing 3d interaction maps from 1d epigenomes. *Nature Communications*, 7:10812, March 2016.
- [10] Dan Benveniste, Hans-Joachim Sonntag, Guido Sanguinetti, and Duncan Sproul. Transcription factor binding predicts histone modifications in human cell lines. *Proceedings of the National Academy of Sciences*, 111(37):13367–13372, September 2014.
- [11] John W. Whitaker, Zhao Chen, and Wei Wang. Predicting the human epigenome from DNA motifs. *Nature Methods*, 12(3):265–272, March 2015.
- [12] Jian Zhou and Olga G. Troyanskaya. Predicting effects of noncoding variants with deep learning-based sequence model. *Nature Methods*, 12(10):931–934, October 2015.
- [13] Vishaka Datta, Rahul Siddharthan, and Sandeep Krishna. Detection Of Cooperatively Bound Transcription Factor Pairs Using ChIP-seq Peak Intensities And Expectation Maximization. *bioRxiv*, page 120113, May 2017.
- [14] Marzia A. Cremona, Laura M. Sangalli, Simone Vantini, Gaetano I. Dellino, Pier Giuseppe Pelicci, Piercesare Secchi, and Laura Riva. Peak shape clustering reveals biological insights. *BMC Bioinformatics*, 16, October 2015.
- [15] Gabriele Schweikert, Botond Cseke, Thomas Clouaire, Adrian Bird, and Guido Sanguinetti. MMDiff: quantitative testing for shape changes in ChIP-Seq data sets. *BMC Genomics*, 14:826, November 2013.
- [16] John A. Capra, Genevieve D. Erwin, Gabriel McKinsey, John L. R. Rubenstein, and Katherine S. Pollard. Many human accelerated regions are developmental enhancers. *Philosophical Transactions of the Royal Society B: Biological Sciences*, 368(1632), December 2013.
- [17] J. Douglas Carroll and Jih-Jie Chang. Analysis of individual differences in multidimensional scaling via an n-way generalization of Eckart-Young decomposition. *Psychometrika*, 35(3):283–319, 1970.
- [18] Anthony P. Fejes, Gordon Robertson, Mikhail Bilenky, Richard Varhol, Matthew Bainbridge, and Steven J. M. Jones. FindPeaks 3.1: a tool for identifying areas of enrichment from massively parallel short-read sequencing technology. *Bioinformatics (Oxford, England)*, 24(15):1729–1730, August 2008.

Reviewers' comments:

Reviewer #1 (Remarks to the Author):

My two biggest concerns with the manuscript were (1) the user-friendliness of the software/algorithm, and (2) the danger that the authors may have overfitted to the data (too many latent factors) which would manifest itself as a strong redundancy in the inferred latent factors. While the authors have been very thorough in responding to all of my concerns and I feel that the revised version has indeed improved accordingly, I am not entirely convinced that they have efficiently addressed the redundancy issue, which I admit is a hard problem. Indeed, the data shown in the response, specifically Fig.2 in the response to my reviewer#1 comments, clearly demonstrates (in my opinion), that there is indeed a lot of redundancy, not only in assay space but also in cell-type and genome-space. And Fig1, btw, is not addressing my concern at all, as my concern relates to negative entries in the latent factors, not to whether the imputed values are negative. Fig.1 only shows that the fitted model is good, as many negative imputed values would imply a very poor fitting, yet this figure could also result from a highly overfitted model. The heatmaps shown can also be misleading as individual colors are hard to appreciate given the large number of factors, but also seem to indicate quite a lot of relatively strong cross-correlations between factors. Ideally, a matrix or tensorial decomposition algorithm should result in factors that are not highly correlated. While I agree with the authors that many additional factors could be explained by the fact that they are capturing higher levels of complexity (resulting say from non-linear interactions which a linear model can't faithfully capture), I am not at all convinced that this additional redundancy is helpful.

Indeed, in this revised version it is still unclear, for instance in Fig.3D, whether all the latent factors shown in one of the panels are actually independent of each other. It could still be the case that one of the latent factors with high average values of the P300 peak is effectively the same latent factor as one of those with large negative values, owing to the fact that latent factors are uniquely determined up to a sign and scaling. So, somehow, I doubt very much that the approximately 20 latent factors shown in this panel, all with high absolute values over the P300 peak, are all biologically significant. If the authors wish to claim that all these ~20 latent factors are biologically relevant, they need to demonstrate this.

The authors further defend their choice of 100 latent factors by saying that this leads to best generalization performance, but I am not entirely convinced by this, as optima are hard to assess. In other words, if we were to perturb the data, we might expect big shifts in the "optimal parameter values" including that of latent factors. I realize this is a very hard problem, but I also feel that it would be wrong to imply that there are 100 biologically relevant latent factors. Such claims need to be backed up with evidence. Alternatively, I am not sure why the authors did not run their algorithm with the smallest number of latent factors which yields a reasonably good generalization performance in blind test sets. I think the authors may need to anchor this type of optimization analysis on latent factors that capture some biological ground truth, as this is more likely to predict a lower number of latent factors.

In summary, a non-negative PARAFAC may have overcome some of the issues raised above, but I also understand if the authors don't wish to pursue this. My advice is that the authors make it clear throughout the manuscript that e.g. in Fig.3D only a couple of latent factors shown in this panel are likely to be truly biologically independent, or that they try to dig deeper and provide distinct biological interpretations to some of these latent factors. Otherwise it is misleading for the reader to assume that are so many "different" latent factors correlating with P300 binding sites. So, the authors need to address this important issue.

Reviewer #2 (Remarks to the Author):

I would like to thank the authors for a comprehensive response, and for significantly modifying the manuscript in response to my and the other reviewers' comments. Overall, I'm still a bit on the fence with this manuscript. While it is clearly a more elegant and natural model than ChromImpute, the results are extremely similar (indeed, visually indistinguishable in Fig S4), and, while claims of an easier training procedure are justified, the issue of the choice of latent dimension is very difficult (as clearly described by the authors in the revised manuscript), and the biological interpretability of the latent factors is rather opaque. Nevertheless, it is a well executed study and I am not in principle against publication, given the more enthusiastic feedback of the other reviewers.

I have the following suggestions which I think would improve the paper:

- in response to my question about binary predictors of epigenomic marks, the authors provided a very convincing argument about the importance of capturing the quantitative and spatially varying component of epigenomic marks. This also provides a strong motivation as to why the overall MSE is the best performance metric to use (and the one in which they improve on ChromImpute, albeit marginally). Therefore, I think such a discussion, or an abridged version, would provide a strong motivation as well as a rationale for preferring Predictd to other methods.
- I would still like to see a more careful discussion about interpretability of the model. What is not sufficiently clear is that the training procedure is more transparent, not the model components themselves; I think the authors should indicate clearly in the discussion that a biological interpretation of the latent factors is in general a topic for further analysis in a study by study basis, and not an easy task in general.

Reviewer #3 (Remarks to the Author):

I applaud the authors for an extremely thorough and well-described revision. These efforts have greatly enhanced the clarity of the paper and resolved all the concerns I had in my original review. This method would be a very valuable resource for the epigenomics community.

February 2, 2018

Dear Editor,

Thank you for managing the review of our manuscript, “PREDICTD: PaRallel Epigenomics Data Imputation with Cloud-based Tensor Decomposition.” Below, we address each of the points raised by the three reviewers and describe the changes we have made to the manuscript. In what follows, the reviewer’s comments are given in bold type, interleaved with our responses. Changes in the text of the revised submission are marked by text that is red and indented. Most of the figures are also new, reflecting the additional analyses that we did in response to the reviewers’ requests.

This revision required extensive work, even though at first glance there are not many changes. In light of Reviewer #1’s comments on the redundancy of latent factors and model overfitting, we conducted a thorough validation of our hyperparameter settings. We show below in our response that the results largely support our original modeling choices. Also, given that we have made updates to the PREDICTD code for model training and hyperparameter tuning over the course of the review process, we also took this opportunity to search again for hyperparameters at a dimensionality of 100 latent factors while holding out a final test set, then to re-train models with these new hyperparameters, and finally to update all of the analyses in the paper based on these new models.

We provide details about the updated hyperparameter search and results in an “Additional Updates” section located below our direct responses to the reviewers, but briefly describe the additional changes here. Our findings are essentially the same as before; the most notable change was that the nCHAR analysis now supports five nCHAR clusters instead of six, which was accompanied by an increase in size of the “No signal” cluster from 1847 nCHARs to 2039 nCHARs. We also changed the software that we used for the Gene Ontology analysis to use Genomic Regions Enrichment of Annotations Tool (GREAT) [1], which was designed for doing just this kind of analysis and has a more sophisticated model for matching nCHARs to genes that are possible targets of regulation than the simple nearest gene approach we took previously.

Our extremely thorough work on this round of revision has reinforced our confidence in our results. We feel that the manuscript has again improved, and we hope that you and the reviewers agree.

Thank you very much for your consideration.

Best regards,

Bill Noble
University of Washington

Reviewer #1

My two biggest concerns with the manuscript were (1) the user-friendliness of the software/algorithm, and (2) the danger that the authors may have overfitted to the data (too many latent factors) which would manifest itself as a strong redundancy in the inferred latent factors. While the authors have been very thorough in responding to all of my concerns and I feel that the revised version has indeed improved accordingly, I am not entirely convinced that they have efficiently addressed the redundancy issue, which I admit is a hard problem.

We thank you for your constructive comments on the manuscript, and we agree that it has been much improved by these revisions. As you will find below, in this revision we endeavored to better present our argument that there is not strong redundancy in the latent factors overall (although there is still evidence of a small amount of correlation among the latent factors), and we conducted an extremely thorough hyperparameter search over different latent factors settings and analysis of the mean squared error during training to show that our models do not overfit the training data.

1) Indeed, the data shown in the response, specifically Fig.2 in the response to my reviewer #1 comments, clearly demonstrates (in my opinion), that there is indeed a lot of redundancy, not only in assay space but also in cell-type and genome-space.

We see now that this figure is difficult to interpret in the way that we hoped. The reason that we concluded that there is little overall redundancy in the latent factors is that each factor has a shared identity across all three matrix factors. Therefore, although two latent factors might be correlated and capturing redundant information in, for example, the small assay dimension (indeed, this is not very surprising, since we have 100 latent factors and only 24 assays), those same factors might be capturing very different patterns from each other in the genome dimension. Here we provide a figure showing the distribution of pairwise correlation coefficients (Fig. 1). Although there is some apparent correlation structure, the coefficients are low (average of the absolute value of Pearson's $r = 0.066$) and most of the latent factors are uncorrelated. Given the nonconvexity of the problem and the noisy nature of biological data, we feel that this level of redundancy is acceptable.

Furthermore, we would like to point out that redundancy does not necessarily imply overfitting. We can describe this distinction mathematically with an example. If $Y = f_{\theta}(X)$ is not redundant, with a dimensionality corresponding to $|\theta|$ parameters, then $Y' = \frac{1}{2}(f_{\theta}(X) + f_{\theta}(X))$ is redundant with $2|\theta|$ parameters, but $Y = Y'$ so the models perform the same despite the redundant latent factors. We can also see empirically that the model is not overfit. If overfitting were an issue, we would expect that its prediction performance would generalize poorly to the held out validation set experiments. However, this is not what we see; our stopping criterion is designed to detect when the validation error stops improving during training and halts the training appropriately based on the minimum validation mean squared error (Fig. 2).

2) And Fig1, btw, is not addressing my concern at all, as my concern relates to negative entries in the latent factors, not to whether the imputed values are negative. Fig.1 only shows that the fitted model is good, as many negative imputed values would imply a very poor fitting, yet this figure could also result from a highly overfitted model.

We apologize for misunderstanding the reviewer's concern. We included this figure to show that the model rarely imputes negative values, thereby giving some more insight into why we did not pursue a non-negative constraint. We appreciate that this does not address the question of whether allowing negative values adds to redundancy of latent factors in the model.

3) The heatmaps shown can also be misleading as individual colors are hard to appreciate given the large number of factors, but also seem to indicate quite a lot of relatively strong cross-correlations between factors. Ideally, a matrix or tensorial decomposition algorithm should result in factors that are not highly correlated.

Again, we expect correlation in the factors for the assay and cell type dimensions because the number of latent factors is close to or exceeding the size of these dimensions. However, there is little correlation in the parameters along the genome dimension, which, being several orders of magnitude larger than either the cell type or assay dimension, accounts for the vast majority of parameters in the model. As mentioned above, Fig. 1, in which we now incorporate a diverging and discretized color scale to improve legibility, suggests that

Figure 1: **The overall correlation between latent factors is close to zero.** **A.** Corresponding latent factors from the cell type, assay, and genome dimensions (100000 random genomic positions) were concatenated and then tested for pairwise correlation. **B.** The Pearson correlation coefficient values from the heatmap in A. as a violin plot showing that the vast majority of correlation coefficients are close to zero (average of the absolute value of Pearson’s $r = 0.066$).

the amount of pairwise correlation among the latent factors is actually quite low (average of the absolute value of Pearson’s $r = 0.066$).

4) While I agree with the authors that many additional factors could be explained by the fact that they are capturing higher levels of complexity (resulting say from non-linear interactions which a linear model cant faithfully capture), I am not at all convinced that this additional redundancy is helpful. Indeed, in this revised version it is still unclear, for instance in Fig.3D, whether all the latent factors shown in one of the panels are actually independent of each other. It could still be the case that one of the latent factors with high average values of the P300 peak is effectively the same latent factor as one of those with large negative values, owing to the fact that latent factors are uniquely determined up to a sign and scaling. So, somehow, I doubt very much that the approximately 20 latent factors shown in this panel, all with high absolute values over the P300 peak, are all biologically significant. If the authors wish to claim that all these 20 latent factors are biologically relevant, they need to demonstrate this.

We agree with the reviewer that one should not necessarily expect any particular latent factor to have a specific biological meaning. Each latent factor contributes to imputing signal across the whole genome in conjunction with the corresponding parameter values from the cell type and assay matrices, and we doubt that it would be possible to define a simple biological meaning for any of them. In presenting Figure 5D our objective is simply to show that the genome parameter values are capturing patterns that correspond to a particular class of putative regulatory element in the genome (enhancers defined by P300 ChIP-seq sites); this “P300 peak” pattern is very different from the pattern we see over the gene model, and this observation gives us confidence that PREDICTD is effectively learning patterns that can distinguish functional elements along the genome. We do not intend to make any claims about the biological relevance of individual latent factors. We have modified this section of the manuscript to clarify this as follows:

The fact that PREDICTD performs well on the imputation task implies that the parameters

Figure 2: **Training is halted before validation error increases.** **A.** Training and validation MSE as a function of the number of parallel SGD iterations during PREDICTD training. The initial vertical drop in the error corresponds to the burn-in phase of training before the parallel SGD iterations begin. The jags in the error curves indicate where the stopping criterion was met, so the training procedure reset the parameters to their values from the iteration with the previous minimum validation MSE, halved the learning rate, and continued training. Orange triangles indicate where the minimum validation MSE was achieved during parallel SGD, and the dotted lines indicate the model training and validation MSE after the final second order update of the genome parameters. The second order update decreased the validation MSE from 0.14034 to 0.14026. **B.** A similar plot of the objective value as a function of parallel SGD iterations shows that the objective value decreases and follows the same trend as the training error.

learned by the model captures patterns that can distinguish among different cell types, assays, and genomic positions, and we next present results showing that this is the case. We think it important to note that it would be incorrect at this point to interpret any particular latent factor as having a specific biological meaning. We place no *a priori* constraints on what patterns in the data PREDICTD uses to arrive at a solution, and any signal with relevance to a particular biological feature is likely distributed across multiple latent factors. As such, here we simply show that the parameters, in aggregate, exhibit different patterns between different cell types, assays, and genomic loci; a full investigation of the ways to gain biological insight from these parameters is outside the scope of our present study.

Although we cannot definitively assign semantics to individual latent factors, we find that their values in aggregate show patterns that recover known relationships among the cell types, assays and genomic loci (Fig. S2). Hierarchical clustering on the rows of the cell type factor matrix shows that similar cell types are grouped together (Fig. S2A), producing large clades for embryonic stem cells (magenta), immune cell types (green), and brain tissues (cyan), among others (Fig. S2A, S3). In the same way, assays with similar targets cluster together (Fig. S2B), with the colored clades from top to bottom representing acetylation marks generally associated with active promoters (magenta), marks strongly associated with active regulatory regions (cyan/blue), and broad marks for transcription (red) and repression (green). The assays cluster perfectly except that, biologically, H3K23ac should be grouped with either the active regulatory marks (cyan/blue) or the active acetylation marks (magenta). This is one of the two assays for which PREDICTD failed to outperform the main effects, and it was one of the worst performing assays for ChromImpute as well, so it appears to be a difficult mark to impute. We investigated whether H3K23ac is an outlier assay in data quality, quantity, or variability across cell types, but we found that it is not appreciably different in these respects from other assays on which PREDICTD performs well (data not shown). Nevertheless, most of the cell types and assays cluster correctly, and

these results are highly non-random. We quantified this by comparing our clustering results to randomly shuffled cluster identities using the Calinski-Harabaz Index, which assesses how well the clustering separates the data by comparing the average distance among points between clusters to the average distance among points within clusters (Fig. S4).

For the genome factor matrix, we projected the coordinates of each gene from the GENCODE v19 human genome annotation [2] (<https://www.encodegenes.org/releases/19.html>) onto an idealized gene model that includes nine parts from 5' to 3' in the gene: the promoter, 5' untranslated region (UTR), first exon, first intron, middle exons, middle introns, last intron, last exon, and 3' UTR. This procedure produced a summary of the genome latent factors (Fig. S2C) that, when reading each column of the heat map as a feature vector for a particular location in a gene, shows distinct patterns at different gene components. For example, latent factors that on average have high or low values at regions (i.e. heat map columns) near the transcription start site are different from those with high or low values at other gene components, like exons and introns.

In addition to investigating patterns in the genome parameters at genes, we checked to see whether distal regulatory regions showed a pattern distinct from gene components. P300 is a chromatin regulator that associates with active enhancers [3]. We therefore decided to search for patterns in the genome latent factors at windows +/- 1 kb around annotated P300 sites from published ENCODE tracks (see Methods). Note that no P300 data was used to train PREDICTD. Nevertheless, we find a striking pattern, with many latent factors showing average values of larger magnitude within the 400 bp region surrounding the center of the peak, and some others showing larger average magnitude in a flanking pattern in the bins 200-400 bp away from the peak center (Fig. S2C,D). Again, note that these results do not imply a biological meaning for any particular latent factor; instead, we hypothesize that the genome latent factors as a whole might be useful as features for classification or deeper characterization of genomic elements.

5) The authors further defend their choice of 100 latent factors by saying that this leads to best generalization performance, but I am not entirely convinced by this, as optima are hard to assess. In other words, if we were to perturb the data, we might expect big shifts in the optimal parameter values including that of latent factors. I realize this is a very hard problem, but I also feel that it would be wrong to imply that there are 100 biologically relevant latent factors. Such claims need to be backed up with evidence. Alternatively, I am not sure why the authors did not run their algorithm with the smallest number of latent factors which yields a reasonably good generalization performance in blind test sets. I think the authors may need to anchor this type of optimization analysis on latent factors that capture some biological ground truth, as this is more likely to predict a lower number of latent factors.

In response to this comment and to verify our choice of model dimensionality, we conducted a large-scale hyperparameter search in which we ran Spearmin [4] at fixed numbers of latent factors covering the range from 2 up to 512 latent factors (Figs. S1). This search took weeks of compute time and required training over 3000 PREDICTD models. We trained each level of latent factors for at least 50 Spearmin iterations or 40% of the number of latent factors, whichever was more, and only stopped Spearmin after it had additionally trained at least 20 iterations or 15% of the number of latent factors, whichever was more, past its best result. During the search, Spearmin picks the next set of hyperparameter values to test based on fitting a Bayesian model to the results of all past hyperparameter sets; in order to minimize the chance of Spearmin optimizing hyperparameters for a particular set of training data, we allowed the training/validation split and the set of genomic windows used for training to vary randomly over the hyperparameter search. Subsequently, to make the results comparable once the hyperparameter search was complete (Fig. S1), we took the best hyperparameter settings found for each number of latent factors and trained ten additional models on a fixed validation set and fixed set of genomic windows, only allowing the random initialization of the factor matrices to vary, and compared the validation MSE distributions of the resulting models (Fig. 3). The results show that, in terms of validation MSE, the models with 100 latent factors perform significantly better than the models at 64 latent factors (Wilcoxon rank-sum test $p < 0.05$), but essentially the same as the best model at 128 latent factors. Thus, the improvement in validation MSE as a function of model dimensionality seems to mostly level off somewhere between 64 and 100 latent factors. If we chose a dimensionality close to 64, then we would expect to lose some model performance; conversely, if we chose a dimensionality greater than 100,

then we would expect to see more redundancy in the latent factors. We conclude that our choice of 100 latent factors is appropriate given that our goal for this manuscript is to achieve the lowest possible validation MSE of the model with respect to real ChIP-seq or DNase-seq data. That being said, as the reviewer suspected, although 100 latent factors produces the model with the best balance of a low dimensionality and low error in our experiments, reasonably good solutions can be obtained with many fewer latent factors. For example, dropping from 100 latent factors to 16 latent factors (84% reduction in the number of latent factors) results in less than a 10% increase in validation MSE (Fig. 3). If one wanted to attempt to train a model that was optimized for biological interpretability, then starting with a much lower dimensionality would be entirely appropriate.

Figure 3: After doing an extensive hyperparameter search at 17 different latent factor settings (Fig. S1), we used the same training/validation sets and the same genomic positions for training all models, and trained ten models with different random initializations for the best hyperparameter settings for each number of latent factors. For the 32 and 64 latent factor models, we used the hyperparameter settings from the expanded Spearmint search (Fig. S1). The 64 latent factor and 100 latent factor distributions are significantly different (Wilcoxon rank sum $p < 0.05$).

In summary, a non-negative PARAFAC may have overcome some of the issues raised above, but I also understand if the authors dont wish to pursue this. My advice is that the authors make it clear throughout the manuscript that e.g. in Fig.3D only a couple of latent factors shown in this panel are likely to be truly biologically independent, or that they try to dig deeper and provide distinct biological interpretations to some of these latent factors. Otherwise it is misleading for the reader to assume that are so many different latent factors correlating with P300 binding sites. So, the authors need to address this important issue.

We appreciate the reviewer’s concern, and we agree that it is important that readers do not make the mistake of inferring a particular biological meaning for any of the individual latent factors. As detailed above, we have edited the relevant sections of the paper to clarify that we are not trying to claim a biological meaning for any individual latent factor. We only present Figure 3 to give the reader a better sense of what the model “looks like,” and to show that the parameters learned by the model make sense because they can, in total, distinguish different types of cell types, assays, and genomic positions. In future work it will certainly be interesting to try training models that are more amenable to biological interpretation, including focusing on training models with fewer latent factors and a non-negative constraint on the parameters.

Reviewer #2

I would like to thank the authors for a comprehensive response, and for significantly modifying the manuscript in response to my and the other reviewers' comments. Overall, I'm still a bit on the fence with this manuscript. While it is clearly a more elegant and natural model than ChromImpute, the results are extremely similar (indeed, visually indistinguishable in Fig S4), and, while claims of an easier training procedure are justified, the issue of the choice of latent dimension is very difficult (as clearly described by the authors in the revised manuscript), and the biological interpretability of the latent factors is rather opaque. Nevertheless, it is a well executed study and I am not in principle against publication, given the more enthusiastic feedback of the other reviewers.

I have the following suggestions which I think would improve the paper:

1) In response to my question about binary predictors of epigenomic marks, the authors provided a very convincing argument about the importance of capturing the quantitative and spatially varying component of epigenomic marks. This also provides a strong motivation as to why the overall MSE is the best performance metric to use (and the one in which they improve on ChromImpute, albeit marginally). Therefore, I think such a discussion, or an abridged version, would provide a strong motivation as well as a rationale for preferring Predictd to other methods.

We have incorporated these arguments into the discussion section, the relevant section of which now reads as follows:

Last, imputed data represents an important tool for guiding epigenomics studies. It is far cheaper to produce than observed data, closely matches the data observed from experimental assays, and is useful in a number of contexts to generate hypotheses that can be explored in the wet lab. We showed that imputed data can provide insights into nCHARs; and Ernst and Kellis [5] previously showed that imputed data tend to have a higher signal-to-noise ratio than observed data, that imputed data can be used to generate high-quality automated genome annotations, and that regions of the genome with high imputed signal tend to be enriched in single nucleotide polymorphisms identified in genome-wide association studies (GWAS). In addition, raw imputed data includes information about signal amplitude and shape, which can provide insight into the types of regulators and binding events that are producing that signal [6–8]. In contrast, other methods that use epigenomics data for various prediction tasks [9–11] all impute binarized epigenomics signal (i.e. peak calls) and do not preserve peak shape or amplitude. Raw imputed data sets make no assumptions about what questions they will be used to address, and are widely applicable to any study that analyzes ChIP-seq or DNase-seq data. Thus, in conclusion, imputed data can provide insight into cell type-specific patterns of chromatin state and act as a powerful hypothesis generator. With just one or two epigenomics maps from a new cell type, PREDICTD can leverage the entire corpus of Roadmap Epigenomics data to generate high quality predictions of all assays.

2) I would still like to see a more careful discussion about interpretability of the model. What is not sufficiently clear is that the training procedure is more transparent, not the model components themselves; I think the authors should indicate clearly in the discussion that a biological interpretation of the latent factors is in general a topic for further analysis in a study by study basis, and not an easy task in general.

We re-worked the wording of the section about the model parameters, and we hope that it more clearly explains that interpretation of the latent factors is not a trivial task and also worthy of further study. The new text is given in response to Reviewer #1, comment 4.

Reviewer #3

I applaud the authors for an extremely thorough and well-described revision. These efforts have greatly enhanced the clarity of the paper and resolved all the concerns I had in my original review. This method would be a very valuable resource for the epigenomics community.

We thank the reviewer for the helpful feedback that so improved the manuscript.

Additional Updates

As we noted in our cover letter to the editor, in response to reviewer comments we performed extensive validation of our original hyperparameter settings. These experiments inspired us to do a new hyperparameter search for model selection, holding out 153 experiments from test set 0 as a final test set, in the hope that we might find still better settings. After letting SpearMint train 188 PREDICTD models with the number of latent factors fixed at 100, we found a new set of hyperparameters that gives slightly better results than our previously reported settings. As a result, we recomputed all analyses in the paper and updated the figures with the new results. The new results do not change any of our major findings or conclusions in the paper, and the biggest change was that in the ncHAR analysis the new results better support five ncHAR clusters instead of six. Also, during the course of analyzing the new results over ncHARs, we realized that GREAT [1] is a better tool for the Gene Ontology analysis due to its more sophisticated method for connecting regulatory regions to the genes that are fed into the ontology enrichment, so we now report ontology results from using GREAT instead of Panther. In all cases we have updated the corresponding sections in the main text and methods, and highlighted sections with more substantial changes in red in the manuscript. We also provide the updated text for the hyperparameter search and the gene ontology analysis here for your convenience.

Updated hyperparameter search text:

We ran SpearMint multiple times as we developed the PREDICTD model, each time holding out the first test set so that we would have some data to test the generalizability of PREDICTD. Early SpearMint runs and some manual grid search of the hyperparameters suggested that 100 latent factors was a good setting for the model dimensionality. Once we settled on 100 latent factors, we ran SpearMint again to fine tune the learning rate and regularization coefficients. We let it train 188 PREDICTD models with different hyperparameter settings and selected the settings from the model that gave the lowest observed validation MSE. During this process, we discovered that PREDICTD is relatively insensitive to the particular values of the three regularization coefficients λ_C , λ_A , and λ_G , but that it seemed to prefer extremely low values (essentially, no regularization) on at least one of the matrix factors. In contrast, the hyperparameter search revealed that PREDICTD performance depends more heavily on particular values for the learning rate, η , and the second order genome update regularization, λ_{G_2} . We also found that our imputation scheme of averaging eight models trained with different validation sets imposed extra regularization on the ultimate averaged solution, and that to achieve the best generalizability of our averaged solution we had to compensate for this additional regularization by choosing a lower λ_{G_2} than the one suggested by SpearMint. After trying different λ_{G_2} values (Fig. S8), we decided to reduce λ_{G_2} by a factor of 10, because this change led to very little change in MSE. Our final chosen hyperparameter values are given in Table 1.

In addition to using SpearMint for model selection, we also used it to systematically explore the effects of changing the model dimensionality by changing the number of latent factors (Figs. S1, 3). In this hyperparameter search, we fixed the dimensionality at one of 17 levels between 2 and 512 latent factors, and then used SpearMint to optimize the other hyperparameters (η , λ_C , λ_A , λ_G , and λ_{G_2}). We allowed the SpearMint runs with larger numbers of latent factors to train longer to give them more opportunity to explore the more complex solution space of these higher dimensionality models. We used a systematic stopping criterion as follows: each SpearMint search had to train for at least 50 iterations or 40% of the number of latent factors, whichever was more, and had to stop after it had trained at least 20 iterations or 15% of the number of latent factors, whichever was more, past its best result (Fig. S1 blue/red bars). After this search,

we noticed that there was a plateau in the validation MSE from 16 latent factors to 64 latent factors, so to gain more resolution on this range of latent factors, we trained the 32 and 64 Spearmint searches out to 120 iterations. We found that the solutions for both models improved, and 64 latent factors improved more than 32 latent factors, but that neither model found a better solution than 100 latent factors (Fig. S1 brown/orange bars). In order to avoid biasing Spearmint’s choice of hyperparameter settings for a particular validation set or subset of genomic locations, we had allowed the validation set and the training subset of genomic windows to vary randomly over the course of the hyperparameter search. However, this meant that any given best Spearmint result could still be due to the model getting “lucky” and finding a validation set or set of genomic windows that was particularly favorable for training. To convince ourselves that the trend our Spearmint search revealed is real, we took the best hyperparameter settings for each latent factor level (for 32 and 64 latent factors these were the results of the expanded search) and trained ten models each with fixed validation sets and a fixed set of genomic windows, only varying the random initialization of the factor matrices from model to model (Fig. 3). The results show the same trend as a function of model dimensionality as in our original hyperparameter search (Fig. S1). We also verified that the distribution of validation MSE for 64 latent factors differed significantly from the MSEs for 100 latent factors (Wilcoxon rank-sum test $p < 0.05$).

To save time on model training during the Spearmint iterations, we relaxed the convergence criteria to use a larger shift between the two samples in the Wilcoxon rank-sum test ($5e^{-05}$ instead of $1e^{-05}$), and we only did a single line search after the model first converged instead of three. It is important to note that, despite our efforts, there may be even better hyperparameter settings that our search did not encounter. As new discoveries concerning hyperparameter tuning unfold in the machine learning literature the settings for PREDICTD can be revisited to perhaps further improve its performance.

Table 1: **Hyperparameter values.** The third column indicates whether the hyperparameter value was selected using Spearmint, and an asterisk indicates the final value was tuned by hand after Spearmint optimization.

Hyperparameter	Value	Spearmint?
η	0.0045	Y
ϕ_η	$1 - 1e^{-6}$	N
β_1	0.9	N
ϕ_{β_1}	$1 - 1e^{-6}$	N
β_2	0.999	N
L	100	Y*
λ_C	4.792	Y
λ_A	$8.757e^{-27}$	Y
λ_G	$8.757e^{-27}$	Y
λ_{G_2}	0.4122	Y*

Updated ncHAR clustering results text:

These biclustering results also agree with and expand upon previously published tissue specificity predictions from EnhancerFinder [12, 13]. The brain enhancer predictions from that study are visibly enriched in our Brain/ES cluster, and limb and heart predictions are enriched in our clusters showing activity in differentiated, epithelial, and mesenchymal cell types (Fig. S9A). If we treat the EnhancerFinder tissue assignments [12] as another clustering of the ncHARs, we find that they are more similar to our clustering (both for observed and imputed data) than to either background clustering (Fig. S9B). In addition, our results expand on EnhancerFinder by assigning to cell type-associated clusters 289 ncHARs (11% of ncHARs) characterized by EnhancerFinder as either having activity in “other” tissues (98 ncHARs) or no developmental enhancer activity (“N/A”, 191 ncHARs). We also find that our clustering successfully predicts enhancer activity for many functionally validated ncHARs, and furthermore assigns most of them to the correct

cell types (Supplementary Table 4). Briefly, we correctly identify enhancer activity in 10 of 23 ncHARs with evidence in the VISTA database [12, 14], and 6 of 7 ncHARs with validation results suggesting enhancer activity specific to the human allele and not the chimp allele [12]; we find evidence of enhancer identity for one of three ncHARs associated with *AUTS2*, a gene associated with autism spectrum disorder, and this was one of two from that study that showed transgenic enhancer activity [15]; *NPAS3* is a gene associated with schizophrenia that lies in a large cluster of 14 ncHARs, and we find enhancer signal for 7 of them, 6 of which have validated enhancer activity [16]; last, *HAR2* is a ncHAR with validated human-specific limb enhancer activity that clusters with our Brain/ES category [17]. Thus, assessing potential enhancer activity based on the Roadmap Epigenomics data, which encompasses different cell types and developmental stages than ENCODE, agrees with previous results and expands on them to characterize more ncHARs as having potential tissue-specific enhancer activity.

Finally, we asked what types of biological processes these putative enhancers might regulate. We extracted the genomic coordinates of the ncHARs in each cluster and used the Genomic Regions Enrichment of Annotations Tool (GREAT) [1] to test for enriched ontology terms. Using the total list of ncHARs as the background, we found that the Brain/ES cluster of ncHARs is enriched for GO Biological Process terms associated with cell migration in different brain regions; the Epithelial/Mesenchymal cluster shows enrichment for terms associated with tissue development, particularly mesenchymal cell differentiation; and, although there are no significantly enriched GO Biological Process terms for the Non-immune cluster, there are enriched terms from a Mouse Phenotype ontology indicating these ncHARs could be associated with embryonic development and morphology (Fig. S9C, Supplementary Tables 5-12). We found no significantly enriched terms for the Immune cluster.

Updated Gene Ontology methods:

We extracted the genomic coordinates of the ncHARs in each cluster and input these regions to the Genomic Regions Enrichment of Annotations Tool (GREAT) [1] to find enriched ontology terms associated with nearby genes. We used GREAT version 3.0.0 on the human hg19 assembly with the default association rule parameters (Basal+extension: 5000 bp upstream, 1000 bp downstream, 1000000 bp max extension, curated regulatory domains included). We first analyzed each cluster for term enrichment against a whole genome background (Supplementary Tables 5,6,8,10,12), and then ran the test with the same parameters against the list of all ncHARs as the background (Supplementary Tables 7,9,11). No terms were significantly enriched for cluster 0 (No Signal) or cluster 4 (Immune) when using the all ncHAR background, and so we omit these results from the supplement. When reporting the results in the main text we used the default GREAT filters for significant terms: FDR \leq 0.05 for the hypergeometric test with at least a two-fold enrichment over expected.

References

- [1] Cory Y McLean, Dave Bristor, Michael Hiller, Shoa L Clarke, Bruce T Schaar, Craig B Lowe, Aaron M Wenger, and Gill Bejerano. GREAT improves functional interpretation of cis-regulatory regions. *Nature Biotechnology*, 28(5):495–501, May 2010.
- [2] Jennifer Harrow, Adam Frankish, Jose M. Gonzalez, Electra Tapanari, Mark Diekhans, Felix Kokocinski, Bronwen L. Aken, Daniel Barrell, Amonida Zadissa, Stephen Searle, If Barnes, Alexandra Bignell, Veronika Boychenko, Toby Hunt, Mike Kay, Gaurab Mukherjee, Jeena Rajan, Gloria Despacio-Reyes, Gary Saunders, Charles Steward, Rachel Harte, Michael Lin, Cdric Howald, Andrea Tanzer, Thomas Derrien, Jacqueline Chrast, Nathalie Walters, Suganthi Balasubramanian, Baikang Pei, Michael Tress, Jose Manuel Rodriguez, Iakes Ezkurdia, Jeltje van Baren, Michael Brent, David Haussler, Manolis Kellis, Alfonso Valencia, Alexandre Reymond, Mark Gerstein, Roderic Guig, and Tim J. Hubbard. GENCODE: The reference human genome annotation for The ENCODE Project. *Genome Research*, 22(9):1760–1774, September 2012.

- [3] Axel Visel, Matthew J. Blow, Zirong Li, Tao Zhang, Jennifer A. Akiyama, Amy Holt, Ingrid Plajzer-Frick, Malak Shoukry, Crystal Wright, Feng Chen, Veena Afzal, Bing Ren, Edward M. Rubin, and Len A. Pennacchio. ChIP-seq accurately predicts tissue-specific activity of enhancers. *Nature*, 457(7231):854–858, February 2009.
- [4] Jasper Snoek, Hugo Larochelle, and Ryan P. Adams. Practical bayesian optimization of machine learning algorithms. In *Advances in neural information processing systems*, pages 2951–2959, 2012.
- [5] Jason Ernst and Manolis Kellis. Large-scale imputation of epigenomic datasets for systematic annotation of diverse human tissues. *Nature Biotechnology*, advance online publication, February 2015.
- [6] Vishaka Datta, Rahul Siddharthan, and Sandeep Krishna. Detection Of Cooperatively Bound Transcription Factor Pairs Using ChIP-seq Peak Intensities And Expectation Maximization. *bioRxiv*, page 120113, May 2017.
- [7] Marzia A. Cremona, Laura M. Sangalli, Simone Vantini, Gaetano I. Dellino, Pier Giuseppe Pelicci, Piercesare Secchi, and Laura Riva. Peak shape clustering reveals biological insights. *BMC Bioinformatics*, 16, October 2015.
- [8] Gabriele Schweikert, Botond Cseke, Thomas Clouaire, Adrian Bird, and Guido Sanguinetti. MMDiff: quantitative testing for shape changes in ChIP-Seq data sets. *BMC Genomics*, 14:826, November 2013.
- [9] Dan Benveniste, Hans-Joachim Sonntag, Guido Sanguinetti, and Duncan Sproul. Transcription factor binding predicts histone modifications in human cell lines. *Proceedings of the National Academy of Sciences*, 111(37):13367–13372, September 2014.
- [10] John W. Whitaker, Zhao Chen, and Wei Wang. Predicting the human epigenome from DNA motifs. *Nature Methods*, 12(3):265–272, March 2015.
- [11] Jian Zhou and Olga G. Troyanskaya. Predicting effects of noncoding variants with deep learning-based sequence model. *Nature Methods*, 12(10):931–934, October 2015.
- [12] John A. Capra, Genevieve D. Erwin, Gabriel McKinsey, John L. R. Rubenstein, and Katherine S. Pollard. Many human accelerated regions are developmental enhancers. *Philosophical Transactions of the Royal Society B: Biological Sciences*, 368(1632), December 2013.
- [13] Genevieve D. Erwin, Nir Oksenberg, Rebecca M. Truty, Dennis Kostka, Karl K. Murphy, Nadav Ahituv, Katherine S. Pollard, and John A. Capra. Integrating Diverse Datasets Improves Developmental Enhancer Prediction. *PLoS Computational Biology*, 10(6):e1003677, June 2014.
- [14] Axel Visel, Simon Minovitsky, Inna Dubchak, and Len A. Pennacchio. VISTA Enhancer Browsera database of tissue-specific human enhancers. *Nucleic Acids Research*, 35(Database issue):D88–D92, January 2007.
- [15] Nir Oksenberg, Laurie Stevison, Jeffrey D. Wall, and Nadav Ahituv. Function and Regulation of AUTS2, a Gene Implicated in Autism and Human Evolution. *PLOS Genetics*, 9(1):e1003221, January 2013.
- [16] G. B. Kamm, F. Pisciotto, R. Kliger, and L. F. Franchini. The Developmental Brain Gene NPAS3 Contains the Largest Number of Accelerated Regulatory Sequences in the Human Genome. *Molecular Biology and Evolution*, 30(5):1088–1102, May 2013.
- [17] Shyam Prabhakar, Axel Visel, Jennifer A. Akiyama, Malak Shoukry, Keith D. Lewis, Amy Holt, Ingrid Plajzer-Frick, Harris Morrison, David R. FitzPatrick, Veena Afzal, Len A. Pennacchio, Edward M. Rubin, and James P. Noonan. Human-Specific Gain of Function in a Developmental Enhancer. *Science*, 321(5894):1346–1350, 2008.

1 Review Response Supplementary Figures

Figure S1: **An extensive hyperparameter search supports selecting 100 latent factors as the model dimensionality for maximizing imputation performance.** **A.** Minimum validation error MSE for the best models found in each of 17 different Spearmint hyperparameter searches with different numbers of latent factors. The minimum validation MSE decreases as a function of increasing latent factor number until about 100 latent factors, suggesting that this dimensionality maximizes model performance while minimizing the redundancy of latent factors. Furthermore, if we allow the Spearmint hyperparameter search to continue until 120 iterations for 32 and 64 latent factors, we see that the 100 latent factor setting still finds a lower validation MSE. **B.** We required hyperparameter searches to get longer as the model dimensionality increased to allow sufficient time for Spearmint to search the solution spaces that become correspondingly more complex. We trained each level of latent factors for at least 50 Spearmint iterations or 40% of the number of latent factors, whichever was more, and only stopped Spearmint after it had additionally trained at least 20 iterations or 15% of the number of latent factors, whichever was more, past its best result (blue/red bars). We expanded the Spearmint search to 120 iterations for the 32 and 64 latent factor settings (orange/brown bars) to better resolve the solution space for the models that were only slightly less complex than our chosen setting of 100 latent factors.

Figure S2: The model parameters can distinguish among elements in each tensor dimension. Plots show the values (or average values) for the 100 latent factors from one of the models trained for this manuscript, and that these values show patterns that distinguish among cell types, assays, and genomic elements. **A.** Hierarchical clustering of the cell types based on cell type factor matrix values shows that similar cell types tend to cluster together. This is a subset of cell types; for a clustering of all cell types see Fig. S3. **B.** Hierarchical clustering of the assays based on assay factor matrix values shows that similar assays tend to cluster together. **C.** Average values from the genome factor matrix show different patterns at different parts of the gene and P300 peaks called from ENCODE data. **D.** Average values from the genome factor matrix for each latent factor plotted as a line spanning the region +/- 1kb around the center of P300 peaks. Parameter values are centered at zero and plotted based on whether they show the highest magnitude at the peak (left, 64 latent factors) or flanking the peak (right, 36 latent factors).

Figure S3: Hierarchical clustering of all cell types based on parameter values from the cell type factor matrix. For legibility, Fig. S2A shows only the lower 62 rows of this figure.

Figure S4: Hierarchical clustering of cell types and assays by latent factor parameter values results in much better cluster separation than randomly assigning cluster identities. A higher value on the Calinski-Harabaz Index indicates that the clusters are denser and better-separated. Linkage trees were cut at eight clusters for cell types and four clusters for assays.

Figure S5: PREDICTD performs comparably to ChromImpute, and combining the models improves the result. **A.** Schematic describing how a ternary plot relates to Cartesian coordinates. Each experiment (represented by a black dot) is plotted in Cartesian space based on the values of a particular quality score for imputed data (in this example, MSEglobal) from PREDICTD, ChromImpute, and Main Effects. Each point in this space is then projected onto a plane by a vector drawn through the point and the origin. The resulting ternary plot summarizes the relative magnitude of the quality score for the three models. If all models achieve the same quality measure score for a particular experiment, then that point will be projected onto the center of the ternary plot. Deviation towards a point of the triangle indicates that one model has a higher value for that quality measure than the other two, and deviation from the center towards one of the edges of the triangle indicates that one model has a lower value. Color shading of the plot area marks the regions of the ternary plot that indicate superior performance of each model on a particular quality measure. The pattern of the colors changes based on whether it is better to have a low value on that quality measure (as with mean squared error) or a high values (for example, the genome wide correlation). **B.** Comparing PREDICTD, ChromImpute, and main effects models across five quality measures: the global mean squared error (MSEglobal), the genome-wide Pearson correlation (GWcorr), the percent of the top 1% of observed data windows by signal value found in the top 5% of imputed windows (Catch1Obs), the percent of the top 1% of imputed windows by signal value found in the top 5% of observed windows (Catch1Imp), and the area under the receiver operating characteristic curve for recovery of observed peak calls from all imputed windows ranked by signal value (CatchPeakObs). **C.** The same as in B, except that the quality measures for the averaged results of ChromImpute and PREDICTD are plotted along the bottom (red) axis instead of the measures for PREDICTD alone.

Figure S6: Comparison of all quality measures for PREDICTD, ChromImpute, and Main Effects models. The first plot in each triplet is the ternary plot, as described in Fig. S5, the second is the Pearson correlation between the quality measure values for each pair of models, and the third is the distribution of the natural log fold-change in quality measure value between corresponding experiments in pairs of models. Note that the correlation of PREDICTD with ChromImpute is always higher than the correlation between Main Effects and ChromImpute, indicating that PREDICTD tends to agree more with ChromImpute than Main Effects does. In addition, the mean log fold-change between PREDICTD and ChromImpute is always either closer to zero than the log fold-change between Main Effects and ChromImpute, indicating more comparable quality measure values between PREDICTD and ChromImpute, or the mean log fold-change indicates stronger performance by PREDICTD (MSEglobal, MSElimp, MSEimpred and MSElimpme) than ChromImpute. In all, the quality measures show that PREDICTD performs very similarly to ChromImpute, and more so than Main Effects does.

Figure S7: Comparison of all quality measures for PREDICTD, ChromImpute, and Main Effects models on just the 153 held out final test experiments that were not used in hyperparameter tuning. The first plot in each triplet is the ternary plot, as described in Fig. S5, the second is the Pearson correlation between the quality measure values for each pair of models, and the third is the distribution of the natural log fold-change in quality measure value between corresponding experiments in pairs of models. Note that the correlation of PREDICTD with ChromImpute is always higher than the correlation between Main Effects and ChromImpute, indicating that PREDICTD tends to agree more with ChromImpute than Main Effects does. In addition, the mean log fold-change between PREDICTD and ChromImpute is always either closer to zero than the log fold-change between Main Effects and ChromImpute, indicating more comparable quality measure values between PREDICTD and ChromImpute, or the mean log fold-change indicates stronger performance by PREDICTD (MSEglobal, MSE1imp, MSEimpred, MSEimpchrmp, and MSE1impme) than ChromImpute. In all, the quality measures show that PREDICTD performs very similarly to ChromImpute, and more so than Main Effects does.

Figure S8: The validation set MSE for the eight different validation folds corresponding to test set 0 after the second order genome update with different values of λ_{G_2} . We chose $\lambda_{G_2} = 0.412$ (black bars) as a value that imposed approximately the same amount of regularization as the stochastic gradient descent phase of training (blue bars) to avoid having too much regularization when we averaged the eight models together to calculate our final imputed values for the test set.

Figure S9: **Imputation of enhancer marks reveals tissue-specific patterns of enhancer-associated marks at non-coding human accelerated regions (ncHARs).** **A.** Average PREDICTD signal at each ncHAR was compiled for H3K4me1, H3K27ac, and DNase assays from all cell types. The first principal component with respect to the three assays was used in a biclustering to find 6 and 5 clusters along the cell type and ncHAR dimensions, respectively. The inverse hyperbolic sine-transformed signal from each of these assays was summed per cell type and ncHAR, and the resulting values were plotted as a heat map. The column marked with a black triangle at the top designates the color key for the ncHAR clusters. The left-most column, designated with a black circle, identifies ncHARs with predicted tissue-specific developmental enhancer activity based on EnhancerFinder analysis from Capra et al, 2013. **B.** Evaluation of the clustering results with the adjusted Rand index. The clustering results for observed data and PREDICTD for the ncHAR and cell type clusterings, and also those from Capra, et al. 2013 for the ncHARs, all show higher adjusted Rand index scores than the clustering results for observed data with shuffled ncHAR coordinates (Shuffled) or for observed data from non-enhancer-associated marks (Other). **C.** We used GREAT to find enriched ontology terms associated with genes that are possibly regulated by ncHARs from each cluster. The list of all ncHARs was used as the background, and the terms are significant at FDR ≤ 0.05 for the hypergeometric test and have at least a two-fold enrichment over expected.

REVIEWERS' COMMENTS:

Reviewer #1 (Remarks to the Author):

The authors should be congratulated on a very thorough and detailed response. Although I am still not entirely convinced that all of the inferred latent factors are biologically relevant (which in my opinion still points towards a degree of overfitting), this study is really interesting and of great importance to the community. So, I feel this paper should be published.

Reviewer #2 (Remarks to the Author):

I thank the authors for the extensive additional work. I think the manuscript has now a good balance in its claims and can be accepted.